# Tree-Based Diffusion Schrödinger Bridge with Applications to Wasserstein Barycenters

**Maxence Noble**[*]
CMAP, CNRS, École polytechnique,
Institut Polytechnique de Paris,
91120 Palaiseau, France

**Valentin De Bortoli**
Computer Science Department,
ENS, CNRS, PSL University

**Arnaud Doucet**
Department of Statistics,
University of Oxford, UK

**Alain Oliviero Durmus**
CMAP, CNRS, École polytechnique,
Institut Polytechnique de Paris,
91120 Palaiseau, France

## Abstract

Multi-marginal Optimal Transport (mOT), a generalization of OT, aims at minimizing the integral of a cost function with respect to a distribution with some prescribed marginals. In this paper, we consider an entropic version of mOT with a tree-structured quadratic cost, i.e., a function that can be written as a sum of pairwise cost functions between the nodes of a tree. To address this problem, we develop Tree-based Diffusion Schrödinger Bridge (TreeDSB), an extension of the Diffusion Schrödinger Bridge (DSB) algorithm. TreeDSB corresponds to a dynamic and continuous state-space counterpart of the multi-marginal Sinkhorn algorithm. A notable use case of our methodology is to compute Wasserstein barycenters which can be recast as the solution of a mOT problem on a star-shaped tree. We demonstrate that our methodology can be applied in high-dimensional settings such as image interpolation and Bayesian fusion.

## 1 Introduction

In the last decade, computational Optimal Transport (OT) has shown great success with applications in various fields such as biology (Schiebinger et al., 2019; Bunne et al., 2022), shape correspondence (Su et al., 2015; Feydy et al., 2017; Eisenberger et al., 2020), control theory (Bayraktar et al., 2018; Acciaio et al., 2019) and computer vision (Schmitz et al., 2018; Carion et al., 2020). While OT commonly seeks at computing the transport plan that minimizes the cost of moving between two distributions, it can naturally be extended to the multi-marginal setting (mOT) when considering several distributions. This extension of OT has notably been studied in quantum chemistry (Cotar et al., 2013), clustering (Cuturi & Doucet, 2014) and statistical inference (Srivastava et al., 2018). In particular, a popular application in unsupervised learning of mOT with Euclidean cost consists in computing the Wasserstein barycenter of a set of probability distributions (Agueh & Carlier, 2011; Benamou et al., 2015; Álvarez-Esteban et al., 2016; Peyré et al., 2019).

Interior point methods can be used to solve OT and mOT problems but they come with computational challenges (Pele & Werman, 2009). In order to mitigate these limitations, one often considers an *entropic regularization* of OT, known as Entropic OT (EOT). This regularized formulation can be efficiently solved in discrete state-spaces using the celebrated *Sinkhorn* algorithm (Cuturi, 2013; Knight, 2008; Sinkhorn & Knopp, 1967), which admits a continuous state-space counterpart referred

---

[*]Corresponding author. Contact at: maxence.noble-bourillot@polytechnique.edu.

37th Conference on Neural Information Processing Systems (NeurIPS 2023).

to as the *Iterative Proportional Fitting* (IPF) procedure (Fortet, 1940; Kullback, 1968; Ruschendorf, 1995). In the case of a quadratic cost, EOT is equivalent to the *static* formulation of the Schrödinger Bridge (SB) problem (Schrödinger, 1932). Given a reference diffusion with finite time horizon $T$ and two probability measures, solving SB amounts to finding the closest diffusion to the reference (in terms of Kullback–Leibler divergence on path spaces) with the given marginals at times $t = 0$ and $t = T$. This framework naturally arises in stochastic control (Dai Pra, 1991) where one aims at controlling the marginal distribution of a stochastic process at a fixed time. Recently, De Bortoli et al. (2021) introduced Diffusion Schrödinger Bridge (DSB), an approximation of a *dynamic* version of the IPF scheme on path spaces, see also Vargas et al. (2021); Chen et al. (2022). This methodology leverages advances in the field of denoising diffusion models (Song et al., 2021; Ho et al., 2020) in order to derive a scalable and efficient scheme to solve SB, and thus EOT.

Similarly to OT, mOT admits an entropic regularization (EmOT), which can be solved via a multi-marginal generalization of Sinkhorn/IPF algorithm (Benamou et al., 2015; Marino & Gerolin, 2020). Recently, Haasler et al. (2021) proposed an extension of the *static* SB problem in *discrete* state-space to any multi-marginal tree-based setting. They notably made the correspondence between this formulation and EmOT, when the cost function writes as the sum of interaction energies onto the given tree structure, and introduced an efficient version of Sinkhorn algorithm to solve it.

**Motivations and contributions.** In this work, we investigate the *continuous* and *dynamic* counterpart of the tree-based framework from Haasler et al. (2021). To be more specific, we present an extension of the static SB formulation in continuous state-space to any multi-marginal tree-based setting, referred to as TreeSB. Then, we establish the equivalence between TreeSB and a formulation of EmOT relying on a (quadratic) tree-structured cost function, analogously to Haasler et al. (2021). Inspired by DSB, we develop TreeDSB, a dynamic counterpart of the multi-marginal IPF (mIPF) to solve it, by operating on path spaces and using score-based diffusion techniques. To bridge gaps in literature, we prove the convergence of mIPF iterations in a *non-compact* setting under mild assumptions, by extending results on IPF convergence (Ruschendorf, 1995). Finally, we illustrate our approach on examples of Wasserstein barycenters from statistical inference and image processing.

Although our approach can be applied to any tree, we focus on *star-shaped trees*. In this setting, we show that TreeSB reduces to a regularized Wasserstein barycenter problem. Our method comes with several benefits compared to existing works. First, it is out-of-sample, *i.e.*, it does not require re-running the full procedure when given a new data point. Second, our formulation of the Wasserstein barycenter problem obtained from TreeSB allows us to avoid numerical issues of having to choose the regularization too small, see Section 5. Finally, to the best of our knowledge, this is the first methodology to extend ideas from diffusion-based models to the computation of Wasserstein barycenters. In particular, we believe that the idea of iterative refinement, *i.e.*, solving the *dynamic* counterpart of a *static* problem, plays a key role in the efficiency and scalability of the method.

**Notation.** For any measurable space $(\mathsf{X}, \mathcal{X})$, we denote by $\mathscr{P}(\mathsf{X})$ the space of probability measures defined on $(\mathsf{X}, \mathcal{X})$. Unless specified, $\mathcal{X}$ is defined as the Borel sets on $\mathsf{X}$. For any $\ell \in \mathbb{N}$, let $\mathscr{P}^{(\ell)} = \mathscr{P}((\mathbb{R}^d)^\ell)$; we denote $\mathscr{P}^{(1)}$ by $\mathscr{P}$. Assume that $\mathsf{X} = (\mathbb{R}^d)^\ell$ for some $\ell \in \mathbb{N}$. For any $x \in \mathsf{X}$ and any $m, n \in \{0, \ldots, \ell\}$ such that $m \leq n$, let $x_{m:n} = (x_m, x_{m+1}, \ldots, x_n)$. Let Leb be the Lebesgue measure. For any non-negative function $f : \mathsf{X} \to \mathbb{R}_+$, such that $\int_{\mathsf{X}} f \mathrm{d}\mathrm{Leb} < +\infty$, define $\mathrm{H}(f) = -\int_{\mathsf{X}} f \log f \mathrm{d}\mathrm{Leb} \in (-\infty, +\infty]$. For any distribution $\mu \in \mathscr{P}(\mathsf{X})$, we define the entropy of $\mu$ as $\mathrm{H}(\mu) = \mathrm{H}(\mathrm{d}\mu/\mathrm{d}\mathrm{Leb})$ if $\mu \ll \mathrm{Leb}$ and $\mathrm{H}(\mu) = +\infty$ otherwise. For any two arbitrary measures $\mu$ and $\nu$ defined on $(\mathsf{X}, \mathcal{X})$, define the Kullback–Leibler divergence between $\mu$ and $\nu$ as $\mathrm{KL}(\mu|\nu) = \int_{\mathsf{X}} \log(\mathrm{d}\mu/\mathrm{d}\nu)\mathrm{d}\mu - \int_{\mathsf{X}} \mathrm{d}\mu + \int_{\mathsf{X}} \mathrm{d}\nu$ if $\mu \ll \nu$ and $\mathrm{KL}(\mu \mid \nu) = +\infty$ otherwise. For any $T > 0$, we denote by $\mathrm{C}([0, T], \mathbb{R}^d)$ the space of continuous functions from $[0, T]$ to $\mathbb{R}^d$. For any path measure $\mathbb{P} \in \mathscr{P}(\mathrm{C}([0, T], \mathbb{R}^d))$, we denote by $\mathrm{Ext}(\mathbb{P}) \in \mathscr{P}^{(2)}$ the coupling between the *extremal* distributions of $\mathbb{P}$, *i.e.*, $\mathrm{Ext}(\mathbb{P}) = \mathbb{P}_{0,T}$. Note that, for a given coupling $\pi_{0,T} \in \mathscr{P}^{(2)}$, there may exist several path measures $\mathbb{P}$ verifying $\mathrm{Ext}(\mathbb{P}) = \pi_{0,T}$. For any undirected tree $\mathsf{T} = (\mathsf{V}, \mathsf{E})$ with vertices $\mathsf{V}$ and edges $\mathsf{E}$, we denote by $\{v, v'\}$ (or $\{v', v\}$) the undirected edge between $v \in \mathsf{V}$ and $v' \in \mathsf{V}$, if it exists. Given $r \in \mathsf{V}$, we denote by $\mathsf{T}_r = (\mathsf{V}, \mathsf{E}_r)$ the directed version of $\mathsf{T}$ rooted in $r$, where the directed edges $\mathsf{E}_r$ are uniquely defined from the edges $\mathsf{E}$, see Appendix B for further details. In this case, the edge linking $v \in \mathsf{V}$ to $v' \in \mathsf{V}$ in $\mathsf{T}_r$ is denoted by $(v, v')$. Finally, for any integers $(n, K) \in \mathbb{N} \times \mathbb{N}^*$, we define $n \bmod (K)$ as the the remainder of the Euclidean division of $n$ by $K$.

## 2 Background and setting

**Multi-marginal optimal transport.** Let $\ell \in \mathbb{N}^*$. Given a cost function $c : (\mathbb{R}^d)^{\ell+1} \to \mathbb{R}$, a subset $\mathsf{S} \subset \{0, \ldots, \ell\}$ and a family of probability measures $\{\mu_i\}_{i \in \mathsf{S}} \in \mathscr{P}^{|S|}$, mOT consists in solving

$$\pi^\star = \arg\min \left\{ \int c(x_{0:\ell}) \mathrm{d}\pi(x_{0:\ell}) : \pi \in \mathscr{P}^{(\ell+1)},\ \pi_i = \mu_i\,, \forall i \in \mathsf{S} \right\}, \qquad \text{(mOT)}$$

where $\pi_i$ is the $i$-th marginal of $\pi$, *i.e.*, $\pi_i(\mathsf{A}) = \pi(\mathrm{proj}_i^{-1}(\mathsf{A}))$ for any $\mathsf{A} \in \mathcal{B}(\mathbb{R}^d)$, with $\mathrm{proj}_i : x_{0:\ell} \mapsto x_i$. Given some weights $(w_i)_{i \in \{1, \ldots, \ell\}} \in (\mathbb{R}_+)^\ell$, the Wasserstein barycenter between the measures $\{\mu_i\}_{i \in \mathsf{S}}$ is given by $\pi_0^\star$ in (mOT), in the case where $\mathsf{S} = \{1, \ldots, \ell\}$ and $c(x_{0:\ell}) = \sum_{i=1}^\ell w_i \|x_0 - x_i\|^2$ (Peyré et al., 2019). In particular, when $w_i = 1/\ell$, the distribution $\pi_0^\star$ can be regarded as the Fréchet mean (Karcher, 2014) of the measures $\{\mu_i\}_{i \in \mathsf{S}}$ for the Wasserstein distance of order 2. Similarly to OT, (mOT) can be relaxed using the following entropic regularization

$$\pi^\star = \arg\min \left\{ \int c(x_{0:\ell}) \mathrm{d}\pi(x_{0:\ell}) + \varepsilon \mathrm{KL}(\pi|\nu) : \pi \in \mathscr{P}^{(\ell+1)},\ \pi_i = \mu_i\,, \forall i \in \mathsf{S} \right\}, \qquad \text{(EmOT)}$$

where $\varepsilon > 0$ is a hyperparameter and $\nu$ is an arbitrary measure defined on $((\mathbb{R}^d)^{\ell+1}, \mathcal{B}((\mathbb{R}^d)^{\ell+1}))$.

**Link with Schrödinger Bridge.** We first recall the relationship between Schrödinger Bridge and EOT. Given $T > 0$, $\mathbb{Q}$ a (reference) path measure, *i.e.*, $\mathbb{Q} \in \mathscr{P}(\mathrm{C}([0, T], \mathbb{R}^d))$ and two measures $\mu_0, \mu_1 \in \mathscr{P}(\mathbb{R}^d)$, solving the SB problem amounts to finding the path measure $\mathbb{P}^\star$ defined by

$$\mathbb{P}^\star = \arg\min\{\mathrm{KL}(\mathbb{P}|\mathbb{Q}) \,:\, \mathbb{P} \in \mathscr{P}(\mathrm{C}([0, T], \mathbb{R}^d)),\ \mathbb{P}_0 = \mu_0,\ \mathbb{P}_T = \mu_1\}. \qquad \text{(SB)}$$

If $\mathbb{Q}$ is associated with a Stochastic Differential Equation (SDE)[2], of the form $\mathrm{d}\mathbf{X}_t = -a\mathbf{X}_t \mathrm{d}t + \mathrm{d}\mathbf{B}_t$, with $a \geq 0$, then it can be shown, see (Léonard, 2014, Proposition 1) that $\mathbb{P}_{0,T}^\star$ verifies

$$\mathbb{P}_{0,T}^\star = \arg\min\{\mathrm{KL}(\pi|\mathbb{Q}_{0,T}) \,:\, \pi \in \mathscr{P}^{(2)},\ \pi_0 = \mu_0,\ \pi_1 = \mu_1\}. \qquad \text{(static-SB)}$$

This is called the *static* formulation of SB. It can be shown that solving (static-SB) is equivalent to solving EOT with quadratic cost and regularization $\varepsilon = 2\sinh(aT)/a$ if $a > 0$, $\varepsilon = 2T$ if $a = 0$. Moreover, since $\mathbb{P}^\star = \mathbb{P}_{0,T}^\star \otimes \mathbb{Q}_{|0,T}$, where $\mathbb{Q}_{|0,T}$ is the measure $\mathbb{Q}$ conditioned on initial and terminal conditions, solving the *dynamic* problem (SB) is equivalent to solving (static-SB).

Similarly, (EmOT) can be easily rewritten in a *static* multi-marginal SB fashion

$$\pi^\star = \arg\min\{\mathrm{KL}(\pi|\pi^0) \,:\, \pi \in \mathscr{P}^{(\ell+1)},\ \pi_i = \mu_i\,, \forall i \in \mathsf{S}\}, \qquad \text{(mSB-like)}$$

with $(\mathrm{d}\pi^0/\mathrm{dLeb})(x_{0:\ell}) \propto \exp[-c(x_{0:\ell})/\varepsilon](\mathrm{d}\nu/\mathrm{dLeb})(x_{0:\ell})$, where $\pi^0$ is the *reference* measure.

**Diffusion Schrödinger Bridge.** Recently, De Bortoli et al. (2021) introduced Diffusion Schrödinger Bridge (DSB), a numerical scheme to solve (SB). It approximates the iterates of a *dynamic* version of the *Iterative Proportional Fitting* (IPF) scheme (Sinkhorn & Knopp, 1967; Knight, 2008; Peyré et al., 2019; Cuturi & Doucet, 2014), which can be described as follows: consider a sequence of path measures $(\mathbb{P}^n)_{n \in \mathbb{N}}$ such that $\mathbb{P}^0 = \mathbb{Q}$ and for any $n \in \mathbb{N}$

$$\mathbb{P}^{2n+1} = \arg\min\{\mathrm{KL}(\mathbb{P}|\mathbb{P}^{2n}) \,:\, \mathbb{P}_T = \mu_1\}, \qquad \mathbb{P}^{2n+2} = \arg\min\{\mathrm{KL}(\mathbb{P}|\mathbb{P}^{2n+1}) \,:\, \mathbb{P}_0 = \mu_0\}.$$

This procedure alternatively projects between the measures with fixed initial distribution and the ones with fixed terminal distribution. For the first iteration, we get that $\mathbb{P}^1 = \mu_1 \otimes \mathbb{Q}_{|T}$. Assuming that $\mathbb{Q}$ is given by $\mathrm{d}\mathbf{X}_t = f_t(\mathbf{X}_t)\mathrm{d}t + \mathrm{d}\mathbf{B}_t$, with $f : [0, T] \times \mathbb{R}^d \to \mathbb{R}^d$, then $\mathbb{P}^1$ is associated with the *time-reversal* of this SDE initialized at $\mu_1$. The time-reversal of an SDE has been derived under mild assumptions on the drift and diffusion coefficients (Haussmann & Pardoux, 1986; Cattiaux et al., 2021). In this case, we have $(\mathbf{Y}_{T-t})_{t \in [0,T]} \sim \mathbb{P}^1$, with $\mathbf{Y}_0 \sim \mu_1$ and

$$\mathrm{d}\mathbf{Y}_t = \{-f_{T-t}(\mathbf{Y}_t) + \nabla \log p_{T-t}(\mathbf{Y}_t)\}\mathrm{d}t + \mathrm{d}\mathbf{B}_t,$$

where $p_t$ is the density of $\mathbb{P}_t^0$ w.r.t. the Lebesgue measure. The score $\nabla \log p_t$ is estimated using score matching techniques (Hyvärinen, 2005; Vincent, 2011). The first iterate of DSB, $\mathbb{P}^1$, corresponds to a *denoising diffusion model* (Ho et al., 2020; Song et al., 2021). DSB iterates further and not only parameterizes the backward process but also the forward process. It can therefore be seen as a refinement of diffusion models drawing a bridge between generative modeling and optimal transport.

---

[2]We refer to Appendix C for details on solutions of SDEs and associated measures.

**Tree-based framework.** Consider an undirected tree $\mathsf{T} = (\mathsf{V}, \mathsf{E})$, with vertices $\mathsf{V}$ and edges $\mathsf{E}$, such that $\mathsf{V}$ is identified with $\{0, \ldots, \ell\}$. Inspired by Haasler et al. (2021), we restrict our study of (EmOT), to the case where the cost function $c$ is the tree-structured *quadratic* cost derived from $\mathsf{T}$

$$c(x_{0:\ell}) = \sum_{\{v,v'\} \in \mathsf{E}} w_{v,v'} \|x_v - x_{v'}\|_2^2 , \tag{1}$$

where $w_{v,v'}$ is a weight on the edge $\{v, v'\}$, which links $v$ to $v'$ (and $v'$ to $v$). Furthermore, as in Haasler et al. (2021), we choose $\mathsf{S}$, *i.e.*, the set of vertices of $\mathsf{T}$ with constrained marginals, to coincide with the *leaves* of $\mathsf{T}$. This framework recovers important applications, from Wasserstein barycenters to Wasserstein propagation, see Solomon et al. (2014, 2015). We emphasize that it differs from an OT problem defined on the space of graphs (Chen et al., 2016). Here, each node represents a probability measure (observed or to be inferred) and each edge represents a coupling between two distributions.

We consider an arbitrary vertex $r \in \mathsf{V}$ and choose $\nu$ in (EmOT) such that $(\mathrm{d}\nu/\mathrm{dLeb})(x_{0:\ell}) = \varphi_r(x_r)$, where $\varphi_r$ is a density defined on $\mathbb{R}^d$. Due to the form of $\nu$ and $c$, the reference measure $\pi^0$ in (mSB-like) is therefore a *probability* distribution which factorizes along $\mathsf{T}_r = (\mathsf{V}, \mathsf{E}_r)$, the directed version of $\mathsf{T}$ rooted in $r$. We refer to Appendix B for more details on the notion of directed trees. In this setting, (EmOT) is equivalent to the tree-based problem

$$\pi^\star = \mathrm{argmin}\{\mathrm{KL}(\pi|\pi^0) \, : \, \pi \in \mathscr{P}^{(|\mathsf{V}|)}, \, \pi_i = \mu_i \, , \forall i \in \mathsf{S}\} , \tag{TreeSB}$$

$$\text{with} \quad \pi^0 = \pi_r^0 \bigotimes_{(v,v') \in \mathsf{E}_r} \pi_{v'|v}^0 , \tag{2}$$

where $\pi_{v'|v}^0(\cdot \mid x_v) = \mathrm{N}(x_v, \varepsilon/(2w_{v,v'})\mathrm{I}_d)$ and $\pi_r^0 \ll \mathrm{Leb}$ with density $\varphi_r$. In a manner akin to Haasler et al. (2021), we thus establish, in *continuous* state-space, the correspondence between (TreeSB), a *static* tree-based version of SB, and a version of EmOT with tree-structured cost (1). In our work, we make the following assumption on the constrained marginals $\{\mu_i\}_{i \in \mathsf{S}}$.

**A0.** *For any $i \in \mathsf{S}$, $\mu_i \ll \mathrm{Leb}$ and* $\mathrm{H}(\mu_i) < \infty$.

In what follows, we define $K$ as the number of leaves of $\mathsf{T}$, denoting $\mathsf{S} = \{i_0, \ldots, i_{K-1}\}$, and define the horizon times $T_{v,v'} = \varepsilon/(2w_{v,v'})$ for any $\{v, v'\} \in \mathsf{E}$. For any $i_k \in \mathsf{S}$, we will denote by $\mathsf{T}_k = (\mathsf{V}, \mathsf{E}_k)$ the directed version of $\mathsf{T}$ rooted in the leaf $i_k$. In the next section, we present our *dynamic* method to solve (TreeSB), called *Tree-based Diffusion Schrödinger Bridge*.

## 3 Tree-based Diffusion Schrödinger Bridge

In this section, we present a method to solve (TreeSB) in the case where $r \in \mathsf{S}$, *i.e.*, $r$ is a leaf of $\mathsf{T}$. We refer to Appendix E for the extension to the case where $r \in \mathsf{V} \backslash \mathsf{S}$. Without loss of generality, see Appendix E, we assume that $r = i_{K-1}$ and choose $\varphi_r = \mathrm{d}\mu_{i_{K-1}}/\mathrm{dLeb}$, such that $\pi_{i_{K-1}}^0 = \mu_{i_{K-1}}$.

**Dynamic approach to mIPF.** In order to approximate solutions of (TreeSB), we consider the *multi-marginal* extension of the IPF algorithm, denoted by mIPF. Namely, we define a sequence of probability distributions $(\pi^n)_{n \in \mathbb{N}}$ such that for any $n \in \mathbb{N}$

$$\pi^{n+1} = \mathrm{argmin}\{\mathrm{KL}(\pi|\pi^n) \, : \, \pi \in \mathscr{P}^{(|\mathsf{V}|)}, \, \pi_{i_{k_n+1}} = \mu_{i_{k_n+1}}\} , \tag{mIPF}$$

where $k_n = (n-1) \bmod (K)$ and $(k_n + 1)$ is identified with $n \bmod (K)$. We define a *mIPF cycle* as a sequence of $K$ consecutive mIPF updates. In particular, each marginal constraint is considered exactly once during one mIPF cycle. In a practical setting, our main aim is to sample from the (mIPF) iterates at the lowest cost. Although these updates can be made explicit, see Marino & Gerolin (2020) for instance, direct sampling is unfeasible in practice when $d$ is large. To overcome this limitation, we suggest to compute these iterates in a *dynamic* fashion with equivalent path measures.

Since $\pi^0$ factorizes along $\mathsf{T}$, see (2), one can show that the iterates of (mIPF) also factorize along $\mathsf{T}$, see Section 4. Since these iterates all have a constrained marginal, we obtain the following decomposition for any $n \in \mathbb{N}$: $\pi^n = \mu_{i_{k_n}} \otimes_{(v,v') \in \mathsf{E}_{k_n}} \pi_{v'|v}^n$ where $\mathsf{E}_{k_n}$ denotes the set of edges of the directed tree $\mathsf{T}_{k_n}$. Then, our approach consists in computing *dynamic* iterates, *i.e.*, path measures, along the edges of $\mathsf{T}$ that coincide on their extremal times with the *static* iterates $(\pi^n)_{n \in \mathbb{N}}$. Namely, for any $n \in \mathbb{N}$, for any edge $(v, v') \in \mathsf{E}_{k_n}$, we define a path measure $\mathbb{P}_{(v,v')}^n \in \mathscr{P}(\mathrm{C}([0, T_{v,v'}], \mathbb{R}^d))$ such that $\mathrm{Ext}(\mathbb{P}_{(v,v')}^n) = \pi_{v,v'}^n$, where $\mathrm{Ext}(\mathbb{P}_{(v,v')}^n)$ stands for the joint distribution of $\mathbb{P}_{(v,v')}^n$ at times $0$ and $T_{v,v'}$. In particular, it comes that $\pi_{v'|v}^n = \mathbb{P}_{(v,v'),T_{v,v'}|0}^n$. Using the tree-based form of the (mIPF) iterates, we can thus sample from $\pi^n$ by (i) following the directed edges of $\mathsf{T}_{k_n}$, (ii) diffusing along them the corresponding path measures $(\mathbb{P}_{(v,v')}^n)_{(v,v') \in \mathsf{E}_{k_n}}$ and (iii) picking the samples on the vertices. When $\mathsf{T}$ is a *bridge-shaped* tree (2 vertices, 1 edge), it simply reduces to the dynamic reformulation of the IPF scheme. In what follows, we explain how to obtain our *dynamic* sequence.

**Definition of the dynamic iterates.** We first compute the iterate $\mathbb{P}^0$, corresponding to the dynamic version of $\pi^0$ defined (2), in Proposition 1. Then, we build the following iterates by recursion on $n \in \mathbb{N}$ and prove their well-posedness in Proposition 2.

**Proposition 1.** *Let* $\mathsf{T}_{K-1} = (\mathsf{V}, \mathsf{E}_{K-1})$, *the directed tree associated with* $\mathsf{T} = (\mathsf{V}, \mathsf{E})$ *and root* $i_{K-1}$. *Then, for any* $(v, v') \in \mathsf{E}_{K-1}$, *there exists* $\mathbb{P}^0_{(v,v')} \in \mathscr{P}(\mathrm{C}([0, T_{v,v'}], \mathbb{R}^d))$ *with* $\mathrm{Ext}(\mathbb{P}^0_{(v,v')}) = \pi^0_{(v,v')}$ *and such that* $\mathbb{P}^0_{(v,v')|0}$ *is the distribution of* $(\mathbf{B}_t)_{t \in [0, T_{v,v'}]}$, *recalling that* $T_{v,v'} = \varepsilon/(2w_{v,v'})$.

Before deriving the dynamic counterpart of the (mIPF) iterates, we introduce several definitions. For any path measure $\mathbb{P}$, we denote by $\mathbb{P}^R$ the *time-reversal* of $\mathbb{P}$. For any directed tree and any vertex $v$ of this tree, $p(v)$ refers to the (unique) *parent* of $v$, and $c(v)$ to the unique *child* of $v$ when it exists, see Appendix B for more details.

Let $n \in \mathbb{N}$. Assume that we have defined the sequence of our dynamic iterates $(\mathbb{P}^m_{(v,v')})_{(v,v') \in \mathsf{E}_{k_m}, m \leq n}$ up to stage $n$.

Consider the path $\mathsf{P}_n = \{(v_j, v_{j+1})\}_{j=1}^J$ in the directed tree $\mathsf{T}_{k_n}$ such that $v_1 = i_{k_n}$ and $v_{J+1} = i_{k_n+1}$. In particular, for any $(v, v') \in \mathsf{E}_{k_n+1}$, either $(v', v) \in \mathsf{P}_n$ or $(v, v') \in \mathsf{E}_{k_n} \backslash \mathsf{P}_n$. This is illustrated in Figure 1 when $\mathsf{V} = \{0, 1, 2, 3, 4\}$, $S = \{2, 3, 4\}$, $i_k = 3$ and $i_{k+1} = 4$: in this case, $\mathsf{P} = \{(3, 1), (1, 0), (0, 4)\}$ and $(1, 2)$ is the only edge common to $\mathsf{E}_k$ and $\mathsf{E}_{k+1}$.

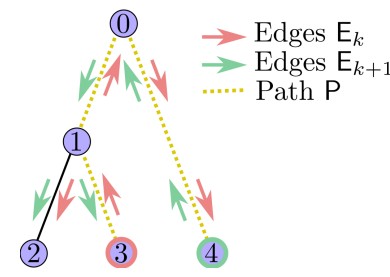

Figure 1: Illustration of the change of root in a toy tree with 5 vertices.

Consider now the directed tree $\mathsf{T}_{k_n+1}$. We define the $(n+1)$-th iterate of our dynamic sequence by recursion on the edges of this tree, following the breadth-first order. In this order, $(i_{k_n+1}, c(i_{k_n+1})) = (v_{J+1}, v_J)$ is the first edge considered.

First, we define $\mathbb{P}^{n+1}_{(v_{J+1}, v_J)} = \mu_{i_{k_n+1}} \otimes (\mathbb{P}^n_{(v_J, v_{J+1})|0})^R$. In the case of a *bridge-shaped* tree, this is exactly the $(n+1)$-th update described in DSB. Then, for any $(v, v') \in \mathsf{E}_{k_n+1} \backslash \{(v_{J+1}, v_J)\}$,

(a) either $(v, v') \in \mathsf{E}_{k_n} \backslash \mathsf{P}_n$, and we define $\mathbb{P}^{n+1}_{(v,v')} = \mathbb{P}^{n+1}_{(p(v),v), T_{p(v),v}} \otimes \mathbb{P}^n_{(v,v')|0}$,

(b) or $(v', v) \in \mathsf{P}_n$, and we define $\mathbb{P}^{n+1}_{(v,v')} = \mathbb{P}^{n+1}_{(p(v),v), T_{p(v),v}} \otimes (\mathbb{P}^n_{(v',v)})^R_{|0}$.

**Proposition 2.** *Consider the sequence of dynamic iterates defined by (a) and (b). Then, for any* $n \in \mathbb{N}$ *and any* $(v, v') \in \mathsf{E}_{k_n}$, $\mathbb{P}^n_{(v,v')} \in \mathscr{P}(\mathrm{C}([0, T_{v,v'}], \mathbb{R}^d))$ *and we have* $\mathrm{Ext}(\mathbb{P}^n_{(v,v')}) = \pi^n_{(v,v')}$.

Proposition 2 highlights the equivalence between the (mIPF) iterates and our dynamic iterates. These path measures are defined iteratively, by following the updates (a) and (b) along the edges of $\mathsf{T}$. The key observation here is that the computation of each dynamic iterate reduces to a sequence of updates (b) on a *path* linking two leaves of $\mathsf{T}$. We emphasize that our iterates could be similarly obtained by directly considering a dynamic formulation of (TreeSB) and introducing the formalism of deterministic time branching processes. We leave the study of this problem for future work. We now get into the details of our practical implementation, which relies on score-based methods.

**Approximation of the dynamic iterates.** The time-reversal operated in the update (b) can be computed explicitly, see Haussmann & Pardoux (1986) for instance. Indeed, assuming that $\mathbb{P}^n_{(v',v)}$ is associated with $\mathrm{d}\mathbf{X}_t = f_{t,v',v}(\mathbf{X}_t)\mathrm{d}t + \mathrm{d}\mathbf{B}_t$ with $\mathbf{X}_0 \sim \pi^n_{v'}$, then, under mild conditions, its time-reversal $(\mathbb{P}^n_{(v',v)})^R$ is associated with $\mathrm{d}\mathbf{Y}_t = \{-f_{T-t,v',v} + \nabla \log p_{v',v,T-t}\}(\mathbf{Y}_t)\mathrm{d}t + \mathrm{d}\mathbf{B}_t$ with $\mathbf{Y}_0 \sim \pi^{n+1}_v$, where $p_{v',v,t}$ is the density of $\mathbb{P}^n_{(v',v),t}$ w.r.t. the Lebesgue measure. The score $\nabla \log p_{v',v,T-t}$ can then be approximated using score-matching techniques (Hyvärinen, 2005; Vincent, 2011) which are now ubiquitous in diffusion models (Song et al., 2021) and used in DSB De Bortoli et al. (2021). Therefore, at iteration $(n+1)$, the update (b) is similar to the one of DSB *for each edge* on the path joining $i_{k_n}$ and $i_{k_n+1}$. In practice, we parameterize the drifts $f_{t,v,v'}$ for any $\{v, v'\} \in \mathsf{E}$ with neural networks $f_{t,\theta_{v,v'}}$ and use the *mean-matching* loss introduced by De Bortoli et al. (2021). Note that doing so, we obtain $2|\mathsf{E}|$ neural networks. The whole procedure consisting in computing our dynamic iterates using the DSB framework is called *Tree-based Diffusion Schrödinger Bridge* (TreeDSB) and is summarized in Algorithm 1.

---
**Algorithm 1** TreeDSB (Training)
---
1: **Input:** $\mathsf{T} = (\mathsf{V}, \mathsf{E})$, $\{\mu_i\}_{i \in \mathsf{S}}$, $\{\theta_{v,v'}\}_{\{v,v'\} \in \mathsf{E}}$, $N \in \mathbb{N}$
2: **for** $n = 0, \ldots, N$ **do**
3:    Let $k_n = (n-1) \bmod(K)$
4:    Get path between $i_{k_n}$ and $i_{k_n+1}$, $\mathsf{P}_n = \{v_j, v_{j+1}\}_{j=1}^J$
5:    **while** not converged **do**
6:      **for** $j = 1, \ldots, J$ **do**
7:        Sample from $\mathbb{P}_{v_j, v_{j+1}}^n$ (Euler-Maruyama)
8:        Compute *mean matching* loss $\ell(\theta_{v_{j+1}, v_j})$
9:        $\theta_{v_{j+1}, v_j} \leftarrow$ Gradient Step($\ell(\theta_{v_{j+1}, v_j})$)
10:       Update $f_{t, \theta_{v_{j+1}, v_j}}$
11:      **end for**
12:    **end while**
13: **end for**
14: **Output:** $\{\theta_{v,v'}\}_{\{v,v'\} \in \mathsf{E}}$
---

The algorithm is initialized with $f_{t, \theta_{v,v'}} = 0$ for all $\{v, v'\} \in \mathsf{E}$. This corresponds to Brownian motion dynamics when sampling at the first iteration of TreeDSB, see Proposition 1. Note that in Algorithm 1, when we sample from $\mathbb{P}_{(v_j, v_{j+1})}^n$, we update $f_{t, \theta_{v_{j+1}, v_j}}$ which will be used to sample from $\mathbb{P}_{(v_{j+1}, v_j)}^{n+1}$ in the next iterations. In order to sample from the dynamics $\mathbb{P}_{(v_j, v_{j+1})}^n$, we consider its Euler–Maruyama discretization, see Appendix F for more details. We describe the different steps of the algorithm in the case of a toy example below, see Figure 2 for an illustration.

**TreeDSB on a toy tree.** We consider a star-shaped tree with three leaves denoted $\{1, 2, 3\}$ and its central node $\{0\}$. Following (2), we define $\pi^0$ with $r = 3$ and $\varphi_r = (\mathrm{d}\mu_3/\mathrm{dLeb})$. During the first iteration of TreeDSB, $\mathsf{T}$ is rooted at vertex 3 and we compute samples from the *forward* path $\mathsf{P}_0 = \{(3, 0), (0, 1)\}$ with Brownian motions, see Proposition 1, in order to learn the *backward* path $\{(1, 0), (0, 3)\}$. In the next iteration, we re-root the tree $\mathsf{T}$ at vertex 1 and consider the *forward* path $\mathsf{P}_1 = \{(1, 0), (0, 2)\}$, where the edges $(1, 0)$ and $(0, 2)$ are respectively given by the first iteration and the initialisation. This highlights that *TreeDSB does not require to update the whole tree*. The following iterations are done similarly. At each iteration $n \in \mathbb{N}$, we sample from $\pi^n$ by first sampling from $\mu_{k_n}$ at leaf $i_{k_n}$ and then following the parameterized SDEs on the directed edges of $\mathsf{T}_{k_n}$.

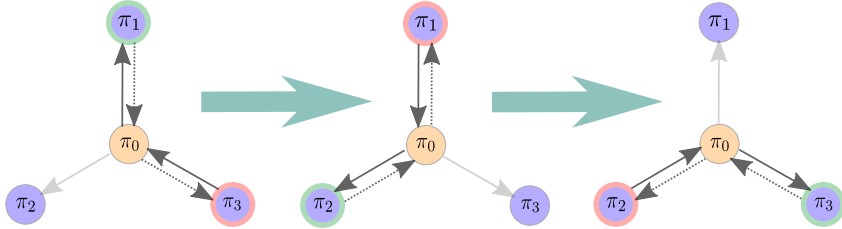

Figure 2: Illustration of one mIPF cycle solved by TreeDSB for a toy star-shaped tree. At each iteration, our method learns the *backward* stochastic process (dotted arrows) that goes from the target leaf (green-circled), corresponding to the constrained marginal, to the current root of the tree (red-circled) by using samples from the *forward* stochastic process (solid arrows).

## 4 Theoretical properties of mIPF

In this section, we study some of the theoretical properties of the *static* iterates $(\pi^n)_{n \in \mathbb{N}}$, that are equivalent to our *dynamic* iterates according to Proposition 2. In the case where the cost function $c$ is bounded in (EmOT), results of convergence of (mIPF) exist (Marino & Gerolin, 2020; Carlier, 2022). However, our setting does not satisfy their assumptions, since our transport cost is quadratic and the measures are defined on $\mathbb{R}^d$. In what follows, we provide the first non-quantitative convergence results for (mIPF) in a *non-compact* setting.

For the rest of the section, we consider a static formulation of the multi-marginal Schrödinger bridge problem which is more general than (TreeSB), defined as

$$\pi^\star = \operatorname{argmin}\{\mathrm{KL}(\pi|\pi^0) : \pi \in \mathscr{P}^{(\ell+1)}, \pi_i = \mu_i, \forall i \in \mathsf{S}\}, \qquad \text{(static-mSB)}$$

where $\mathsf{S} \subset \{0, \ldots, \ell\}$, $\pi^0 \in \mathscr{P}$, $\{\mu_i\}_{i \in \mathsf{S}} \in \mathscr{P}^{|\mathsf{S}|}$. We consider the following set of assumptions.

**A1.** *There exists a family of measures* $\{\nu_i\}_{i \in \{0, \ldots, \ell\}}$ *defined on* $(\mathbb{R}^d, \mathcal{B}(\mathbb{R}^d))$ *such that* $\pi^0 \ll \bigotimes_{i=0}^\ell \nu_i$ *with density* $h = \mathrm{d}\pi^0/(\mathrm{d}\bigotimes_{i=0}^\ell \nu_i)$ *and* $\mu_i \ll \nu_i$ *with density* $r_i = \mathrm{d}\mu_i/\mathrm{d}\nu_i$ *for any* $i \in \mathsf{S}$.

**A2.** $\{\pi \in \mathscr{P}^{(\ell+1)} : \mathrm{KL}(\pi \mid \pi^0) < \infty, \pi_i = \mu_i, \forall i \in \mathsf{S}\} \neq \emptyset$.

**A3.** *There exists a family of probability measures* $\{\tilde{\mu}_j\}_{j \in \{0, \ldots, \ell\} \setminus \mathsf{S}}$ *such that* $\pi^0 \sim \tilde{\pi}^0$, *where* $\tilde{\pi}^0 = \bigotimes_{i \in \mathsf{S}} \mu_i \bigotimes_{j \in \{0, \ldots, \ell\} \setminus \mathsf{S}} \tilde{\mu}_j$.

In particular, (static-mSB) recovers (TreeSB) by considering $\nu_i = \text{Leb}$ for any $i \in \{0, \dots, \ell\}$ and $h(x_{0:\ell}) = \varphi_r(x_r) \exp[-c(x_{0:\ell})/\varepsilon]$ in **A**1. We detail in Appendix D how **A**2 and **A**3 can be met in (TreeSB). Under these assumptions, the multi-marginal Schrödinger Bridge exists.

**Proposition 3.** *Assume* **A**1 *and* **A**2. *Then, there exists a unique solution* $\pi^\star$ *to* (static-mSB). *In addition, assume* **A**3. *Then, there exists a family* $\{\psi_i^\star\}_{i \in \mathsf{S}}$ *of measurable functions* $\psi_i^\star : \mathbb{R}^d \to \mathbb{R}$ *such that*

$$(\mathrm{d}\pi^\star/\mathrm{d}\pi^0) = \exp[\textstyle\bigoplus_{i \in \mathsf{S}} \psi_i^\star] \quad \pi^0\text{-a.s.}$$

In order to establish the existence and uniqueness result of Proposition 3, we extend results from Nutz (2021) to the multi-marginal setting. A consequence of Proposition 3 is that the iterates of (mIPF) can be described using potentials.

**Corollary 4.** *Assume* **A**1, **A**2 *and* **A**3. *Let* $(\pi^n)_{n \in \mathbb{N}}$ *be the sequence given by* (mIPF). *Then, for any* $n \in \mathbb{N}^*$ *with* $k_n = (n-1) \bmod(K)$ *and* $q_n \in \mathbb{N}$ *such that* $n = q_n K + k_n + 1$, *there exists a family of measurable functions* $\{\psi_{i_0}^{q_n+1}, \dots, \psi_{i_{k_n}}^{q_n+1}, \psi_{i_{k_n+1}}^{q_n}, \dots, \psi_{i_{K-1}}^{q_n}\}$ *such that*

$$(\mathrm{d}\pi^n/\mathrm{d}\pi^0)(x_{0:\ell}) = \exp[\textstyle\bigoplus_{j=0}^{k_n} \psi_{i_j}^{q_n+1}(x_{i_j}) \bigoplus_{j=k_n+1}^{K-1} \psi_{i_j}^{q_n}(x_{i_j})] \quad \pi^0\text{-a.s.}$$

In the tree-based setting, Corollary 4 explains why the (mIPF) iterations preserve the tree-based Markovian nature of $\pi^0$. We now prove that the marginal $\pi_i^n$ converges to $\mu_i$ for any $i \in \mathsf{S}$, as $n$ goes to infinity, *i.e.*, we have marginal convergence on the leaves of $\mathsf{T}$.

**Proposition 5.** *Assume* **A**1 *and* **A**2. *Let* $(\pi^n)_{n \in \mathbb{N}}$ *be the sequence given by* (mIPF). *Then, we have* $\lim_{n \to \infty} \|\pi_i^n - \mu_i\|_{\mathrm{TV}} = 0$ *for any* $i \in \mathsf{S}$.

The previous result does not ensure the convergence of $(\pi^n)_{n \in \mathbb{N}}$ to the solution to (static-mSB). In particular, Proposition 5 does not provide the convergence of the marginals on the nodes $v \in \mathsf{V} \backslash \mathsf{S}$, which is key to compute regularized Wasserstein barycenters with TreeDSB. Relying on additional assumptions, we now derive the convergence of (mIPF).

**A**4. $\bigoplus_{i \in \mathsf{S}} \mathrm{L}^1(\mu_i) \subset \mathrm{L}^1(\pi^\star)$ *is closed.*

**A**5. *There exist* $\bar{c} \in (0, \infty)$ *such that* $\exp(\psi_{i_k}^n - \psi_{i_k}^{n+1}) \leq \bar{c}$, *for any* $n \in \mathbb{N}$, *any* $k \in \{0, \dots, K-2\}$.

These assumptions can be seen as multi-marginal extensions of the ones of Ruschendorf (1995), see Appendix D for a discussion and examples.

**Proposition 6.** *Assume* **A**1, **A**2, **A**3, **A**4 *and* **A**5. *Let* $(\pi^n)_{n \in \mathbb{N}}$ *be the sequence given by* (mIPF). *Then, we have* $\lim_{n \to \infty} \|\pi^n - \pi^\star\|_{\mathrm{TV}} = 0$, *where* $\pi^\star$ *is given in Proposition 3.*

To the best of our knowledge, Proposition 6 is the first convergence result of (mIPF) without assuming that the space is compact or that the cost is bounded. We highlight that traditional techniques to prove the convergence of IPF cannot be easily extended to the multi-marginal setting as pointed by Carlier (2022). In the case of bounded cost, quantitative results exist (Marino & Gerolin, 2020; Carlier, 2022). We leave the study of such results in the *unbounded* cost setting for future work.

## 5   Application to Wasserstein barycenters

Although Algorithm 1 can be applied to trees $\mathsf{T}$ with fixed marginals on the leaves, one case of particular interest is star-shaped trees, *i.e.*, trees with a central node, denoted by index 0, and such that $\mathsf{S} = \{1, \dots, \ell\}$ (see Figure 2 for an illustration with $\ell = 3$). In this section, we draw a link between (TreeSB) and regularized Wasserstein barycenters. We recall the definition of the Wasserstein distance of order 2 with $\varepsilon$-entropic regularization between $\mu$ and $\nu$ (Peyré et al., 2019, Chapter 4)

$$W_{2,\varepsilon}^2(\mu, \nu) = \inf\{\textstyle\int \|x_1 - x_0\|^2 \mathrm{d}\pi(x_0, x_1) - \varepsilon \mathrm{H}(\pi) : \pi \in \mathscr{P}^{(2)}, \pi_0 = \mu, \pi_1 = \nu\}. \quad (3)$$

In this work, we consider the $(\ell\varepsilon, (\ell-1)\varepsilon)$-doubly-regularized Wasserstein-2 barycenter problem (Chizat, 2023) defined as follows

$$\mu_\varepsilon^\star = \arg\min\{\textstyle\sum_{i=1}^{\ell} w_i W_{2,\varepsilon/w_i}^2(\mu, \mu_i) + (\ell-1)\varepsilon \mathrm{H}(\mu) : \mu \in \mathscr{P}\}, \quad (\text{regWB})$$

where $(w_i)_{i \in \{1,\dots,\ell\}} \in (0, +\infty)^\ell$. The following proposition shows the equivalence between the barycenter problem (regWB) and the multi-marginal Schrödinger bridge problem (TreeSB) over $\mathsf{T}$. In particular, it allows us to use TreeDSB to estimate the solution $\mu_\varepsilon^\star$ of (regWB).

**Proposition 7.** *Let $\varepsilon > 0$. Assume **A**0. Also assume that* T *is a star-shaped tree with central node indexed by* 0*, and that the reference measure of* (TreeSB) *defined in* (2) *verifies* $r = i_{K-1}$ *and* $\varphi_r = \mathrm{d}\mu_{i_{K-1}}/\mathrm{dLeb} > 0$. *Under* **A**2*,* (regWB) *has a unique solution* $\pi_0^\star$*, where* $\pi^\star$ *solves* (TreeSB).

The proof of this result is postponed to Appendix D. More generally, we show in Appendix D that, for any tree T, (TreeSB) is equivalent to a regularized version of the Wasserstein propagation problem (Solomon et al., 2014, 2015). Moreover, we present in Appendix E an extension of Proposition 7 in the case where the chosen root $r$ is not a leaf of T. We finally emphasize that the formulation of (regWB) leads to a *minimization* of the entropy of the barycenter. In particular, this allows us to choose $\varepsilon$ reasonably large in TreeDSB, which is a stability advantage compared to other regularized methods which do not consider this further regularization.

## 6   Related work

**Diffusion Schrödinger Bridge.**   Schrödinger Bridges (Schrödinger, 1932) have been extensively studied using tools from stochastic control and probability theory (Léonard, 2014; Dai Pra, 1991; Chen et al., 2021). More recently, algorithms were proposed to efficiently approximate such bridges in the context of machine learning. In particular, De Bortoli et al. (2021) proposed DSB while Vargas et al. (2021); Chen et al. (2022) developed related algorithms. In Chen et al. (2023), the authors study a multi-marginal version of DSB in a linear tree-based setting, where the set of observed nodes is the whole set of vertices. However, contrary to our setting, Chen et al. (2023) introduced a momentum variable. This allows for smoother trajectories which are desirable for single-cell trajectories applications and correspond to some spline interpolation in the space of probability measures (Chen et al., 2018). A general framework for tree-based static Schrödinger Bridges on discrete state-spaces was given in Haasler et al. (2021). In this work, we extend their formulation to a dynamic and continuous setting, see Appendix D for more a thorough comparison.

**Wasserstein barycenters.**   The notion of Wasserstein barycenter was first introduced in Rabin et al. (2012) and then later studied in Agueh & Carlier (2011). The algorithms to solve this problem can be split into two families: the in-sample based approaches and the parametric ones. In-sample approaches require access to all the measures $\mu_i$ which are assumed to be empirical measures (Cuturi & Doucet, 2014; Benamou et al., 2015; Solomon et al., 2015). Related to this class of algorithms is the semi-discrete approach, which aims at computing a Wasserstein barycenter between continuous distribution but rely on a discretization of the barycenter (Claici et al., 2018; Staib et al., 2017; Mi et al., 2020). Most recent approaches do not rely on a discrete representation of the barycenter, but instead parameterize it using neural networks. These approaches can be further split into two categories. First, *measure-based optimization* approaches parameterize the measures using a neural network. This is the case of Cohen et al. (2020), where the barycenter is given by a generative model, which is then optimized . Fan et al. (2020) introduce an optimization procedure which relies on a *min-max-min* problem using the framework of Makkuva et al. (2020). More recently, Korotin et al. (2022) considered a fixed point-based algorithm introduced in Álvarez-Esteban et al. (2016) to update a generative model parametrizing the barycenter. On the one hand, *potential-based methods* rely on a dual formulation of the barycenter. Korotin et al. (2021) parameterized the dual potentials using Input Convex Neural Network and considered regularizing losses imposing conjugacy and congruency. On the other hand, Li et al. (2020) consider a dual version of the *regularized* Wasserstein barycenter problem contrary to other works. Our approach applied to start-shaped trees also approximates a *regularized* Wasserstein barycenter. However, contrary to Li et al. (2020), we do not consider a parameterization of the potentials in the *static* setting but instead, parameterize the drift of an associated *dynamic* formulation using Schrödinger bridges. To the best of our knowledge TreeDSB is the first approach leveraging DSB-like algorithms to compute Wasserstein barycenters.

## 7   Experiments

In our experiments[3], we illustrate the performance of TreeDSB to compute entropic regularized Wasserstein barycenters for various tasks . We choose to compare our method with state-of-the-art regularized algorithms: fast free-support Wasserstein barycenter (fsWB) (Cuturi & Doucet, 2014) , and continuous regularized Wasserstein barycenter (crWB) (Li et al., 2020). In all of our settings, we consider a star-shaped tree with $K$ leaves and edge weights that are equal to $1/K$, resulting in a

---

[3]Code available at `https://github.com/maxencenoble/tree-diffusion-schrodinger-bridge`.

sequential training procedure over $2K$ neural networks. The initial diffusion is always a Brownian motion parameterized as explained in Proposition 1. Hence, the time horizon on each edge is defined by $T = K\varepsilon/2$. The order of the leaves is randomly shuffled between the mIPF cycles. We consider 50 steps for the time discretization on $[0, T]$. We refer to Appendix G for details on the choice of the schedule, the architecture of the neural networks and the settings of our experiments.

**Synthetic two dimensional datasets.** We first illustrate TreeDSB in a synthetic two dimensional setting. We consider three different datasets *Swiss-roll* (vertex 0, starting node $r$), *Circle* (vertex 2) and *Moons* (vertex 3) and compute their Wasserstein barycenter (vertex 1) by running TreeDSB for 50 mIPF cycles with $\varepsilon = 0.1$. In Figure 3, we show the estimated densities of the datasets on the leaves of the tree (we emphasize that the distributions plotted on each leaf are generated from the central barycenter measure). In Figure 4, we observe the consistency between the barycenters generated from the different leaves. In Appendix G, we present additional results for this setting.

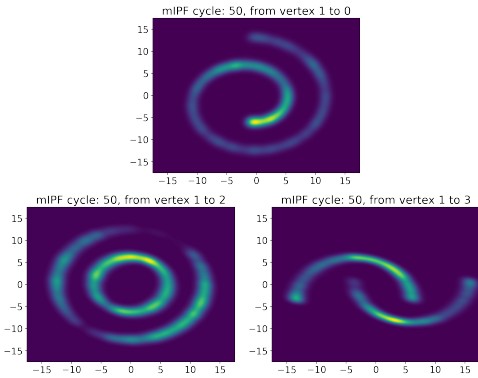

Figure 3: Estimated densities on the leaves.

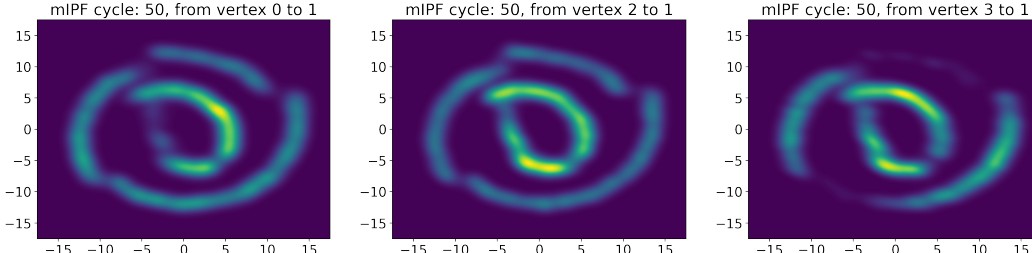

Figure 4: From left to right: barycenter estimated from the leaves *Swiss-roll*, *Circle* and *Moons*.

**Synthetic Gaussian datasets.** Next, we consider three independent Gaussian distributions with zero mean and random non-diagonal covariance matrices whose conditional number is less than 10, following Fan et al. (2020). In this case, the non-regularized barycenter can be exactly computed. To evaluate the performance of the algorithms, we use the Bures-Wasserstein Unexplained Variance Percentage (UVP), following (Korotin et al., 2021, Section 5). Given a target distribution $\mu^\star \in \mathscr{P}$ and some approximation $\mu \in \mathscr{P}$, we define

$$\mathrm{BW}_2^2\text{-UVP}(\mu, \mu^\star) = 100 \cdot 2\,\mathrm{BW}_2^2(\mu, \mu^\star)/\mathrm{Var}(\mu^\star)\%\,,$$

where $\mathrm{BW}_2^2(\mu, \mu^\star) = W_2^2(\mathrm{N}(\mathbb{E}[\mu], \mathrm{Cov}(\mu)), \mathrm{N}(\mathbb{E}[\mu^\star], \mathrm{Cov}(\mu^\star)))$.

| Method | $d = 2$ | $d = 16$ | $d = 64$ | $d = 128$ | $d = 256$ |
|---|---|---|---|---|---|
| fsWB (Cuturi & Doucet, 2014) | $0.06_{\pm 0.01}$ | $2.86_{\pm 0.06}$ | $11.12_{\pm 0.06}$ | $14.47_{\pm 0.07}$ | $17.41_{\pm 0.05}$ |
| crWB (Li et al., 2020) | $\mathbf{0.02_{\pm 0.01}}$ | $\mathbf{1.52_{\pm 0.11}}$ | $11.41_{\pm 0.73}$ | $5.75_{\pm 0.02}$ | $18.27_{\pm 0.54}$ |
| Tree DSB | $0.63_{\pm 0.26}$ | $1.07_{\pm 0.58}$ | $\mathbf{1.39_{\pm 0.07}}$ | $\mathbf{1.92_{\pm 0.02}}$ | $\mathbf{2.62_{\pm 0.07}}$ |

Table 1: Gaussian setting: comparison with the regularized methods crWB and fsWB.

In this setting, we choose $\mu^\star$ to be the non-regularized barycenter and assess the dependency w.r.t. the dimension of the algorithms using the $\mathrm{BW}_2^2$-UVP metric. In Table 1, we compare ourselves with the two regularized methods Li et al. (2020) ($L_2$-reg. equal to $10^{-4}$) and Cuturi & Doucet (2014). We run TreeDSB for 10 mIPF cycles with $\varepsilon = 0.1$. Bold numbers represent the best values up to statistical significance. While Li et al. (2020) and Cuturi & Doucet (2014) enjoy better performance in low dimensions ($d = 2$), TreeDSB outperforms these methods as the dimension increases.

**MNIST Wasserstein barycenter.** We then turn to an image experiment using MNIST dataset (LeCun, 1998). Here, an image is not considered as a 2D-dimensional distribution as in Cuturi & Doucet (2014) and Li et al. (2020), but as a sample from a high-dimensional probability measure ($d = 784$). We aim at computing a Wasserstein barycenter between the digits 2,4 and 6. To do so, we

run TreeDSB for 10 mIPF cycles with $r$ that corresponds to the digit 6 and $\varepsilon = 0.5$. In Figure 5, we display samples from the estimated marginals on the leaves, to assess the reconstruction of the digits 2, 4 and 6, and samples from the barycenter, obtained by diffusing from the leaf corresponding to the digit 6. Our results prove the scalability of TreeDSB to the high-dimensional setting, compared to state-of-the-art regularized methods. Additional results on MNIST dataset are given in Appendix G.

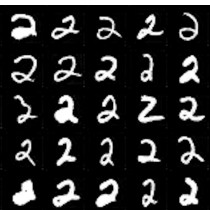  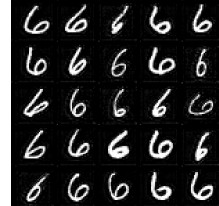 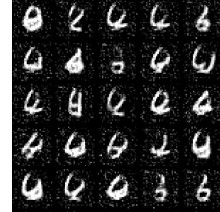

Figure 5: Samples from the estimated MNIST 2-4-6 marginals and from their Wasserstein barycenter.

**Subset posterior aggregation.** Finally, we evaluate TreeDSB in the context of *Bayesian fusion* (Srivastava et al., 2018), also called posterior aggregation. Given a Bayesian model and a dataset partitioned into several shards, this task aims at recovering the full data posterior distribution from the posterior distributions computed on each shard.

| Method | Without het. | With het. |
|---|---|---|
| fsWB (Cuturi & Doucet, 2014) | $12.95_{\pm 0.35}$ | $14.43_{\pm 0.51}$ |
| crWB (Li et al., 2020) | $20.66_{\pm 0.71}$ | $23.06_{\pm 0.12}$ |
| Tree DSB | $\mathbf{8.69_{\pm 0.12}}$ | $\mathbf{8.90_{\pm 0.68}}$ |

Table 2: Bayesian fusion setting: comparison with the regularized methods crWB and fsWB.

In particular, it has been proved that the barycenter of the subdataset posteriors is close to the full data posterior under mild assumptions (Srivastava et al., 2018). Here, we consider a logistic regression model applied to the `wine` dataset[4] ($d = 42$) and proceed as follows. We first split this dataset into 3 subsets, with or without heterogeneity, and estimate the posterior parameters on each shard. Then, we draw samples from the obtained logistic distributions to define $\mu_1, \mu_2, \mu_3$. Then, we compute the Wasserstein barycenter of these measures, and compare it to the posterior computed on the full dataset. As in the synthetic Gaussian experiment, we run TreeDSB for 10 mIPF cycles $\varepsilon = 0.1$ and we compare ourselves with Li et al. (2020) ($L_2$-reg. equal to $10^{-4}$) and Cuturi & Doucet (2014). We evaluate the methods using the $\mathrm{BW}_2^2$-UVP metric, where $\mu^\star$ is the estimated full data posterior, and report the results in Table 2. In both settings, we observe that our method outperforms existing regularized methods to compute Wasserstein barycenters.

**Limitations.** One of the main limitation of entropic regularized OT approach is that their behavior is usually badly conditioned as $\varepsilon \to 0$. In our setting, we observe that if $\varepsilon$, or equivalently $T$, is too low then the algorithm becomes less stable as the training of the models slows down. In the future, we plan to mitigate this issue by incorporating fixed point techniques like the one used in Korotin et al. (2022). Finally, since our algorithm is based on DSB (De Bortoli et al., 2021), it suffers from the same limitations. In particular, training different neural networks iteratively incurs some bias in the SDE which is harmful for large number of mIPF iterations.

## 8 Discussion

In this paper, we introduced Tree-based Diffusion Schrödinger Bridge (TreeDSB) a scalable scheme to approximate solutions of entropic-regularized multi-marginal Optimal Transport (mOT) problems. Our methodology leverages tools from the diffusion model literature and extends Diffusion Schrödinger Bridge (De Bortoli et al., 2021). In particular, it approximates the iterates of the multi-marginal Iterative Proportional Fitting (mIPF) algorithm, for which we prove its convergence under mild assumptions. We illustrate the efficiency of TreeDSB for image processing and Bayesian fusion, using the link between mOT and Wasserstein barycenters. In future work, we would like to study quantitative convergence bounds for mIPF in the *unbounded* cost setting. Another line of work would be to scale TreeDSB to higher dimensional problems building on recent developments in the diffusion model and flow matching community (Lipman et al., 2023; Peluchetti, 2023; Shi et al., 2023).

---

[4] https://archive.ics.uci.edu/ml/datasets/wine

## Acknowledgments

We thank James Thornton for the DSB codebase[5] and useful discussions. AD acknowledges support from the Lagrange Mathematics and Computing Research Center. AD and MN would like to thank the Isaac Newton Institute for Mathematical Sciences for support and hospitality during the programme *The mathematical and statistical foundation of future data-driven engineering* when work on this paper was undertaken. MN acknowledges funding from the grant SCAI (ANR-19-CHIA-0002).

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

## Appendix organization

First, additional notation is introduced in Appendix A. Then, we briefly recall some notions on undirected and directed trees in Appendix B. Similarly, martingale problems are introduced in Appendix C. The proofs of the main manuscript and additional theoretical results on Tree Schrödinger Bridges are given in Appendix D. Additional details on our consideration of the tree-based static SB problem are described in Appendix E. Details on the implementation of TreeDSB are given in Appendix F and the experiments are investigated in Appendix G.

## A Additional notation

For any finite set $\mathsf{E}$, we equivalently refer to the cardinal of $\mathsf{E}$ as $\mathrm{card}(\mathsf{E})$ or $|\mathsf{E}|$. Let $(\mathsf{X}, \mathcal{X})$ be a measurable space. For any $x \in (\mathbb{R}^d)^{\ell+1}$ and any $m \in \{0, \ldots, \ell\}$, let $x_{-m} = (x_0, \ldots, x_{m-1}, x_{m+1}, \ldots, x_\ell)$. For any family of measures $\{\nu_j\}_{j \in \{0, \ldots, \ell\}}$ defined on $(\mathsf{X}, \mathcal{X})$ and any $i \in \{0, \ldots, \ell\}$, let $\nu_{-i} = \bigotimes_{j \in \{0, \ldots, \ell\} \setminus \{i\}} \nu_j$. Let $I = \{i_1, \ldots, i_q\} \subset \{1, \ldots, \ell\}$ and $\mu \in \mathscr{P}^{(\ell)}$ such that $\mu \ll \mathrm{Leb}$. We define $I^c = \{1, \ldots, \ell\} \setminus I$ and denote it by $\{i_1^c, \ldots, i_{\bar{q}}^c\}$ where $\bar{q} = \ell - q$. We denote the marginal of $\mu$ along $I$ by $\mu_I$, *i.e.*, $\mu_I \in \mathscr{P}^{(q)}$ and we have for any $\mathsf{A} \in \mathcal{B}((\mathbb{R}^d)^q)$, $\mu_I(\mathsf{A}) = \int_{\mathsf{X}} \mu(x) \prod_{j=1}^q \delta_{x_{i_j}}(\mathsf{A}_j)\mathrm{d}x$. In addition, note that $\mu_I \ll \mathrm{Leb}$. We denote the conditional distribution of $\mu$ given $I$ by $\mu_{|I}(\cdot|\cdot)$, *i.e.*, $\mu_{|I}(\cdot|\cdot) \in \mathscr{P}^{(\bar{q})} \times (\mathbb{R}^d)^q$ and we have for any $y \in (\mathbb{R}^d)^q$ and any $\mathsf{A} \in \mathcal{B}((\mathbb{R}^d)^{\bar{q}})$, $\mu_{|I}(\mathsf{A}|y) = \int_{\mathsf{X}} \mu(x)/\mu_I(y) \prod_{j=1}^q \delta(x_{i_j} - y_j) \prod_{j'=1}^{\bar{q}} \delta_{x_{i_{j'}^c}}(\mathsf{A}_{j'})\mathrm{d}x$. Remark that for any $y \in (\mathbb{R}^d)^q$, $\mu_{|I}(\cdot|y) \ll \mathrm{Leb}$. For any subset $\mathsf{J} \subset I^c$ with $\mathrm{card}(\mathsf{J}) = q_\mathsf{J}$, we also define $\mu_{\mathsf{J}|I}(\cdot|\cdot) \in \mathscr{P}^{(q_\mathsf{J})} \times (\mathbb{R}^d)^q$ such that for any $y \in (\mathbb{R}^d)^q$, $\mu_{\mathsf{J}|I}(\cdot|y) = \{\mu_{|I}(\cdot|y)\}_\mathsf{J}$. For a collection of functions $\{f_i\}_{i \in \mathsf{I}}$, with $\mathsf{I} \subset \{1, \ldots, n\}$ and $n \in \mathbb{N}$ such that $f_i : \mathbb{R}^d \to \mathbb{R}$, we define $\oplus_{i \in \mathsf{I}} f_i : (\mathbb{R}^d)^n \to \mathbb{R}$ such that for any $x = (x_1, \ldots, x_n) \in (\mathbb{R}^d)^n$, $\oplus_{i \in \mathsf{I}} f(x) = \sum_{i \in \mathsf{I}} f_i(x_i)$.

## B Introduction to trees

**Undirected tree.** An undirected graph $\mathsf{T} = (\mathsf{V}, \mathsf{E})$, with vertices $\mathsf{V}$ and edges $\mathsf{E}$, is said to be an *undirected tree* if it is *acyclic* and *connected* (Valiente, 2002, Definition 1.19.). In particular, we have $\mathrm{card}(\mathsf{E}) = \mathrm{card}(\mathsf{V}) - 1$. The undirected edge between two nodes $v_1$ and $v_2$ is similarly denoted by $\{v_1, v_2\}$ or $\{v_2, v_1\}$. We say that $\mathsf{T}' = (\mathsf{V}', \mathsf{E}')$ is a *sub-tree* of $\mathsf{T}$ if $\mathsf{T}'$ is an undirected tree with vertices $\mathsf{V}' \subset \mathsf{V}$ and edges $\mathsf{E}' \subset \mathsf{E}$. For any vertex $v \in \mathsf{V}$, we define the set of its *neighbours* $\mathsf{N}_v$ as the set of vertices $v' \in \mathsf{V}$ such that $\{v, v'\} \in \mathsf{E}$. The integer $\mathrm{card}(\mathsf{N}_v)$ is referred to as the degree of $v$. The vertices with degree 1 are called *leaves*, and we denote the set of leaves by $\mathsf{V}_\mathsf{L} \subset \mathsf{V}$. The (unique) *path* in $\mathsf{T}$ between two vertices $v$ and $v'$ is the sequence of two-by-two distinct edges $\{\{v_i, v_{i+1}\}\}_{i=1}^n$ (with $n \geq 1$) such that $v_k = v_{k+1}$ for any $k \in \{1, \ldots, n\}$ such that $k = 0 \mod(2)$, $v_1 = v$ and $v_{n+1} = v'$. This path can be seen as a linear sub-tree of $\mathsf{T}$, and we define $n$ as the *length* of this path. We say that $\mathsf{T}$ is *weighted* if there exists a map $w : \mathsf{E} \mapsto \mathbb{R}_+$; in this case, $w(\{v_1, v_2\})$, or equivalently $w(\{v_2, v_1\})$ (also denoted by $w_{v_1, v_2}$ or $w_{v_2, v_1}$) is called the weight of the edge $\{v_1, v_2\}$. The tree $\mathsf{T}$ is said to be *rooted* in $r \in \mathsf{V}$ if $r$ defines a partial ordering $\leq_{\mathsf{T},r} \subset \mathsf{V} \times \mathsf{V}$ such that for any $v_1, v_2 \in \mathsf{V}$, $v_1 \leq_{\mathsf{T},r} v_2$ if the node $v_1$ lies on the unique path between $r$ and $v_2$.

**Directed tree.** Consider a directed graph $\mathsf{T}_r = (\mathsf{V}, \mathsf{E}_r)$ rooted in $r \in \mathsf{V}$. Any directed edge $e \in \mathsf{E}_r$ from $v_1 \in \mathsf{V}$ to $v_2 \in \mathsf{V}$ is denoted by $(v_1, v_2)$. $\mathsf{T}_r$ is a said to be a *directed tree* rooted in $r$ if (i) the underlying undirected graph $\mathsf{T} = (\mathsf{V}, \mathsf{E})$ is an undirected tree rooted in $r$ and (ii) any $(v_1, v_2) \in \mathsf{E}_r$ is directed according to the partial ordering $\leq_{\mathsf{T},r}$, *i.e.*, $\{v_1, v_2\} \in \mathsf{E}$ and $v_1 \leq_{\mathsf{T},r} v_2$. For any vertices $(v, v') \in \mathsf{V} \times \mathsf{V}$ such that $v \leq_{\mathsf{T},r} v'$, the (unique) *path* in $\mathsf{T}_r$ from $v$ to $v'$, denoted by $\mathrm{path}_{\mathsf{T}_r}(v, v')$, is defined as the directed version of the path in $\mathsf{T}$ between $v$ and $v'$ (viewed as a sub-tree of $\mathsf{T}$), which is rooted in $v$. We say that $\mathsf{T}_r$ is *weighted*, if $\mathsf{T}$ is weighted and the edges of $\mathsf{T}_r$ have the same weights as the corresponding undirected edges of $\mathsf{T}$. For any $(v_1, v_2) \in \mathsf{E}_r$, we denote this weight by $w_{v_1, v_2}$. We say that $\mathsf{T}_r$ is the (unique) *directed version* of $\mathsf{T}$ rooted in $r$. It is endowed with a canonical vertex numbering $\zeta : \mathsf{V} \to \{0, \ldots, \mathrm{card}(\mathsf{V}) - 1\}$, corresponding to a depth-first traversal of its nodes, starting from the root $r$ (Valiente, 2002, Definition 3.1.). This numbering is consistent with the partial ordering on $\mathsf{T}$, *i.e.*, if $v_1 \leq_{\mathsf{T},r} v_2$, $\zeta(v_1) \leq \zeta(v_2)$, and satisfies $\zeta(r) = 0$. In the rest of the paper, we will write in an equivalent manner $v$ or $\zeta(v)$.

For any vertices $(v_1, v_2) \in \mathsf{E} \times \mathsf{E}$ such that $v_1 \leq_{\mathsf{T},r} v_2$, $\mathrm{path}_{\mathsf{T}_r}(v_1, v_2)$ corresponds to the ordered set of edges in $\mathsf{E}_r$ which define the ordered path between two vertices $v_1$ and $v_2$. For any vertex $v \in \mathsf{V}$, we define:

(a) the set of its *children* $\mathsf{C}_v$ as the set of vertices $v' \in \mathsf{V}$ such that $(v, v') \in \mathsf{E}_r$. In particular, for any $v \in \mathsf{V}_L$, the set of leaves, one has $\mathsf{C}_v = \emptyset$.

(b) its *parent* as the unique vertex $p(v)$ such that $(p(v), v) \in \mathsf{E}_r$, if $v \neq r$ (the parent of the root is not defined).

Note that $\mathsf{N}_r = \mathsf{C}_r$ and, for any vertex $v \in \mathsf{V}\backslash\{r\}$, $\mathsf{N}_v = \{p(v)\} \cup \mathsf{C}_v$.

**Definition 8** (Tree-structured directed Probabilistic Graphical Model (PGM)). *Consider a directed tree $\mathsf{T}_r = (\mathsf{V}, \mathsf{E}_r)$. The directed PGM induced by $\mathsf{T}_r$ ([Koller & Friedman, 2009](), Definition 3.4.), denoted by $\mathscr{P}_{\mathsf{T}_r}$, is the family of distributions $\pi \in \mathscr{P}^{(|\mathsf{V}|)}$ which have a Markovian factorization along $\mathsf{T}_r$, i.e.,*

$$\mathscr{P}_{\mathsf{T}_r} = \{\pi \in \mathscr{P}^{(|\mathsf{V}|)} : \pi = \pi_r \bigotimes_{(v,v')\in\mathsf{E}_r} \pi_{v'|v}\} \ .$$

**Lemma 9.** *Consider an undirected tree $\mathsf{T} = (\mathsf{V}, \mathsf{E})$. Let $(r, r') \in \mathsf{V} \times \mathsf{V}$. Let $\mathsf{T}'$ be a sub-tree of $\mathsf{T}$ with vertices $\mathsf{V}'$ such that $r' \in \mathsf{V}'$. Denote by $\mathsf{T}'_{r'}$ the directed version of $\mathsf{T}'$ rooted in $r'$. Then, for any $\pi \in \mathscr{P}_{\mathsf{T}_r}$, we have $\pi_{\mathsf{V}'} \in \mathscr{P}_{\mathsf{T}'_{r'}}$.*

*Proof.* Let $(r, r') \in \mathsf{V} \times \mathsf{V}$. We denote by $\mathsf{T}_r = (\mathsf{V}, \mathsf{E}_r)$, respectively $\mathsf{T}_{r'} = (\mathsf{V}, \mathsf{E}_{r'})$, the directed version of $\mathsf{T}$ rooted in $r$, respectively $r'$. We define the paths $\mathsf{P}_{r,r'} = \mathrm{path}_{\mathsf{T}_r}(r, r') \subset \mathsf{E}_r$ and $\mathsf{P}_{r',r} = \mathrm{path}_{\mathsf{T}_{r'}}(r', r) \subset \mathsf{E}_{r'}$. It is easy to see that

(a) $\mathsf{E}_r\backslash\mathsf{P}_{r,r'} = \mathsf{E}_{r'}\backslash\mathsf{P}_{r',r}$,

(b) $\mathsf{P}_{r,r'} = \{(v_2, v_1) : (v_1, v_2) \in \mathsf{P}_{r',r}\}$,

(c) $\mathsf{P}_{r',r} = \{(v_2, v_1) : (v_1, v_2) \in \mathsf{P}_{r,r'}\}$.

Let $\pi \in \mathscr{P}_{\mathsf{T}_r}$. First note that for any $(v_1, v_2) \in \mathsf{E}_r$, we have by Bayes decomposition $\pi_{v_1}\pi_{v_2|v_1} = \pi_{v_2}\pi_{v_1|v_2} = \pi_{v_1,v_2}$. Then it comes

$$
\begin{aligned}
\pi &= \pi_r \bigotimes_{(v_1,v_2)\in\mathsf{E}_r} \pi_{v_2|v_1} \\
&= \pi_r \bigotimes_{(v_1,v_2)\in\mathsf{P}_{r,r'}} \pi_{v_2|v_1} \bigotimes_{(v_1,v_2)\in\mathsf{E}_r\backslash\mathsf{P}_{r,r'}} \pi_{v_2|v_1} \\
&= \pi_r \bigotimes_{(v_2,v_1)\in\mathsf{P}_{r',r}} \pi_{v_2|v_1} \bigotimes_{(v_1,v_2)\in\mathsf{E}_{r'}\backslash\mathsf{P}_{r',r}} \pi_{v_2|v_1} \\
&= \pi_{r'} \bigotimes_{(v_1,v_2)\in\mathsf{P}_{r',r}} \pi_{v_2|v_1} \bigotimes_{(v_1,v_2)\in\mathsf{E}_{r'}\backslash\mathsf{P}_{r',r}} \pi_{v_2|v_1} \\
&= \pi_{r'} \bigotimes_{(v_1,v_2)\in\mathsf{E}_{r'}} \pi_{v_2|v_1} \ ,
\end{aligned}
$$

and therefore, we have $\pi \in \mathscr{P}_{\mathsf{T}_{r'}}$.

Let $\mathsf{T}'$ be a sub-tree of $\mathsf{T}$ with vertices $\mathsf{V}'$ such that $r' \in \mathsf{V}'$. First note that $\mathsf{E}'_{r'} \subset \mathsf{E}_{r'}$. Using the previous computation, we have for any $\mathsf{A} \in \mathcal{B}((\mathbb{R}^d)^{|\mathsf{V}'|})$,

$$
\begin{aligned}
\pi_{\mathsf{V}'}(\mathsf{A}) &= \int_{(\mathbb{R}^d)^{|\mathsf{V}|}} \pi_{r'}(x_{r'}) \bigotimes_{(v_1,v_2)\in\mathsf{E}_{r'}} \pi_{v_2|v_1}(x_{v_2}|x_{v_1}) \prod_{v'\in\mathsf{V}'} \delta_{x_{v'}}(\mathsf{A}_{v'})\mathrm{d}x \\
&= \int_{(\mathbb{R}^d)^{|\mathsf{V}|-|\mathsf{V}'|}} \{\pi_{r'}(\mathsf{A}_{r'}) \bigotimes_{(v_1,v_2)\in\mathsf{E}'_{r'}} \pi_{v_2|v_1}(\mathsf{A}_{v_2}|x_{v_1})\} \bigotimes_{(v_1,v_2)\in\mathsf{E}_{r'}\backslash\mathsf{E}'_{r'}} \pi_{v_2|v_1}(x_{v_2}|x_{v_1})\mathrm{d}x_{\mathsf{V}\backslash\mathsf{V}'} \\
&= \{\pi_{r'} \bigotimes_{(v_1,v_2)\in\mathsf{E}'_{r'}} \pi_{v_2|v_1}\}(\mathsf{A}) \ ,
\end{aligned}
$$

which proves that $\pi_{\mathsf{V}'} \in \mathscr{P}_{\mathsf{T}'_{r'}}$. $\qquad\square$

**Discretized undirected tree.** Let $N \geq 1$. Consider an undirected tree $\mathsf{T} = (\mathsf{V}, \mathsf{E})$ with weights $w$. We say that $\mathsf{T}^{(N)} = (\mathsf{V}^{(N)}, \mathsf{E}^{(N)})$ is a $N$-discretized version of $\mathsf{T}$ if it is an undirected tree with weights $w^{(N)}$ such that

(a) $\mathsf{V}^{(N)} = \mathsf{V} \bigsqcup \cup_{\substack{e\in\mathsf{E}, \\ k\in\{1,\dots,N-1\}}} \{v_e^k\}$,

(b) $\mathsf{E}^{(N)} = \cup_{e \in \mathsf{E}} \cup_{k=0,\ldots,N-1} \left\{ \{v_e^k, v_e^{k+1}\} \right\}$ with the convention that the vertices $v_e^N$ and $v_e^N$ are defined such that $\{v_e^0, v_e^N\} = e$,

(c) $\sum_{e \in \mathrm{path}_\mathsf{T}(v,v')} 1/w_e^{(N)} = 1/w_{v,v'}$, if $\{v, v'\} \in \mathsf{E}$.

Remark that the leaves of $\mathsf{T}^{(N)}$ are exactly the original leaves of $\mathsf{T}$ and that $\mathsf{T}^{(1)} = \mathsf{T}$. The non-uniqueness of $\mathsf{T}^{(N)}$ comes from the freedom of choice on the weights of its edges.

**Discretized directed tree.** Let $N \geq 1$. Consider a directed tree $\mathsf{T}_r = (\mathsf{V}, \mathsf{E}_r)$ rooted in $r \in \mathsf{V}$ with weights $w$. We say that $\mathsf{T}_r^{(N)} = (\mathsf{V}^{(N)}, \mathsf{E}_r^{(N)})$ is a $N$-discretized version of $\mathsf{T}_r$ if it is the directed version of $\mathsf{T}^{(N)}$ rooted in $r$, where $\mathsf{T}^{(N)}$ is a $N$-discretized version of the underlying undirected tree of $\mathsf{T}_r$.

# C  Background on martingale problems

In this section, we introduce the background on Stochastic Differential Equations (SDEs) and weak solutions of SDEs following the framework of (Stroock & Varadhan, 1997, Section 10.1, page 249). We recall that $\mathrm{C}_0^\infty(\mathbb{R}^d)$ is the space of infinitely differentiable real-valued functions which vanish at infinity. In addition, we have that $\mathcal{S}_+^d$ is the space of $d \times d$, symmetric, non-negative matrices.

**Definition 10.** *Let $T > 0$ or $T = +\infty$, $\sigma : [0, T) \times \mathbb{R}^d \to \mathcal{S}_+^d$ and $b : [0, T) \times \mathbb{R}^d \to \mathbb{R}^d$, locally bounded measurable functions. We define the* infinitesimal generator, $\mathcal{A}$, *given for any $f \in \mathrm{C}_0^\infty(\mathbb{R}^d)$, $t \in [0, T)$ and $x \in \mathbb{R}^d$ by*

$$\mathcal{A}_t(f)(x) = \langle b_t(x), \nabla f(x) \rangle + \tfrac{1}{2} \langle \sigma_t(x)\sigma_t(x)^\top, \nabla^2 f(x) \rangle. \tag{4}$$

*We say that a probability measure $\mathbb{P}$ satisfies the martingale problem for $\mathcal{A}$ if for any $t \in [0, T)$ and $f \in \mathrm{C}_0^\infty(\mathbb{R}^d)$, we have that $(f(\mathbf{X}_t) - \int_0^t \mathcal{A}_s(f)(\mathbf{X}_s)\mathrm{d}s)_{s \in [0,t]}$ is a $\mathbb{P}$-martingale.*

In the main document, see Section 2, we say that "a path measure $\mathbb{P}$ is associated with $\mathrm{d}\mathbf{X}_t = b(t, \mathbf{X}_t)\mathrm{d}t + \sigma(t, \mathbf{X}_t)\mathrm{d}\mathbf{B}_t$ with $(\mathbf{B}_t)_{t \geq 0}$ a $d$-dimensional Brownian motion" if $\mathbb{P}$ solves the martingale problem associated with $\mathcal{A}$ given by (4). Unless specified, we always assume that such a path measure exists and is unique. Below, we recall the following theorem, see (Stroock & Varadhan, 1997, Theorem 10.2.2), which gives sufficient conditions for the existence and uniqueness of solutions to the martingale problem.

**Theorem 11.** *Assume that for any $x \in \mathbb{R}^d$ we have*

$$\inf\{\langle \theta, \sigma\sigma^\top(s, x)\theta \rangle \ : \ \theta \in \mathbb{R}^d, \ \|\theta\| = 1, \ s \in [0, T]\} > 0,$$
$$\limsup_{y \to x}\{\|\sigma(s, x) - \sigma(s, y)\| \ : \ s \in [0, T]\} = 0.$$

*In addition, assume that there exists $C > 0$ such that for any $x \in \mathbb{R}^d$*

$$\sup\{\|\sigma\sigma^\top(t, x)\| \ : \ s \in [0, T]\} + \sup\{\langle x, b(t, x) \rangle \ : \ s \in [0, T]\} \leq C(1 + \|x\|^2).$$

*Then, there exists a unique solution to the martingale problem with initialization $x_0 \in \mathbb{R}^d$.*

# D  Theoretical results on Tree Schrödinger Bridges

We respectively provide in Appendix D.1, Appendix D.2 and Appendix D.3 the proofs of the results of the main manuscript presented in Section 3, Section 4 and Section 5. Finally, we make a detailed comparison between our setting and the framework of Haasler et al. (2021) in Appendix D.4. In the rest of this section, we consider an undirected tree $\mathsf{T} = (\mathsf{V}, \mathsf{E})$, where $|\mathsf{V}| = \ell + 1$, and some subset $\mathsf{S} \subset \mathsf{V}$ which we denote by $\mathsf{S} = \{i_0, \ldots, i_{K-1}\}$. We define $\mathsf{S}^c = \mathsf{V} \backslash \mathsf{S}$.

## D.1  Proofs of Section 3

Proposition 1 is straightforward to obtain by combining the definition of the Brownian motion with the definition of $\pi^0$ given in (2). The following lemma details the recursion relation between the (mIPF) iterates, which is key to prove Proposition 2.

**Lemma 12.** *Let $(\pi^n)_{n\in\mathbb{N}}$ be the sequence given by* (mIPF). *Let $n \in \mathbb{N}$, $k_n = (n-1)\bmod(K)$, $k_n + 1 = n\bmod(K)$. Denote by $\mathsf{T}_{k_n}$, respectively $\mathsf{T}_{k_n+1}$ with edges $\mathsf{E}_{k_n+1}$, the directed version of $\mathsf{T}$ rooted in $i_{k_n}$, respectively in $i_{k_n+1}$. We have:*

(i) $\pi^n \in \mathscr{P}_{\mathsf{T}_{k_n}}$,

(ii) $\pi^{n+1} = \mu_{i_{k_n+1}} \bigotimes_{(v,v')\in\mathsf{E}_{k_n+1}} \pi^n_{v'|v}$. *In particular, for any $(v,v') \in \mathsf{E}_{k_n+1}$, $\pi^{n+1}_{v'|v} = \pi^n_{v'|v}$.*

*Proof.* We show the result (i) by recursion on $n \in \mathbb{N}$, and will deduce (ii) from the proof. Using (2), we first have $\pi^0 \in \mathscr{P}_{\mathsf{T}_r}$, where $r$ is chosen as $i_{K-1}$, see Section 3. Thus, we obtain the result (i) at step $n = 0$. Assume now that $\pi^n \in \mathscr{P}_{\mathsf{T}_{k_n}}$ for some $n \in \mathbb{N}$.

Consider the paths $\mathsf{P}_n = \mathrm{path}_{\mathsf{T}_{k_n}}(i_{k_n}, i_{k_n+1})$ and $\mathsf{P}_{n+1} = \mathrm{path}_{\mathsf{T}_{k_n+1}}(i_{k_n+1}, i_{k_n})$. Note that these two paths have the same length, denoted by $J$, and contain the same vertices, denoted by $\mathsf{V}_n$. Let $\pi \in \mathscr{P}^{(\ell+1)}$ such that $\mathrm{KL}(\pi|\pi^n) < +\infty$. We have the following decomposition

$$\mathrm{KL}(\pi|\pi^n) = \mathrm{KL}(\pi_{\mathsf{V}_n}|\pi^n_{\mathsf{V}_n}) + \int_{(\mathbb{R}^d)^{J+1}} \mathrm{KL}(\pi_{|\mathsf{V}_n}|\pi^n_{|\mathsf{V}_n})\mathrm{d}\pi_{\mathsf{V}_n}(x_{\mathsf{V}_n}) \ .$$

Hence, the $(n+1)$-th iterate of (mIPF) is given by $\pi^{n+1} = \pi^{n+1}_{\mathsf{V}_n} \otimes \pi^n_{|\mathsf{V}_n}$, with

$$\pi^{n+1}_{\mathsf{V}_n} = \mathrm{argmin}\{\mathrm{KL}(\pi|\pi^n_{\mathsf{V}_n}) \ : \ \pi \in \mathscr{P}^{(J+1)}, \ \pi_{i_{k_n+1}} = \mu_{i_{k_n+1}}\} \ .$$

Since $\pi^n \in \mathscr{P}_{\mathsf{T}_{k_n}}$, we have (i) $\pi^n_{|\mathsf{V}_n} = \bigotimes_{(v,v')\in\mathsf{E}_{k_n}\backslash\mathsf{P}_n} \pi^n_{v'|v}$ and (ii) $\pi^n_{\mathsf{V}_n} \in \mathscr{P}_{\mathsf{P}_{n+1}}$ by Lemma 9, where $\mathsf{P}_{n+1}$ is viewed as a directed tree rooted in $i_{k_n+1}$. Defining $\mathsf{V}_{n+1} = \mathsf{V}_n \backslash \{i_{k_n+1}\}$, we thus have $\pi^n_{\mathsf{V}_n} = \pi^n_{i_{k_n+1}} \otimes \pi^n_{\mathsf{V}_{n+1}|i_{k_n+1}}$ where $\pi^n_{\mathsf{V}_{n+1}|i_{k_n+1}} = \bigotimes_{(v,v')\in\mathsf{P}_{n+1}} \pi^n_{v'|v}$.

Let $\pi \in \mathscr{P}^{(J+1)}$ such that $\pi_{i_{k_n+1}} = \mu_{i_{k_n+1}}$ and $\mathrm{KL}(\pi|\pi^n_{\mathsf{V}_n}) < +\infty$. Similarly to the previous computation, we have the following decomposition

$$\mathrm{KL}(\pi|\pi^n_{\mathsf{V}_n}) = \mathrm{KL}(\pi_{i_{k_n+1}}|\pi^n_{i_{k_n+1}}) + \int_{\mathbb{R}^d} \mathrm{KL}(\pi_{|i_{k_n+1}}|\pi^n_{\mathsf{V}_{n+1}|i_{k+1}})\mathrm{d}\pi_{i_{k_n+1}}(x_{i_{k_n+1}})$$
$$= \mathrm{KL}(\mu_{i_{k_n+1}}|\pi^n_{i_{k_n+1}}) + \int_{\mathbb{R}^d} \mathrm{KL}(\pi_{|i_{k_n+1}}|\pi^n_{\mathsf{V}_{n+1}|i_{k_n+1}})\mathrm{d}\mu_{i_{k_n+1}}(x_{i_{k_n+1}}) \ .$$

Therefore, we obtain

$$\pi^{n+1}_{\mathsf{V}_n} = \mu_{i_{k_n+1}} \otimes \pi^n_{\mathsf{V}_{n+1}|i_{k+1}} = \mu_{i_{k+1}} \bigotimes_{(v,v')\in\mathsf{P}_{n+1}} \pi^n_{v'|v} \ .$$

Noting that $\mathsf{E}_{k_n}\backslash\mathsf{P}_n = \mathsf{E}_{k_n+1}\backslash\mathsf{P}_{n+1}$ and recalling that $\pi^{n+1} = \pi^{n+1}_{\mathsf{V}_n} \otimes \pi^n_{|\mathsf{V}_n}$, it finally comes

$$\pi^{n+1} = \mu_{i_{k_n+1}} \bigotimes_{(v,v')\in\mathsf{P}_{n+1}} \pi^n_{v'|v} \bigotimes_{(v,v')\in\mathsf{E}_{k_n+1}\backslash\mathsf{P}_{n+1}} \pi^n_{v'|v} = \mu_{i_{k_n+1}} \bigotimes_{(v,v')\in\mathsf{E}_{k_n+1}} \pi^n_{v'|v} \ . \quad (5)$$

Therefore, $\pi^{n+1} \in \mathscr{P}_{\mathsf{T}_{k_n+1}}$, which achieves the recursion for (i), and we obtain (ii) by (5). $\qquad\square$

Hence, Lemma 12 shows that the (mIPF) iterates admit a Markovian factorization on $\mathsf{T}$, and can be defined recursively using the edges of $\mathsf{T}$. We now provide the proof of Proposition 2.

*Proof of Proposition 2.* We will prove this result by recursion on $n \in \mathbb{N}$. Observe that the initialisation is directly given by Proposition 1. Assume now that the result of Proposition 2 stands for some $n \in \mathbb{N}$. Let $k_n = (n-1)\bmod(K)$, $k_n + 1 = n\bmod(K)$. Denote by $\mathsf{T}_{k_n}$ with edges $\mathsf{E}_{k_n}$, respectively $\mathsf{T}_{k_n+1}$ with edges $\mathsf{E}_{k_n+1}$, the directed version of $\mathsf{T}$ rooted in $i_{k_n}$, respectively in $i_{k_n+1}$. For any vertex $v$ of $\mathsf{T}_{k_n+1}$, we define $p(v)$ as the (unique) parent of $v$ and $c(v)$ as the unique child of $v$ when it exists. Consider the $(n+1)$-th dynamic iterate defined by (a) and (b), *i.e.*, $(\mathbb{P}^{n+1}_{(v,v')})_{(v,v')\in\mathsf{E}_{k_n+1}}$. To prove that this iterate has the properties stated in Proposition 2, we proceed by recursion on the edges of $\mathsf{T}_{k_n+1}$, following the bread-first order in $\mathsf{T}_{k_n+1}$. In this order, the edge $(i_{k_n+1}, c(i_{k_n+1}))$ is the first to be considered. Remark that $c(i_{k_n+1})$ is well defined since $i_{k_n+1}$ is a leaf of $\mathsf{T}$.

Here, we denote $T_{c(i_{k_n+1}),i_{k_n+1}}$ by $T$. By construction, we have $\mathbb{P}^{n+1}_{(i_{k_n+1},c(i_{k_n+1}))} = \mu_{i_{k_n+1}} \otimes (\mathbb{P}^n_{(c(i_{k_n+1}),i_{k_n+1})})_{|0}^R$. By recursion assumption, $\mathbb{P}^n_{(c(i_{k_n+1}),i_{k_n+1})} \in \mathscr{P}(\mathrm{C}([0,T],\mathbb{R}^d))$ since

$(c(i_{k_n+1}), i_{k_n+1}) \in \mathsf{E}_{k_n}$. Then, $\mathbb{P}^{n+1}_{(i_{k_n+1}, c(i_{k_n+1}))}$ is a well defined path measure on $[0, T]$. By definition of the (mIPF) sequence, we have $\mu_{i_{k_n+1}} = \pi^{n+1}_{i_{k_n+1}}$. By recursion assumption, we also have that $\mathrm{Ext}(\mathbb{P}^n_{(c(i_{k_n+1}), i_{k_n+1})}) = \pi^n_{c(i_{k_n+1}), i_{k_n+1}}$. Hence, it comes that $(\mathbb{P}^n_{(c(i_{k_n+1}), i_{k_n+1})})^R_{T|0} = \pi^n_{c(i_{k_n+1})|i_{k_n+1}} = \pi^{n+1}_{c(i_{k_n+1})|i_{k_n+1}}$, where the last equality comes from Lemma 12. Finally, we obtain that $\mathrm{Ext}(\mathbb{P}^{n+1}_{(i_{k_n+1}, c(i_{k_n+1}))}) = \pi^{n+1}_{i_{k_n+1}, c(i_{k_n+1})}$, which proves the initialisation.

Assume now that $\mathbb{P}^{n+1}$ is well defined and has the right properties, up to some edge in $\mathsf{T}_{k_n+1}$. Consider the following edge, denoted by $(v, v') \in \mathsf{E}_{k_n+1}$, in the breadth-first order. By edge recursion, we have that $\mathrm{Ext}(\mathbb{P}^{n+1}_{(p(v), v)}) = \pi^{n+1}_{p(v), v}$, and thus $\mathbb{P}^{n+1}_{(p(v), v), T_{p(v), v}} = \pi^{n+1}_v$. Define the path $\mathsf{P}_n = \mathrm{path}_{\mathsf{T}_{k_n}}(i_{k_n}, i_{k_n+1})$. Then, we face two cases.

(i) Either $(v, v') \in \mathsf{E}_{k_n} \backslash \mathsf{P}_n$. Then, we have by (a) that

$$\mathbb{P}^{n+1}_{(v, v')} = \mathbb{P}^{n+1}_{(p(v), v), T_{p(v), v}} \otimes \mathbb{P}^n_{(v, v')|0} = \pi^{n+1}_v \otimes \mathbb{P}^n_{(v, v')|0}$$

In particular, $\mathbb{P}^{n+1}_{(v, v')}$ is a well defined path measure on $[0, T_{v, v'}]$. Since $(v, v') \in \mathsf{E}_{k_n}$, $\mathrm{Ext}(\mathbb{P}^n_{(v, v')}) = \pi^n_{v, v'}$ by recursion assumption. In particular, $\mathbb{P}^n_{(v, v'), T_{v, v'}|0} = \pi^n_{v'|v} = \pi^{n+1}_{v'|v}$ where the last equality comes from Lemma 12. We thus have $\mathrm{Ext}(\mathbb{P}^{n+1}_{(v, v')}) = \pi^{n+1}_{v, v'}$.

(ii) Or $(v', v) \in \mathsf{P}_n$. Then, we have by (b) that

$$\mathbb{P}^{n+1}_{(v, v')} = \mathbb{P}^{n+1}_{(p(v), v), T_{v, v'}} \otimes (\mathbb{P}^n_{(v', v)})^R_{|0} = \pi^{n+1}_v \otimes (\mathbb{P}^n_{(v', v)})^R_{|0}$$

In particular, $\mathbb{P}^{n+1}_{(v, v')}$ is a well defined path measure on $[0, T_{v, v'}]$. Here, $(v', v) \in \mathsf{E}_{k_n}$ and thus, $\mathrm{Ext}(\mathbb{P}^n_{(v', v)}) = \pi^n_{v', v}$ by recursion assumption. In particular, $(\mathbb{P}^n_{(v', v)})^R_{T_{v', v}|0} = \pi^n_{v'|v} = \pi^{n+1}_{v'|v}$ where the last equality comes from Lemma 12. We thus have $\mathrm{Ext}(\mathbb{P}^{n+1}_{(v, v')}) = \pi^{n+1}_{v, v'}$.

This achieves the recursion. $\qquad \square$

## D.2 Proofs of Section 4

**Remark on assumption A1.** Although *A1 is not needed to establish the result of Proposition 3, Corollary 4 and Proposition 5*, it is however crucial in the proof of convergence of (mIPF) stated in Proposition 6. Nevertheless, we choose to keep **A1** as an assumption in the statement of every theoretical result presented in Section 4 for sake of clarity.

**Additional definitions.** We define the set $\mathscr{P}_\mathsf{S} = \cap_{i \in \mathsf{S}} \mathscr{P}_i$, where $\mathscr{P}_i = \{\pi \in \mathscr{P}^{(\ell+1)} : \pi_i = \mu_i\}$, i.e., $\mathscr{P}_\mathsf{S}$ is the set of all probability measures $\pi \in \mathscr{P}^{(\ell+1)}$ which verify

$$\int_{(\mathbb{R}^d)^{\ell+1}} f_i(x_i) \mathrm{d}\pi(x_{0:\ell}) = \int_{\mathbb{R}^d} f_i(x_i) \mathrm{d}\mu_i(x_i) ,$$

for any family of bounded measurable functions $\{f_i\}_{i \in \mathsf{S}} \in \mathrm{C}(\mathbb{R}^d, \mathbb{R})^K$. Since $\mathbb{R}^d$ is separable, there exists a dense family of functions $\{f_i^k\}_{k \in \mathbb{N}^*, i \in \mathsf{S}}$, with $f_i^k \in \mathrm{L}^\infty(\mu_i)$ for any $k \in \mathbb{N}^*$ and any $i \in \mathsf{S}$, such that $\pi \in \mathscr{P}_\mathsf{S}$ if and only if

$$\int_{(\mathbb{R}^d)^{\ell+1}} f_i^k(x_i) \mathrm{d}\pi(x_{0:\ell}) = \int_{\mathbb{R}^d} f_i^k(x_i) \mathrm{d}\mu_i(x_i)$$

or equivalently, upon centering $f_i^k$,

$$\int_{(\mathbb{R}^d)^{\ell+1}} f_i^k(x_i) \mathrm{d}\pi(x_{0:\ell}) = 0 .$$

In the rest of the section, we consider such family $\{f_i^k\}_{k \in \mathbb{N}^*, i \in \mathsf{S}}$.

For any $n \in \mathbb{N}^*$, we also define $\mathscr{P}_\mathsf{S}^n = \cap_{i \in \mathsf{S}} \mathscr{P}_i^n$, where $\mathscr{P}_i^n = \{\pi \in \mathscr{P}^{(\ell+1)} : \int_{(\mathbb{R}^d)^{\ell+1}} f_i^k(x_i) \mathrm{d}\pi(x_{0:\ell}) = 0, \forall k \in \{1, \ldots, n\}\}$. In particular, we have

$$\mathscr{P}_\mathsf{S} = \cap_{n \in \mathbb{N}^*} \mathscr{P}_\mathsf{S}^n . \tag{6}$$

Finally, (static-mSB) can be rewritten as

$$\pi^\star = \mathrm{argmin}\{\mathrm{KL}(\pi \mid \pi^0) : \pi \in \mathscr{P}_\mathsf{S}\} . \tag{7}$$

**Proof of Proposition 3 and Corollary 4.** In this part of the section, we present an extension of the theoretical results from Nutz (2021) to the multi-marginal setting. We first present two technical results, Lemma 13 and Lemma 14, which are respectively adapted from (Nutz, 2021, Lemma 2.10.) and (Nutz, 2021, Lemma 2.11.).

**Lemma 13.** *Let $\{\tilde{\mu}_j\}_{j \in S^c}$ be a family of probability measures defined on $(\mathbb{R}^d, \mathcal{B}(\mathbb{R}^d))$. We define $\tilde{\pi}^0 = \bigotimes_{i \in S} \mu_i \bigotimes_{j \in S^c} \tilde{\mu}_j$. Let $A \in \bigotimes_{m=0}^{\ell} \mathcal{B}(\mathbb{R}^d)$ such that $\tilde{\pi}^0(A) = 1$. Then, for $\tilde{\pi}^0$-almost any $x^\star \in A$, there exists a family of sets $\{X_m^0\}_{m=0}^{\ell} \subset (\mathbb{R}^d)^{\ell+1}$ such that*

*(a) $\mu_i(X_i^0) = 1$ for any $i \in S$, and $\tilde{\mu}_j(X_j^0) = 1$ for any $j \in S^c$,*

*(b) $A^0 = A \cap (\prod_{m=0}^{\ell} X_m^0)$ satisfies $x^\star \in A^0$ and*

$$(x_0^\star, \ldots, x_{m-1}^\star, x_m, x_{m+1}^\star, \ldots, x_\ell^\star) \in A^0, \forall x \in A^0, \forall m \in \{0, \ldots, \ell\} .$$

*Proof.* Consider such set A. We define for any $m \in \{0, \ldots, \ell\}$ the set

$$X_m = \{u \in \mathbb{R}^d : \tilde{\pi}_{-m}^0(A_m^u) = 1\} ,$$

where $A_m^u = \{y \in (\mathbb{R}^d)^\ell : (y_0, \ldots, y_{m-1}, u, y_m, \ldots, y_{\ell-1}) \in A\}$.

Take $i \in S$. Assume that $\mu_i(X_i) < 1$. We recall that $\tilde{\pi}^0 = \tilde{\pi}_{-i}^0 \otimes \mu_i$. Using Fubini's theorem and that $\int_{A_i^{x_i}} d\tilde{\pi}_{-i}^0(x_{-i}) < 1$ for any $x_i \notin X_i$, we have

$$\begin{aligned}
1 = \tilde{\pi}^0(A) &= \int_A d\tilde{\pi}_{-i}^0(x_{-i}) \otimes d\mu_i(x_i) \\
&= \int_{\mathbb{R}^d} \{\int_{A_i^{x_i}} d\tilde{\pi}_{-i}^0(x_{-i})\} d\mu_i(x_i) \\
&= \int_{X_i} \{\int_{A_i^{x_i}} d\tilde{\pi}_{-i}^0(x_{-i})\} d\mu_i(x_i) + \int_{X_i^c} \{\int_{A_i^{x_i}} d\tilde{\pi}_{-i}^0(x_{-i})\} d\mu_i(x_i) \\
&< \mu_i(X_i) + \mu_i(X_i^c) = 1 ,
\end{aligned}$$

which is absurd. Therefore, we obtain $\mu_i(X_i) = 1$, and similarly, we have $\tilde{\mu}_j(X_j) = 1$ for any $j \in S^c$. For any $y \in (\mathbb{R}^d)^\ell$, any $m \in \{0, \ldots, \ell\}$, we define the set

$$\bar{A}_m^y = \{u \in \mathbb{R}^d : (y_0, \ldots, y_{m-1}, u, y_m, \ldots, y_{\ell-1}) \in A\} .$$

Let $i \in S$. We have by Fubini's theorem

$$\begin{aligned}
1 = \tilde{\pi}^0(A) &= \int_A d\mu_i(x_i) \otimes d\tilde{\pi}_{-i}^0(x_{-i}) \\
&= \int_{(\mathbb{R}^d)^\ell} \{\int_{\bar{A}_i^{x_{-i}}} d\mu_i(x_i)\} d\tilde{\pi}_{-i}^0(x_{-i}) \\
&= \int_{\prod_{\substack{m=0 \\ m \neq i}}^{\ell} X_i} \{\int_{\bar{A}_i^{x_{-i}}} d\mu_i(x_i)\} d\tilde{\pi}_{-i}^0(x_{-i}) ,
\end{aligned}$$

where the last equality comes from the fact that $\mu_i(X_i) = 1$ for any $i \in S$, $\tilde{\mu}_j(X_j) = 1$ for any $j \in S^c$ and that $\tilde{\pi}^0 = \bigotimes_{i \in S} \mu_i \bigotimes_{j \in S^c} \tilde{\mu}_j$. Consequently, there exists a measurable set $A_{-i} \subset \prod_{\substack{m=0 \\ m \neq i}}^{\ell} X_i$ such that the following properties hold: (a) $\mu_i(\bar{A}_i^y) = 1$ for any $y \in A_{-i}$, (b) $\tilde{\pi}_{-i}^0(A_{-i}) = 1$. Similarly, this result holds for any $j \in S^c$, *i.e.*, there exists a measurable set $A_{-j} \subset \prod_{\substack{m=0 \\ m \neq j}}^{\ell} X_i$ such that the following properties hold: (a) $\tilde{\mu}_j(\bar{A}_j^y) = 1$ for any $y \in A_{-j}$, (b) $\tilde{\pi}_{-j}^0(A_{-j}) = 1$. We consider such sets $\{A_{-m}\}_{m=0}^{\ell}$ for the rest of the proof and finally define the set

$$\tilde{A} = \cap_{m=0}^{\ell} \tilde{A}_m ,$$

where $\tilde{A}_m = A_{-m} \times \{u \in \bar{A}_m^y : y \in A_{-m}\}$. By definition, we have $\tilde{A} \subset A \cap \prod_{m=0}^{\ell} X_m$, using the fact that $\tilde{A}_m \subset A$ for any $m \in \{0, \ldots, \ell\}$. In addition, for any $i \in S$, we get by Fubini's theorem

$$\tilde{\pi}^0(\tilde{A}_i) = \int_{\tilde{A}_i} d\mu_i(x_i) \otimes d\tilde{\pi}_{-i}^0(x_{-i}) = \int_{A_{-i}} \{\int_{\bar{A}_i^{x_{-i}}} d\mu_i(x_i)\} d\tilde{\pi}_{-i}^0(x_{-i}) = \tilde{\pi}_{-i}^0(A_{-i}) = 1 ,$$

and similarly, we get $\tilde{\pi}^0(\tilde{A}_j) = 1$ for any $j \in S^c$. We can deduce that $\tilde{\pi}^0(\tilde{A}) = 1$ since $\tilde{\pi}^0(\tilde{A}^c) \leq \sum_{m=0}^{\ell} \tilde{\pi}^0(\tilde{A}_m^c) = 0$.

Let $x^\star \in \tilde{\mathsf{A}}$. In particular, $x^\star \in \mathsf{A}$. We define the set $\mathsf{A}^0 = \mathsf{A} \cap (\prod_{m=0}^{\ell} \mathsf{X}_m^0)$, where $\mathsf{X}_m^0 = \mathsf{X}_m \cap \bar{\mathsf{A}}_m^{x_{-m}^\star}$ for any $m \in \{0, \ldots, \ell\}$. We now establish the result of Lemma 13.

We first prove (a). Let $i \in \mathsf{S}$. Since $x^\star \in \tilde{\mathsf{A}}$, we have $x^\star \in \tilde{\mathsf{A}}_i$ and therefore $x_{-i}^\star \in \mathsf{A}_{-i}$. By definition of $\mathsf{A}_{-i}$, we obtain that $\mu_i(\bar{\mathsf{A}}_i^{x_{-i}^\star}) = 1$ and thus,

$$\mu_i(\{\mathsf{X}_i^0\}^c) \leq \mu_i(\mathsf{X}_i^c) + \mu_i(\{\bar{\mathsf{A}}_i^{x_{-i}^\star}\}^c) = 0,$$

which gives $\mu_i(\mathsf{X}_i^0) = 1$, and similarly, we have $\tilde{\mu}_j(\mathsf{X}_j^0) = 1$ for any $j \in \mathsf{S}^c$.

We now prove (b). Let $m \in \{0, \ldots, \ell\}$. Since $x^\star \in \tilde{\mathsf{A}} \subset \mathsf{A}$, we get $x_m^\star \in \bar{\mathsf{A}}_m^{x_{-m}^\star}$. Using that $\tilde{\mathsf{A}} \subset \mathsf{A} \cap_{m=0}^{\ell} \mathsf{X}_m$, we get $x^\star \in \mathsf{A}^0$. Let $x \in \mathsf{A}^0$. We denote $x^m = (x_0^\star, \ldots, x_{m-1}^\star, x_m, x_{m+1}^\star, \ldots, x_\ell^\star)$. We need to show that $x^m \in \mathsf{A}$ and $x^m \in \prod_{j=1}^{\ell} \mathsf{X}_j^0 = \prod_{j=1}^{\ell} (\mathsf{X}_j \cap \bar{\mathsf{A}}_j^{x_{-m}^\star})$. First, since $x_j^m = x_j$ or $x_j^\star$ for any $j \in \{0, \ldots, \ell\}$, and $x \in \mathsf{A}^0$ and $x^\star \in \mathsf{A}^0$, we get that for any $j \in \{0, \ldots, \ell\}$, $x_j^m \in \mathsf{X}_j$. Similarly, for any $j \in \{0, \ldots, \ell-1\}$, $x_j^m \in \bar{\mathsf{A}}_j^{x_{-m}^\star}$. Therefore, we get that $x^m \in \prod_{j=1}^{\ell} (\mathsf{X}_j \cap \bar{\mathsf{A}}_j^{x_{-m}^\star})$. Since $x_m \in \mathsf{A}_m^{x_{-m}^\star}$ (because $x \in \prod_{j=1}^{\ell} (\mathsf{X}_j \cap \bar{\mathsf{A}}_j^{x_{-m}^\star})$), we get that $x \in \mathsf{A}$, which concludes the proof. $\square$

**Lemma 14.** *Let $\mathsf{A}^0 \subset (\mathbb{R}^d)^{\ell+1}$. For any $m \in \{0, \ldots, \ell\}$, we denote $\mathsf{X}_m^0 = \mathrm{proj}_m(\mathsf{A}^0)$. We make the following assumptions.*

*(a) Assume there exists $x^\star \in \mathsf{A}^0$ such that for any $x \in \mathsf{A}^0$, for any $m \in \{0, \ldots, \ell\}$, we have $(x_0^\star, \ldots, x_{m-1}^\star, x_m, x_{m+1}^\star, \ldots, x_\ell^\star) \in \mathsf{A}^0$.*

*(b) Assume there exists a family of functions $\{\varphi_{i_k}^n\}_{n \in \mathbb{N}^*, k \in \{0, \ldots, K-1\}}$ with $\varphi_{i_k}^n : \mathsf{X}_{i_k}^0 \to [-\infty, +\infty]$ such that for any $n \in \mathbb{N}^*$ and any $k \in \{0, \ldots, K-2\}$, we have $\varphi_{i_k}^n(x_{i_k}^\star) = 0$.*

*(c) Denote $F^n(x) = \sum_{k=0}^{K-1} \varphi_{i_k}^n(x_{i_k})$ for any $x \in \mathsf{A}^0$. Assume that for any $x \in \mathsf{A}^0$, $F(x) = \lim_{n \to \infty} F^n(x)$ exists and is such that $F(x) \in [-\infty, +\infty)$ with $F(x^\star) \in \mathbb{R}$.*

*Then, for any $i \in \mathsf{S}$, for any $x_i \in \mathsf{X}_i^0$, $\varphi_i(x_i) = \lim_{n \to \infty} \varphi_i^n(x_i)$ exists and is such that $\varphi_i(x_i) \in [-\infty, +\infty)$.*

*Proof.* Consider $\mathsf{A}^0 \subset (\mathbb{R}^d)^{\ell+1}$ such that assumptions (a), (b) and (c) hold. Remark that we have $F^n(x^\star) = \varphi_{i_{K-1}}^n(x_{i_{K-1}}^\star)$.

Let $x \in \mathsf{A}^0$. We denote $x^m = (x_1^\star, \ldots, x_{m-1}^\star, x_m, x_{m+1}^\star, \ldots, x_\ell^\star)$ for any $m \in \{0, \ldots, \ell\}$. In particular, we have $x^m \in \mathsf{A}^0$ by assumption (a). Let us define

$$\varphi_{i_k}(x_{i_k}) = F(x^{i_k}) - F(x^\star), \quad \forall k \in \{0, \ldots, K-2\},$$
$$\varphi_{i_{K-1}}(x_{i_{K-1}}) = F(x^{i_{K-1}}).$$

Using assumption (c), we have $\varphi_i(x_i) \in [-\infty, +\infty)$ for any $i \in \mathsf{S}$. Let $k \in \{0, \ldots, K-2\}$. We have by definition of $F^n$,

$$\varphi_{i_k}^n(x_{i_k}) = F^n(x^{i_k}) - \sum_{\substack{m=0 \\ m \neq k}}^{K-1} \varphi_{i_m}^n(x_{i_m}^\star) = F^n(x^{i_k}) - F^n(x^\star),$$

where we used assumption (b) in the last equality. Since $x^{i_k} \in \mathsf{A}^0$ and $x^\star \in \mathsf{A}^0$, we have by assumption (c),

$$\lim_{n \to \infty} \varphi_{i_k}^n(x_{i_k}) = F(x^{i_k}) - F(x^\star) = \varphi_{i_k}(x_{i_k}).$$

Furthermore, by combining the definition of $F^n$ with assumption (b), we have

$$\lim_{n \to \infty} \varphi_{i_{K-1}}^n(x_{i_{K-1}}) = F(x^{i_{K-1}}) = \varphi_{i_{K-1}}(x_{i_{K-1}}),$$

which concludes the proof. $\square$

In what follows, before proving Proposition 3, we respectively show in Proposition 15 and Proposition 16 how **A**2 and **A**3 can be satisfied in the case where $\pi^0 \in \mathscr{P}_{\mathsf{T}_r}$, as in (2), that is

$$\pi^0 = \pi_r^0 \bigotimes_{(v,v') \in \mathsf{E}_r} \pi_{v'|v}^0 .$$

**Proposition 15.** *Let $\pi^0 \in \mathscr{P}_{\mathsf{T}_r}$. Assume that $\pi_r^0 = \mu_r$ if $r \in \mathsf{S}$ or $\pi_r^0 = \mathrm{N}(m_r, \sigma_r \mathrm{Id})$, with $m_r \in \mathbb{R}^d$ and $\sigma_r > 0$ if $r \in \mathsf{S}^c$. In addition, assume that for any $(v, v') \in \mathsf{E}_r$, $\pi_{v'|v}^0(\cdot|x_v) = \mathrm{N}(x_v, \sigma_{v,v'} \mathrm{Id})$ with $\sigma_{v,v'} > 0$. Finally, assume that for any $i \in \mathsf{S}$, $\int_{\mathbb{R}^d} \|x\|^2 \mathrm{d}\mu_i(x) < +\infty$ and $\mathrm{H}(\mu_i) < +\infty$. Then **A**2 is satisfied.*

*Proof.* Let $\pi = \otimes_{i \in \mathsf{S}} \mu_i \otimes_{i \in \mathsf{S}^c} \nu_i$ with $\nu_i$ any Gaussian measure with positive definite covariance matrix. First, we have that

$$\mathrm{KL}(\pi \mid \pi^0) = \mathrm{KL}(\pi_r \mid \pi_r^0) + \sum_{(v,v') \in \mathsf{E}_r} \int_{\mathbb{R}^d} \mathrm{KL}(\pi_{v'|v}|\pi_{v'|v}^0) \mathrm{d}\pi_v .$$

For any $(v, v') \in \mathsf{E}_r$, there exists $C_{v,v'} \geq 0$ such that

$$\int_{\mathbb{R}^d} \mathrm{KL}(\pi_{v'|v}|\pi_{v'|v}^0) \mathrm{d}\pi_v \leq C_{v,v'} - \mathrm{H}(\pi_{v'}) + \int_{\mathbb{R}^d \times \mathbb{R}^d} \|x_v - x_{v'}\|^2 / (2\sigma_{v,v'}^2) \mathrm{d}\pi_v \otimes \pi_{v'}(x_v, x_{v'})$$

$$\leq C_{v,v'} - \mathrm{H}(\pi_{v'}) + (1/\sigma_{v,v'}^2) \int_{\mathbb{R}^d} \|x_v\|^2 \mathrm{d}\pi_v(x_v) + (1/\sigma_{v,v'}^2) \int_{\mathbb{R}^d} \|x_{v'}\|^2 \mathrm{d}\pi_{v'}(x_{v'}) < +\infty.$$

We conclude the proof upon remarking that $\mathrm{KL}(\pi_r \mid \pi_r^0) < +\infty$. □

**Proposition 16.** *Let $\pi^0 \in \mathscr{P}_{\mathsf{T}_r}$. Assume that $\pi_r^0 = \mu_r$ if $r \in \mathsf{S}$ or $\pi_r^0 = \mathrm{N}(m_r, \sigma_r \mathrm{Id})$, with $m_r \in \mathbb{R}^d$ and $\sigma_r > 0$ if $r \in \mathsf{S}^c$. In addition, assume that for any $(v, v') \in \mathsf{E}_r$, $\pi_{v'|v}^0(\cdot|x_v) = \mathrm{N}(x_v, \sigma_{v,v'} \mathrm{Id})$ with $\sigma_{v,v'} > 0$. Finally, assume that for any $i \in \mathsf{S}$, $\mu_i$ admits a positive density w.r.t. the Lebesgue measure. Then **A**3 is satisfied.*

*Proof.* We have that $\pi^0$ admits a positive density w.r.t the Lebesgue measure. Letting $\tilde{\pi}^0 = \bigotimes_{i \in \mathsf{S}} \mu_i \bigotimes_{j \in \mathsf{S}^c} \tilde{\mu}_j$ where $\tilde{\mu}_j$ which admits a positive density w.r.t. the Lebesgue measure for any $j \in \mathsf{S}^c$, we get that $\tilde{\pi}^0$ admits a positive density w.r.t. the Lebesgue measure and therefore $\pi^0 \sim \tilde{\pi}^0$, which concludes the proof. □

Using the preliminary results presented above, we are now ready to prove Proposition 3.

*Proof of Proposition 3.* Assume **A**1 and **A**2. Since $\mathscr{P}_\mathsf{S}$ is convex and closed in total-variation norm, there exists a probability distribution $\pi^\star$ solution to (7), or equivalently to (static-mSB), by using **A**2 with (Csiszár, 1975, Theorem 2.1.). Moreover, this solution is unique by strict convexity of $\mathrm{KL}(\cdot \mid \pi^0)$.

We now turn to the proof of existence of potentials defining $(\mathrm{d}\pi^\star/\mathrm{d}\pi^0)$, by adapting the arguments of (Nutz, 2021, Section 2.3.). Define $\nu^n = \operatorname{argmin}\{\mathrm{KL}(\pi \mid \pi^0) : \pi \in \mathscr{P}_\mathsf{S}^n\}$ for any $n \in \mathbb{N}^*$. Since $\{\mathscr{P}_\mathsf{S}^n\}_{n \in \mathbb{N}^*} \subset \mathscr{P}^{(\ell+1)}$ is a decreasing sequence of sets that are convex and closed in total-variation norm such that (6) holds, we get from (Nutz, 2021, Proposition 1.17.) with **A**2 that

$$\lim_{n \to \infty} \|\nu^n - \pi^\star\|_{\mathrm{TV}} = 0 ,$$

or equivalently

$$\lim_{n \to \infty} \|(\mathrm{d}\nu^n/\mathrm{d}\pi^0) - (\mathrm{d}\pi^\star/\mathrm{d}\pi^0)\|_{\mathrm{L}^1(\pi^0)} = 0 . \tag{8}$$

Following (Nutz, 2021, Example 1.18), there exists a family of bounded measurable functions $\{\varphi_i^n\}_{n \in \mathbb{N}^*, i \in \mathsf{S}}$ with $\varphi_i^n : \mathbb{R}^d \to \mathbb{R}$ such that for any $n \in \mathbb{N}^*$

$$(\mathrm{d}\nu^n/\mathrm{d}\pi^0) = \exp[\bigoplus_{i \in \mathsf{S}} \varphi_i^n] . \tag{9}$$

We consider such family $\{\varphi_i^n\}_{n \in \mathbb{N}^*, i \in \mathsf{S}}$ for the rest of the proof. By combining (8) and (9), we obtain, up to extraction,

$$(\mathrm{d}\pi^\star/\mathrm{d}\pi^0) = \lim_{n \to \infty} \exp[\bigoplus_{i \in \mathsf{S}} \varphi_i^n] \quad \pi^0\text{-a.s.} . \tag{10}$$

We now define the following sets

$$\mathsf{A}^\star = \{x \in (\mathbb{R}^d)^{\ell+1} : \lim_{n\to\infty} \bigoplus_{i\in\mathsf{S}} \varphi_i^n(x_i) \in [-\infty, +\infty)\} \,,$$

$$\mathsf{B}^\star = \{x \in (\mathbb{R}^d)^{\ell+1} : \lim_{n\to\infty} \bigoplus_{i\in\mathsf{S}} \varphi_i^n(x_i) > -\infty\} \subset \mathsf{A}^\star$$

Using (10), we have $\pi^0(\mathsf{A}^\star) = 1$. Using **A**3, it comes $\tilde{\pi}^0(\mathsf{A}^\star) = 1$. Moreover, we also get that $\pi^\star(\mathsf{B}^\star) = 1$ by (10). Thus, it comes $\pi^0(\mathsf{B}^\star) > 0$, and $\tilde{\pi}^0(\mathsf{B}^\star) > 0$ using **A**3.

We then apply Lemma 13 to $\tilde{\pi}^0$ and $\mathsf{A} = \mathsf{A}^\star$. Since $\tilde{\pi}^0(\mathsf{B}^\star) > 0$, it implies that there exists $x^\star \in \mathsf{B}^\star$ and a measurable set $\mathsf{A}^0 \subset \mathsf{B}^\star$ verifying the properties (a) and (b). Following (Nutz, 2021, Corollary 2.12), we may assume without loss of generality in the statement of Lemma 13 that the sets $\mathsf{X}_m^0$ are measurable with $\prod_{m=0}^\ell \mathsf{X}_m^0 \subset \mathsf{A}$. In this case, we obtain that $\mu_i(\mathrm{proj}_i(\mathsf{A}^0)) = 1$ for any $i \in \mathsf{S}$.

We now aim at applying Lemma 14 to the set $\mathsf{A}^0$. Remark that $\mathsf{A}^0$ directly satisfies assumption (a). For any $n \in \mathbb{N}^*$, consider the following transformation of the functions $\{\varphi_i^n\}_{i\in\mathsf{S}}$

$$\varphi_{i_k}^n \leftarrow \varphi_{i_k}^n - \varphi_{i_k}^n(x_{i_k}^\star), \quad \forall k \in \{0, \dots, K-2\} \,,$$

$$\varphi_{i_{K-1}}^n \leftarrow \varphi_{i_{K-1}}^n + \sum_{k=0}^{K-2} \varphi_{i_k}^n(x_{i_k}^\star) \,.$$

For any $i \in \mathsf{S}$, we restrict $\varphi_{i_k}^n$ to $\mathsf{X}_{i_k}^0$, so that the family $\{\varphi_i^n\}_{n\in\mathbb{N}^*, i\in\mathsf{S}}$ now verifies assumption (b). Finally, since $\mathsf{A}^0 \subset \mathsf{A}^\star$ and $x^\star \in \mathsf{B}^\star$, we directly obtain assumption (c).

Therefore, Lemma 14 may be applied. It provides us with the family of functions $\{\varphi_i\}_{i\in\mathsf{S}}$ defined by $\varphi_i : \mathsf{X}_i^0 \to [-\infty, +\infty)$ with $\varphi_i = \lim_{n\to\infty} \varphi_i^n$ $\mu_i$-a.s. for any $i \in \mathsf{S}$. Since $\mu_i(\mathrm{proj}_i(\mathsf{A}^0)) = 1$ for any $i \in \mathsf{S}$, we may extend the functions $\varphi_i$ to $\mathbb{R}^d$. In particular, we can find a family of functions $\{\psi_i^\star\}_{i\in\mathsf{S}}$ with $\psi_i^\star : \mathbb{R}^d \to [-\infty, +\infty)$ such that $\psi_i^\star = \varphi_i$ $\mu_i$-a.s. Note that these functions are measurable as limits of measurable functions.

Since $\pi^0 \sim \tilde{\pi}^0$ by **A**3, (10) turns into

$$(\mathrm{d}\pi^\star/\mathrm{d}\pi^0) = \exp[\bigoplus_{i\in\mathsf{S}} \psi_i^\star] \quad \pi^0\text{-a.s.} \,. \tag{11}$$

Finally, we show that the functions $\psi_i^\star$ are $\mu_i$-a.s. finite. Let $i \in \mathsf{S}$. Let us define $\mathsf{A}_i = \{x_i \in \mathbb{R}^d : \psi_i^\star(x_i) = -\infty\}$. Using (11), we obtain $(\mathrm{d}\pi^\star/\mathrm{d}\pi^0)(\mathsf{A}_i \times (\mathbb{R}^d)^\ell) = 0$. Since $\pi_i^\star = \mu_i$, we have

$$\mu_i(\mathsf{A}_i) = \pi^\star(\mathsf{A}_i \times (\mathbb{R}^d)^\ell) = \int_{\mathsf{A}_i \times (\mathbb{R}^d)^\ell} (\mathrm{d}\pi^\star/\mathrm{d}\pi^0)\mathrm{d}\pi^0 = 0 \,,$$

which gives the result. $\qquad\square$

We now turn to the proof of Corollary 4, which states that the iterates of (mIPF) can be expressed via potentials, in the same manner as the solution $\pi^\star$ to (static-mSB).

*Proof of Corollary 4.* Assume **A**1, **A**2 and **A**3. We prove the result of this corollary by recursion on $n \in \mathbb{N}^*$. First take $n = 1$. In this case, the first iteration of (mIPF) is a multi-marginal SB problem of the form (static-mSB) where $S = \{i_0\}$ with reference measure $\pi^0$. Therefore, using **A**2 and **A**3, we can apply Proposition 3 and obtain existence of $\psi_{i_0}^1 : \mathbb{R}^d \to \mathbb{R}$ such that

$$(\mathrm{d}\pi^1/\mathrm{d}\pi^0) = \exp[\psi_{i_0}^1] \quad \pi^0\text{-a.s} \,.$$

By taking $\psi_{i_k}^0 = 0$ for $k \in \{1, \dots, K-1\}$, we thus obtain the result at step $n = 1$.

Now assume that the result is verified for some $n \in \mathbb{N}^*$, with $k_n = (n-1) \bmod(K)$. We define $k_n + 1 = n \bmod(K)$ and $q_n \in \mathbb{N}$ as the quotient of the Euclidean division of $n$ by $K$. In this case, the $(n+1)$-th iteration of (mIPF) is a multi-marginal SB problem of the form (static-mSB) where $\mathsf{S} = \{i_{k_n+1}\}$ with reference measure $\pi^n$. Using (13), we have that **A**2 is satisfied for this new (static-mSB) problem. **A**1 and **A**3 are satisfied for this problem, given the form of $\pi^n$. Therefore, we can apply Proposition 3 and obtain existence of $\psi_{i_{k_n+1}}^{q_n+1} : \mathbb{R}^d \to \mathbb{R}$ such that

$$(\mathrm{d}\pi^{n+1}/\mathrm{d}\pi^n) = \exp[\psi_{i_{k_n+1}}^{q_n+1}] \quad \pi^n\text{-a.s} \,. \tag{12}$$

By assumption, we have that $\pi^n \ll \pi^0$. Hence, we obtain $\pi^{n+1} \ll \pi^0$ and thus,

$$(\mathrm{d}\pi^{n+1}/\mathrm{d}\pi^0) = (\mathrm{d}\pi^{n+1}/\mathrm{d}\pi^n)(\mathrm{d}\pi^n/\mathrm{d}\pi^0) \quad \pi^0\text{-a.s} \,.$$

By combining (12) with the result of the recursion at step $n$, we directly obtain the result at step $n + 1$, which achieves the proof. $\qquad\square$

**Proofs of Proposition 5 and Proposition 6.** In this part of the section, we establish the proofs of results related to the convergence of (mIPF), respectively Proposition 5 and Proposition 6, which can be seen as a natural extension of (Ruschendorf, 1995, Proposition 2.1.) and (Ruschendorf, 1995, Theorem 3.1.).

*Proof of Proposition 5.* Under **A**1 and **A**2, we obtain by Proposition 3 existence and uniqueness of a solution to (static-mSB), which we denote by $\pi^\star$. Since $\pi^\star \in \mathscr{P}_{\mathsf{S}}$, using recursively (Csiszár, 1975, Theorem 3.12.), the fact that $\{\pi_{i_k} = \mu_{i_k} \,:\, \pi \in \mathscr{P}^{(|\mathsf{V}|)}\}$ is convex for any $k \in \{0, \dots, K-1\}$ and (mIPF), we obtain

$$\mathrm{KL}(\pi^\star \mid \pi^0) = \mathrm{KL}(\pi^\star \mid \pi^n) + \sum_{i=1}^n \mathrm{KL}(\pi^i \mid \pi^{i-1}) \,. \tag{13}$$

Therefore, we have $\sum_{i=1}^\infty \mathrm{KL}(\pi^i \mid \pi^{i-1}) \leq \mathrm{KL}(\pi^\star \mid \pi^0) < \infty$ and thus,

$$\lim_{i \to +\infty} \mathrm{KL}(\pi^i \mid \pi^{i-1}) = 0 \,. \tag{14}$$

Let $n \in \mathbb{N}^*$ with $n > 2K$, $k \in \{0, \dots, K-1\}$ and let $q_n \in \mathbb{N}$ be the quotient of the Euclidean division of $n-1$ by $K$. We define $n_k = q_n K + k + 1$ with $(n_k - 1) = k \bmod(K)$ if $n_k \leq n$. Otherwise, we set $n_k = (q_n - 1)K + k + 1$ with $(n_k - 1) = k \bmod(K)$. Note that we always have $|n - n_k| \leq 2K$. In particular, we have $\pi_{i_k}^{n_k} = \mu_{i_k}$ by definition of (mIPF). Therefore, we obtain

$$\begin{aligned}
\|\pi_{i_k}^n - \mu_{i_k}\|_{\mathrm{TV}} &\leq \|\pi^n - \pi^{n_k}\|_{\mathrm{TV}} \\
&\leq \|\pi^n - \pi^{n-1}\|_{\mathrm{TV}} + \dots + \|\pi^{n_k+1} - \pi^{n_k}\|_{\mathrm{TV}} &\text{(triangle inequality)} \\
&\leq (2\mathrm{KL}(\pi^n \mid \pi^{n-1}))^{1/2} + \dots + (2\mathrm{KL}(\pi^{n_k+1} \mid \pi^{n_k}))^{1/2} \,, &\text{(Pinsker's inequality)}
\end{aligned}$$

where each term goes to 0 as $n \to +\infty$ in the last inequality by (14), which achieves the proof. □

We now turn to the proof Proposition 6, which requires several preliminary technical results. For the rest of this section, we define, for any $n \in \mathbb{N}$, $q_n$ as the quotient of the Euclidean division of $n-1$ by $K$ (in particular, $q_0 = -1$).

**Schrödinger equations.** Under **A**1, **A**2 and **A**3, we know from Proposition 3 that the unique solution $\pi^\star$ to (static-mSB) can be $\pi^0$-a.s. written as $(\mathrm{d}\pi^\star/\mathrm{d}\pi^0) = \exp[\bigoplus_{i \in \mathsf{S}} \psi_i^\star]$, where $\{\psi_i^\star\}_{i \in \mathsf{S}}$ are measurable potentials, referred to as *Schrödinger potentials*. These functions are determined by the fixed-point *Schrödinger equations*

$$\psi_i^\star(x_i) = \log[r_i(x_i)/\textstyle\int_{(\mathbb{R}^d)^\ell} \exp[\sum_{j \in \mathsf{S}\setminus\{i\}} \psi_j^\star(x_j)] h(x_{0:\ell}) \mathrm{d}\nu_{-i}(x_{-i})] \quad \mu_i\text{-a.s.}, \quad \forall i \in \mathsf{S} \,,$$

which are obtained by marginalising $\pi^\star$ along its constrained marginals. This family of potentials is not unique. Indeed, for any family of real numbers $\{\lambda_{i_k}\}_{k \in \{0, \dots, K-2\}}$, we have

$$(\mathrm{d}\pi^\star/\mathrm{d}\pi^0) = \exp[\textstyle\bigoplus_{i \in \mathsf{S}} \tilde{\psi}_i] \,,$$

where $\tilde{\psi}_{i_k} = \psi_{i_k}^\star + \tilde{\lambda}_{i_k}$ for any $k \in \{0, \dots, K-1\}$ with $\tilde{\lambda}_{i_k} = \lambda_{i_k}$ if $k \in \{0, \dots, K-2\}$ and $\tilde{\lambda}_{i_{K-1}} = -\sum_{i=0}^{K-2} \lambda_{i_k}$.

**Remark on the initialisation of (mIPF).** Consider a probability measure $\bar{\pi}^0 \in \mathscr{P}^{(\ell+1)}$ of the form

$$(\mathrm{d}\bar{\pi}^0/\mathrm{d}\pi^0) = \exp[\textstyle\bigoplus_{i \in \mathsf{S}} \psi_i^0] \,, \tag{15}$$

where $\{\psi_i^0\}_{i \in \mathsf{S}}$ is a family of measurable potentials with $\psi_i^0 : \mathbb{R}^d \to \mathbb{R}$ such that $\left|\int_{\mathbb{R}^d} \psi_i^0 \mathrm{d}\mu_i\right| < \infty$ for any $i \in \mathsf{S}$. Then, for any $\pi \in \mathscr{P}_{\mathsf{S}}$, we have

$$\mathrm{KL}(\pi \mid \pi^0) = \mathrm{KL}(\pi \mid \bar{\pi}^0) + \textstyle\int_{(\mathbb{R}^d)^K} \bigoplus_{i \in \mathsf{S}} \psi_i^0 \mathrm{d}\pi = \mathrm{KL}(\pi \mid \bar{\pi}^0) + \sum_{i \in \mathsf{S}} \int_{\mathbb{R}^d} \psi_i^0 \mathrm{d}\mu_i \,.$$

Hence, (static-mSB) is equivalent to the multi-marginal SB problem

$$\mathrm{argmin}\{\mathrm{KL}(\pi|\bar{\pi}^0) \,:\, \pi \in \mathscr{P}^{(\ell+1)}, \, \pi_i = \mu_i \,, \forall i \in \mathsf{S}\} \,.$$

We refer to (Peyré et al., 2019, Proposition 4.2) for the EOT counterpart of this result. This means that the solutions of the multi-marginal Schrödinger Bridge problem are invariant by multiplication of the reference measure by potentials on the *fixed* marginals. Consequently, the initialisation of the (mIPF) sequence may be chosen as $\bar{\pi}^0$ instead of $\pi^0$.

**For sake of clarity, we now refer to the reference probability measure of (static-mSB) as $\bar{\pi}$ or $\pi^{-1}$ and to the initialisation of the (mIPF) iterates as $\pi^0$.**

**Solving (mIPF) with potentials.** To prove the convergence of the (mIPF) iterates to the solution $\pi^\star$ given by Proposition 3, we first rewrite these iterates with potentials, following the form of $\pi^\star$.

To do so, we recursively define the sequence of potentials $\{\psi_i^n\}_{n\in\mathbb{N}, i\in\mathsf{S}}$ by

$$\psi_{i_0}^0 = \ldots = \psi_{i_{K-2}}^0 = 0 , \tag{16}$$

$$\psi_{i_{K-1}}^0(x_{i_{K-1}}) = \log(r_{i_{K-1}}(x_{i_{K-1}})/\int_{(\mathbb{R}^d)^\ell} h(x_{0:\ell})\mathrm{d}\nu_{-i_{K-1}}(x_{-i_{K-1}})) ,$$

and for any $n \in \mathbb{N}^*$ and $k \in \{0, \ldots, K-1\}$

$$\psi_{i_k}^{q_n+1}(x_{i_k}) = \log[r_{i_k}(x_{i_k})/\int_{(\mathbb{R}^d)^\ell} \exp[\bigoplus_{\ell=0}^k \psi_{i_\ell}^{q_n+1}(x_{i_\ell}) \bigoplus_{m=k+1}^{K-1} \psi_{i_m}^{q_n}(x_{i_m})]$$

$$\times h(x_{0:\ell})\mathrm{d}\nu_{-i_k}(x_{-i_k})] , \tag{17}$$

recalling that $q_n$ is the quotient of the Euclidean division of $n-1$ by $K$.

We now define the sequence of probability measures $\{\pi^n\}_{n\in\mathbb{N}}$ by

$$\mathrm{d}\pi^n/\mathrm{d}\bar{\pi} = \exp[\bigoplus_{\ell=0}^{k_n} \psi_{i_\ell}^{q_n+1} \bigoplus_{m=k_n+1}^{K-1} \psi_{i_m}^{q_n}], \ k_n = (n-1) \bmod(K), \ n = q_n K + k_n + 1. \tag{18}$$

In particular, we have $(\mathrm{d}\pi^0/\mathrm{d}\bar{\pi}) = \exp[\oplus_{\ell=0}^{K-1} \psi_{i_\ell}^0] = \exp[\psi_{i_{K-1}}^0]$, and thus $\int_{\mathbb{R}^d} \psi_{i_{K-1}}^0 \mathrm{d}\mu_{i_{K-1}} = \mathrm{KL}(\mu_{i_{K-1}} \mid \bar{\pi}_{i_{K-1}})$. Consequently, $\pi^0$ can be chosen as the initialisation of (mIPF), following the previous remark, if we assume that $\mathrm{KL}(\mu_{i_{K-1}} \mid \bar{\pi}_{i_{K-1}}) < \infty$. In (TreeSB) with $r = i_{K-1}$, the latter assumption is directly verified since we choose $\bar{\pi}_{i_{K-1}} = \mu_{i_{K-1}}$.

Let $n \in \mathbb{N}$, with $k_n = (n-1) \bmod(K)$, $k_n + 1 = n \bmod(K)$. Using (16) and (17), we get that $\pi_{i_{k_n}}^n = \mu_{i_{k_n}}$. Moreover, we have

$$\mathrm{d}\pi^n/\mathrm{d}\pi^{n-1} = \exp[\psi_{i_{k_n}}^{q_n+1} - \psi_{i_{k_n}}^{q_n}] , \tag{19}$$

with the convention that $\psi_{i_{K-1}}^{-1} = 0$. In particular, we obtain that $\pi_{|i_{k_n+1}}^{n+1} = \pi_{|i_{k_n+1}}^n$.

In conclusion, the sequence $\{\pi^n\}_{n\in\mathbb{N}}$ defined in (18) verifies $\pi^{n+1} = \mu_{i_{k_n+1}}\pi_{|i_{k_n+1}}^n$ for any $n \in \mathbb{N}$. By decomposition property of the Kullback-Leibler divergence, this sequence solves (mIPF) with initialisation $\pi^0$. *We consider such iterates in the following.*

Since $\pi_{i_{k_n}}^n = \mu_{i_{k_n}}$, we have that

$$\mathrm{KL}(\pi^n \mid \pi^{n-1}) = \int_{\mathbb{R}^d} (\psi_{i_{k_n}}^{q_n+1} - \psi_{i_{k_n}}^{q_n})\mathrm{d}\mu_{i_{k_n}} . \tag{20}$$

Before proving a multi-marginal counterpart to (Ruschendorf, 1995, Lemma 4.1), we state and prove the following result.

**Proposition 17.** *Let $\pi_0, \pi_1$ two probability measures on $\mathbb{R}^d$ such that $\pi_0 \ll \pi_1$. Then, denoting $f = \mathrm{d}\pi_0/\mathrm{d}\pi_1$, the following assertions are equivalent:*

*(a) $\mathrm{KL}(\pi_0 \mid \pi_1) < +\infty$*

*(b) $\int_{\mathbb{R}^d} |\log(f)(x)|\mathrm{d}\pi_0(x) < +\infty$*

*(c) $\int_{\mathbb{R}^d} \log(f)(x)\mathbb{1}_{f(x)>1}\mathrm{d}\pi_0(x) < +\infty$*

*If one of these conditions is satisfied then $\int_{\mathbb{R}^d} |\log(f)(x)| \mathrm{d}\pi_0 \leq \mathrm{KL}(\pi_0 \mid \pi_1) + 2/\mathrm{e}$.*

*Proof.* First, note that

$$\int_{\mathbb{R}^d} |\log(f)(x)|\mathbb{1}_{f<1}\mathrm{d}\pi_0(x) \leq \int_{\mathbb{R}^d} |\log(f)(x)f(x)|\mathbb{1}_{f<1}\mathrm{d}\pi_1(x) \leq 1/\mathrm{e} , \tag{21}$$

where we have used that for any $u \in [0,1]$, $|u\log(u)| \leq 1/\mathrm{e}$. We have that (b) implies (c). Using the previous result we have that (c) implies (b). Hence (c) and (b) are equivalent. In addition, it is clear that (b) implies (a). Finally (this is more of a convention), we have that $\mathrm{KL}(\pi_0 \mid \pi_1) = \int_{\mathbb{R}^d} \log(f)(x)\mathbb{1}_{f(x)>1}\mathrm{d}\pi_0(x) + \int_{\mathbb{R}^d} \log(f)(x)\mathbb{1}_{f(x)<1}\mathrm{d}\pi_0(x) < +\infty$. Using (21) this implies (c). Finally, we have

$$\int_{\mathbb{R}^d} |\log(f)(x)| \mathrm{d}\pi_0(x) = \int_{\mathbb{R}^d} \log(f)(x)\mathrm{d}\pi_0(x) - 2 \int_{\mathbb{R}^d} \log(f)(x)\mathbb{1}_{f(x)<1}\mathrm{d}\pi_0(x)$$

$$\leq \mathrm{KL}(\pi_0 \mid \pi_1) + 2/\mathrm{e},$$

which concludes the proof. $\square$

We begin with the following lemma which controls the integral of the potentials uniformly w.r.t. $n \in \mathbb{N}$. It can be seen as the *multi-marginal* counterpart of (Ruschendorf, 1995, Lemma 4.1).

**Lemma 18.** *Assume* **A**4. *There exist* $\{c_i\}_{i \in \mathsf{S}} \in (0, +\infty)^K$ *such that for any function* $f : (\mathbb{R}^d)^{\ell+1} \to \mathbb{R}$ *of the form* $f = \bigoplus_{i \in \mathsf{S}} f_i$, *we have*

$$c_i \|f\|_{\mathrm{L}^1(\pi^\star)} \geq \|f_i\|_{\mathrm{L}^1(\mu_i)}, \quad \forall i \in \mathsf{S} . \tag{22}$$

*For any* $n \in \mathbb{N}^\star$, *we have*

*(a)* $\sum_{i \in \mathsf{S}} \int_{\mathbb{R}^d} \psi_i^n \mathrm{d}\mu_i \leq \mathrm{KL}(\pi^\star \mid \bar{\pi}) < \infty$,

*(b)* $\int_{(\mathbb{R}^d)^{\ell+1}} (\bigoplus_{i \in \mathsf{S}} \psi_i^\star - \bigoplus_{i \in \mathsf{S}} \psi_i^n) \mathrm{d}\pi^\star \leq \mathrm{KL}(\pi^\star \mid \bar{\pi}) < \infty$,

*(c)* $\sup_{n \in \mathbb{N}} \int_{\mathbb{R}^d} |\psi_i^n| \, \mathrm{d}\mu_i < \infty$, $\forall i \in \mathsf{S}$.

*Proof.* First, we have that (22) is a direct consequence of (Kober, 1940, Theorem 1) and **A**4. Let us now prove (a). Using (20), we have

$$
\begin{aligned}
\sum_{m=0}^{Kn} \mathrm{KL}(\pi^m \mid \pi^{m-1}) &= \sum_{\ell=0}^{n-1} \sum_{k=0}^{K-1} \mathrm{KL}(\pi^{\ell K+k+1} \mid \pi^{\ell K+k}) + \mathrm{KL}(\pi^0 \mid \pi^{-1}) \\
&= \sum_{\ell=0}^{n-1} \sum_{i \in \mathsf{S}} \int_{\mathbb{R}^d} (\psi_i^{\ell+1} - \psi_i^\ell) \mathrm{d}\mu_i + \int_{\mathbb{R}^d} (\psi_{i_{K-1}}^0 - \psi_{i_{K-1}}^{-1}) \mathrm{d}\mu_{i_{K-1}} \\
&= \sum_{i \in \mathsf{S}} \sum_{\ell=0}^{n-1} \int_{\mathbb{R}^d} (\psi_i^{\ell+1} - \psi_i^\ell) \mathrm{d}\mu_i + \int_{\mathbb{R}^d} (\psi_{i_{K-1}}^0 - \psi_{i_{K-1}}^{-1}) \mathrm{d}\mu_{i_{K-1}} \\
&= \sum_{i \in \mathsf{S}} \int_{\mathbb{R}^d} (\psi_i^n - \psi_i^0) \mathrm{d}\mu_i + \int_{\mathbb{R}^d} (\psi_{i_{K-1}}^0 - \psi_{i_{K-1}}^{-1}) \mathrm{d}\mu_{i_{K-1}} \\
&= \sum_{i \in \mathsf{S}} \int_{\mathbb{R}^d} \psi_i^n \mathrm{d}\mu_i \leq \mathrm{KL}(\pi^\star \mid \bar{\pi}). ,
\end{aligned}
$$

where the last inequality follows the proof of Proposition 5.

Since the first term in the inequality of (b) is equal to $\mathrm{KL}(\pi^\star \mid \pi^{nK})$, we obtain (b) using that $\mathrm{KL}(\pi^\star \mid \pi^{nK}) \leq \mathrm{KL}(\pi^\star \mid \bar{\pi})$ following the proof of Proposition 5.

Let us now prove (c). Since $\mathrm{KL}(\pi^\star \mid \bar{\pi}) < \infty$, using Proposition 17, we have that $\bigoplus_{i \in \mathsf{S}} \psi_i^\star \in \mathrm{L}^1(\pi^\star)$. From (b) and Proposition 17, we also get that $\bigoplus_{i \in \mathsf{S}} (\psi_i^\star - \psi_i^n) \in \mathrm{L}^1(\pi^\star)$, and thus $\int_{(\mathbb{R}^d)^{\ell+1}} |\bigoplus_{i \in \mathsf{S}} (\psi_i^\star - \psi_i^n)| \, \mathrm{d}\pi^\star \leq C_0$ with $C_0 > 0$. Therefore, we have

$$\int_{(\mathbb{R}^d)^{\ell+1}} \left|\bigoplus_{i \in \mathsf{S}} \psi_i^n\right| \mathrm{d}\pi^\star \leq \int_{(\mathbb{R}^d)^{\ell+1}} \left|\bigoplus_{i \in \mathsf{S}} \psi_i^\star\right| \mathrm{d}\pi^\star + \int_{(\mathbb{R}^d)^{\ell+1}} \left|\bigoplus_{i \in \mathsf{S}} (\psi_i^\star - \psi_i^n)\right| \mathrm{d}\pi^\star \leq 2C_0 .$$

Using (22), we conclude with **A**4 that for any $i \in \mathsf{S}$, we have

$$\int_{\mathbb{R}^d} |\psi_i^n| \, \mathrm{d}\mu_i \leq 2c_i C_0 ,$$

which concludes the proof of (c). $\qquad \square$

The next lemma gives an explicit expression for $\mathrm{KL}(\pi^n \mid \bar{\pi})$. It can be seen as the *multi-marginal* counterpart of (Ruschendorf, 1995, Lemma 4.2).

**Lemma 19.** *For any* $n \in \mathbb{N}$, *with* $k_n = (n-1) \bmod(K)$, *we have*

$$
\begin{aligned}
\mathrm{KL}(\pi^n \mid \bar{\pi}) = \int_{\mathbb{R}^d} \psi_{i_{k_n}}^{q_n+1} \mathrm{d}\mu_{i_{k_n}} &+ \sum_{\ell=0}^{k_n-1} \int_{\mathbb{R}^d} \psi_{i_\ell}^{q_n+1} \exp[\psi_{i_\ell}^{q_n+1} - \psi_{i_\ell}^{q_n+2}] \mathrm{d}\mu_{i_\ell} \\
&+ \sum_{m=k_n+1}^{K-1} \int_{\mathbb{R}^d} \psi_{i_m}^{q_n} \exp[\psi_{i_m}^{q_n} - \psi_{i_m}^{q_n+1}] \mathrm{d}\mu_{i_m} .
\end{aligned}
$$

*Proof.* Let $n \in \mathbb{N}$, with $k_n = (n-1) \bmod(K)$. Using (18), we have

$$\mathrm{KL}(\pi^n \mid \bar{\pi}) = \int_{\mathbb{R}^d} \psi_{i_{k_n}}^{q_n+1} \mathrm{d}\mu_{i_{k_n}} + \sum_{\ell=0}^{k_n-1} \int_{\mathbb{R}^d} \psi_{i_\ell}^{q_n+1} \mathrm{d}\pi_{i_\ell}^n + \sum_{m=k_n+1}^{K-1} \int_{\mathbb{R}^d} \psi_{i_m}^{q_n} \mathrm{d}\pi_{i_m}^n . \tag{23}$$

Consider $m \in \{k_n+1, \ldots, K-1\}$. Let $m_n$ be the closest integer to $n$ such that $m_n > n$ and $m = (m_n - 1) \bmod(K)$. By (19), we have

$$\mathrm{d}\pi^n = \exp\left[\bigoplus_{j=k_n+1}^m \psi_{i_j}^{q_n} - \psi_{i_j}^{q_n+1}\right] \mathrm{d}\pi^{m_n}.$$

Using (19) recursively, we obtain

$$\mathrm{d}\pi_{i_m}^n = \exp[\psi_{i_m}^{q_n} - \psi_{i_m}^{q_n+1}] \mathrm{d}\pi_{i_m}^{m_n}, \tag{24}$$

where we recall that $\pi_{i_m}^{m_n} = \mu_{i_m}$.

Consider now $\ell \in \{0, \ldots, k_n - 1\}$. Let $\ell_n$ be the closest integer to $n$ such that $\ell_n > n$ and $\ell = (\ell_n - 1) \bmod(K)$. By (19), we have

$$\mathrm{d}\pi^n = \exp[\bigoplus_{j=k_n+1}^{K-1} \{\psi_{i_j}^{q_n} - \psi_{i_j}^{q_n+1}\} \bigoplus_{j'=0}^{\ell} \{\psi_{i_{j'}}^{q_n+1} - \psi_{i_{j'}}^{q_n+2}\}] \mathrm{d}\pi^{\ell_n},$$

and using (19) recursively, we obtain

$$\mathrm{d}\pi_{i_\ell}^n = \exp[\psi_{i_\ell}^{q_n+1} - \psi_{i_\ell}^{q_n+2}] \mathrm{d}\pi_{i_\ell}^{\ell_n}, \tag{25}$$

where we recall that $\pi_{i_\ell}^{\ell_n} = \mu_{i_\ell}$. We conclude the proof upon combining (23), (24) and (25). $\qquad\square$

We are now ready to prove a *uniform integrability* result which is the multi-marginal counterpart of (Ruschendorf, 1995, Lemma 4.4). Before stating Lemma 21, we prove the following well-known lemma. We recall that a sequence $(\Psi_n)_{n\in\mathbb{N}}$ such that for any $n \in \mathbb{N}$, $\Psi_n \in \mathrm{L}^1(\mu)$, is *uniformly integrable* w.r.t. $\mu$ if (i) $\sup_{n\in\mathbb{N}} \int_{\mathbb{R}^d} |\Psi_n| \mathrm{d}\mu < +\infty$ and (ii) for any $\varepsilon > 0$, there exists $K > 0$ such that for any $n \in \mathbb{N}$, $\int_{\overline{\mathrm{B}}(0,K)^c} |\Psi_n| \mathrm{d}\mu \leq \varepsilon$.

**Lemma 20.** *Let* $f : \mathbb{R} \to \mathbb{R}$, *convex and non-decreasing on* $[A, +\infty)$ *with* $A > 0$ *and* $\lim_{x\to+\infty} f(x)/x = +\infty$. *Assume that* $\sup_{n\in\mathbb{N}} \int_{\mathbb{R}^d} f(|\Psi_n|) \mathrm{d}\mu < +\infty$. *Then,* $(\Psi_n)_{n\in\mathbb{N}}$ *is uniformly integrable w.r.t.* $\mu$.

*Proof.* Since $f$ is convex, using Jensen's inequality, we get that $\sup_{n\in\mathbb{N}} f(\int_{\mathbb{R}^d} |\Psi_n| \mathrm{d}\mu) < +\infty$ and since $\lim_{x\to+\infty} f(x)/x = +\infty$ we have $\sup_{n\in\mathbb{N}} \int_{\mathbb{R}^d} |\Psi_n| \mathrm{d}\mu < +\infty$. Let $\varepsilon > 0$, there exists $K > 0$ such that for any $x > K$, $x \leq \varepsilon f(x)/B$ with $B = \sup_{n\in\mathbb{N}} \int_{\mathbb{R}^d} f(|\Psi_n|) \mathrm{d}\mu < +\infty$. Therefore, we have for any $n \in \mathbb{N}$

$$\int_{\overline{\mathrm{B}}(0,K)^c} |\Psi_n| \mathrm{d}\mu \leq (\varepsilon/B) \int_{\overline{\mathrm{B}}(0,K)^c} f(|\Psi_n|) \mathrm{d}\mu \leq \varepsilon,$$

which concludes the proof. $\qquad\square$

**Lemma 21.** *Assume* **A**4 *and* **A**5. *Then,* $\{\exp[\bigoplus_{i\in\mathsf{S}} \psi_i^n]\}_{n\in\mathbb{N}}$ *is uniformly integrable w.r.t.* $\bar{\pi}$.

*Proof.* It is enough to show that the sequence $\{f(\exp[\bigoplus_{i\in\mathsf{S}} \psi_i^n])\}_{n\in\mathbb{N}}$ is bounded in $\mathrm{L}^1(\bar{\pi})$, where $f : u \mapsto u \log(u)$ is continuous, convex and such that $\lim_{u\to\infty} f(u)/u = +\infty$, see Lemma 20. Let $n \in \mathbb{N}$. We have

$$\int_{(\mathbb{R}^d)^{\ell+1}} f(\exp[\bigoplus_{i\in\mathsf{S}} \psi_i^n]) \mathrm{d}\bar{\pi} = \mathrm{KL}(\pi^{nK} \mid \bar{\pi})$$

$$= \int_{\mathbb{R}^d} \psi_{i_{K-1}}^n \mathrm{d}\mu_{i_{K-1}} + \sum_{k=0}^{K-2} \int_{\mathbb{R}^d} \psi_{i_k}^n \exp[\psi_{i_k}^n - \psi_{i_k}^{n+1}] \mathrm{d}\mu_{i_k} \qquad \text{(Lemma 19)}$$

$$= \sum_{k=0}^{K-1} \int_{\mathbb{R}^d} \psi_{i_k}^n \mathrm{d}\mu_{i_k} + \sum_{k=0}^{K-2} \int_{\mathbb{R}^d} \psi_{i_k}^n \{\exp[\psi_{i_k}^n - \psi_{i_k}^{n+1}] - 1\} \mathrm{d}\mu_{i_k}$$

$$\leq \mathrm{KL}(\pi^\star \mid \bar{\pi}) + (\bar{c}+1) \sum_{k=0}^{K-2} \int_{\mathbb{R}^d} \psi_{i_k}^n \mathrm{d}\mu_{i_k} \qquad \text{(Lemma 18-(a), \textbf{A}5)}$$

$$\leq \mathrm{KL}(\pi^\star \mid \bar{\pi}) + (\bar{c}+1) \sum_{k=0}^{K-2} \sup_{n\in\mathbb{N}} \int_{\mathbb{R}^d} |\psi_{i_k}^n| \mathrm{d}\mu_{i_k} < \infty . \qquad \text{(Lemma 18-(c))}$$

$$\square$$

With the preliminary results stated above, we are now ready to prove Proposition 6.

*Proof of Proposition 6.* Using **A**4 and **A**5, we have, by Lemma 21, uniform integrability of $\{\exp[\bigoplus_{i\in\mathsf{S}} \psi_i^n]\}_{n\in\mathbb{N}}$ in $\mathrm{L}^1(\bar{\pi})$. Therefore, the sequence $\{\pi^{nK}\}_{n\in\mathbb{N}}$ is relatively compact with respect to the weak topology of $\sigma(\mathrm{L}^1(\bar{\pi}), \mathrm{L}^\infty(\bar{\pi}))$, denoted as the $\tau$-topology. We recall that $\lim_{n\to\infty} \mathrm{KL}(\pi^{nK+1} \mid \pi^{nK}) = 0$. This implies that $\{\pi^{nK+1}\}_{n\in\mathbb{N}}$ is also relatively $\tau$-compact. By trivial recursion, we obtain that the sequences $\{\pi^{nK+k}\}_{n\in\mathbb{N}}$, where $k \in \{2, \ldots, K-1\}$ are also relatively $\tau$-compact. Therefore, $\{\pi^n\}_{n\in\mathbb{N}}$ is relatively $\tau$-compact and $\tau$-sequentially compact.

We consider an increasing function $\Phi : \mathbb{N} \to \mathbb{N}$ such that $\{\pi^m\}_{m\in\Phi(\mathbb{N})}$ is a $\tau$-convergent subsequence, and we denote by $\tilde{\pi}$ its limit for this topology. In particular, $\tilde{\pi} \in \mathscr{P}_\mathsf{S}$ by Proposition 5. We assume without loss of generality that $\Phi(\mathbb{N}) \subset K\mathbb{N}$.

Using the lower semi-continuity of the Kullback-Leibler divergence (Dupuis & Ellis, 2011, Lemma 1.4.3), we get

$$\mathrm{KL}(\tilde{\pi} \mid \bar{\pi}) \le \liminf \mathrm{KL}(\pi^m \mid \bar{\pi}) \le \limsup \mathrm{KL}(\pi^m \mid \bar{\pi}) \ .$$

Consider $k \in \{0, \ldots, K-2\}$. By (19), we have

$$\frac{\mathrm{d}\mu_{i_k}}{\mathrm{d}\pi_{i_k}^{nK+k}} = \frac{\mathrm{d}\pi_{i_k}^{nK+k+1}}{\mathrm{d}\pi_{i_k}^{nK+k}} = \frac{\mathrm{d}\pi^{nK+k+1}}{\mathrm{d}\pi^{nK+k}} = \exp[\psi_{i_k}^{n+1} - \psi_{i_k}^n] \ ,$$

and thus,

$$\|\mu_{i_k} - \pi_{i_k}^{nK+k}\|_{\mathrm{TV}} = (1/2) \int_{\mathbb{R}^d} \left| \mathrm{d}\pi_{i_k}^{nK+k}/\mathrm{d}\mu_{i_k} - 1 \right| \mathrm{d}\mu_{i_k} = (1/2) \int_{\mathbb{R}^d} \left| \exp[\psi_{i_k}^n - \psi_{i_k}^{n+1}] - 1 \right| \mathrm{d}\mu_{i_k} \ .$$

With Proposition 5, we obtain that $\{\exp[\psi_{i_k}^n - \psi_{i_k}^{n+1}]\}_{n \in \mathbb{N}}$ converges to 1 in $\mathrm{L}^1(\mu_{i_k})$. In addition using the uniform integrability of $\{\psi_{i_k}^n\}_{n \in \mathbb{N}}$ and **A**5, we get

$$\limsup_{n \to +\infty} \int_{\mathbb{R}^d} \psi_{i_k}^n \exp[\psi_{i_k}^n - \psi_{i_k}^{n+1}] \mathrm{d}\mu_{i_k} = \limsup_{n \to +\infty} \int_{\mathbb{R}^d} \psi_{i_k}^n \mathrm{d}\mu_{i_k} \ .$$

We denote $m = K\ell$. Since $\mathrm{KL}(\pi^m \mid \bar{\pi}) = \int_{\mathbb{R}^d} \psi_{i_{K-1}}^\ell \mathrm{d}\mu_{i_{K-1}} + \sum_{k=0}^{K-2} \int_{\mathbb{R}^d} \psi_{i_k}^\ell \exp[\psi_{i_k}^\ell - \psi_{i_k}^{\ell+1}] \mathrm{d}\mu_{i_k}$ by Lemma 19, we finally have

$$\mathrm{KL}(\tilde{\pi} \mid \bar{\pi}) \le \limsup\{\textstyle\sum_{k=0}^{K-1} \int_{\mathbb{R}^d} \psi_{i_k}^\ell \mathrm{d}\mu_{i_k}\} \le \mathrm{KL}(\pi^\star \mid \bar{\pi})$$

where the last inequality comes from Lemma 18.

Since $\tilde{\pi}_i = \mu_i$ for any $i \in \mathsf{S}$, using Proposition 5, we have $\tilde{\pi} = \pi^\star$ by uniqueness of $\pi^\star$. Hence, $\pi^\star$ is the only limit point of $\{\pi^n\}_{n \in \mathbb{N}}$ in the $\tau$-topology. In particular, $\mathrm{KL}(\pi^n \mid \bar{\pi}) \to \mathrm{KL}(\pi^\star \mid \bar{\pi})$. Since $\mathscr{P}_\mathsf{S}$ is convex, this last result implies $\|\pi^\star - \pi^n\|_{\mathrm{TV}} \to 0$, see the proof of Theorem 2.1 in Csiszár (1975). $\qquad\square$

We finish this section by highlighting that **A**5 is stronger than (Ruschendorf, 1995, B1). A natural extension of the latter assumption would consist of having a guarantee on the $(K-1)$ first potentials given by (17), as presented below.

**A6.** *There exist $0 < \underline{c} < \bar{c}$ such that for any $k \in \{0, \ldots, K-2\}$, we have $\underline{c} \le \exp(-\psi_{i_k}^1) \le \bar{c}$.*

Under **A**6, (Ruschendorf, 1995, Lemma 4.3) can be adapted as written below.

**Lemma 22.** *Assume **A**6. Then, for any $n \in \mathbb{N}^*$*

*(a) for any $k \in \{0, \ldots, K-2\}$, there exists $\alpha_{n,k} \in \mathbb{N}$ such that*

$$\underline{c} \cdot (\underline{c}/\bar{c})^{\alpha_{n,k}(K-2)} \le \exp[\psi_{i_k}^{n-1} - \psi_{i_k}^n] \le \bar{c} \cdot (\bar{c}/\underline{c})^{\alpha_{n,k}(K-2)}$$

*(b) there exists $\alpha_{n,K-1} \in \mathbb{N}$ such that*

$$1/\bar{c}^{K-1} \cdot (\underline{c}/\bar{c})^{\alpha_{n,K-1}(K-2)} \le \exp[\psi_{i_{K-1}}^{n-1} - \psi_{i_{K-1}}^n] \le 1/\underline{c}^{K-1} \cdot (\bar{c}/\underline{c})^{\alpha_{n,K-1}(K-2)}$$

*where $\{\alpha_{n,k}\}_{n \in \mathbb{N}^*, k \in \{0,\ldots,K-1\}}$ is a strictly increasing sequence that can be explicitly defined.*

*Proof.* We prove the result by recursion on $n \in \mathbb{N}^*$.

Take $n = 1$. Let $k \in \{0, \ldots, K-2\}$. We define $\alpha_{1,k} = 0$ and directly obtain (a) by **A**5 since $\psi_{i_k}^0 = 0$. Let us prove (b). We have by (17)

$$
\begin{aligned}
\exp[\psi_{i_{K-1}}^0 - \psi_{i_{K-1}}^1] &= \frac{\int_{(\mathbb{R}^d)^\ell} \exp[\bigoplus_{k=0}^{K-2} \psi_{i_k}^1] h \mathrm{d}\nu_{-i_{K-1}}}{\int_{(\mathbb{R}^d)^\ell} \exp[\bigoplus_{k=0}^{K-2} \psi_{i_k}^0] h \mathrm{d}\nu_{-i_{K-1}}} \\
&= \frac{\int_{(\mathbb{R}^d)^\ell} \exp[\bigoplus_{k=0}^{K-2} \{\psi_{i_k}^1 - \psi_{i_k}^0\} + \bigoplus_{k=0}^{K-2} \psi_{i_k}^0] h \mathrm{d}\nu_{-i_{K-1}}}{\int_{(\mathbb{R}^d)^\ell} \exp[\bigoplus_{k=0}^{K-2} \psi_{i_k}^0] h \mathrm{d}\nu_{-i_{K-1}}} \ .
\end{aligned}
$$

Using (a) at rank $n = 1$, we have

$$1/\bar{c}^{K-1} \le \exp[\textstyle\bigoplus_{k=0}^{K-2} \{\psi_{i_k}^1 - \psi_{i_k}^0\}] \le 1/\underline{c}^{K-1} \ ,$$

and therefore, we obtain (b) by taking $\alpha_{1,K-1} = 0$. Let us assume that the result is verified for some $n \in \mathbb{N}^*$. We have

$$\exp[\psi_{i_0}^n - \psi_{i_0}^{n+1}] = \frac{\int \exp[\bigoplus_{k=1}^{K-1} \psi_{i_k}^n] h \mathrm{d}\nu_{-i_0}}{\int \exp[\bigoplus_{k=1}^{K-1} \psi_{i_k}^{n-1}] h \mathrm{d}\nu_{-i_0}}$$

$$= \frac{\int \exp[\bigoplus_{k=1}^{K-2} \{\psi_{i_k}^n - \psi_{i_k}^{n-1}\} \oplus \{\psi_{i_{K-1}}^n - \psi_{i_{K-1}}^{n-1}\} + \bigoplus_{k=1}^{K-1} \psi_{i_k}^{n-1}] h \mathrm{d}\nu_{-i_0}}{\int \exp[\bigoplus_{k=1}^{K-1} \psi_{i_k}^{n-1}] h \mathrm{d}\nu_{-i_0}}$$

Using (a) and (b) at rank $n$, we have

$$1/\bar{c}^{K-2} \cdot (\underline{c}/\bar{c})^{(K-2)\sum_{k=1}^{K-2} \alpha_{n,k}} \le \exp[\bigoplus_{k=1}^{K-2} \{\psi_{i_k}^n - \psi_{i_k}^{n-1}\}]$$
$$\le 1/\underline{c}^{K-2} \cdot (\bar{c}/\underline{c})^{(K-2)\sum_{k=1}^{K-2} \alpha_{n,k}} ,$$
$$\underline{c}^{K-1} \cdot (\underline{c}/\bar{c})^{\alpha_{n,K-1}(K-2)} \le \exp[\psi_{i_{K-1}}^n - \psi_{i_{K-1}}^{n-1}] \le \bar{c}^{K-1} \cdot (\bar{c}/\underline{c})^{\alpha_{n,K-1}(K-2)} .$$

Therefore, we obtain

$$\underline{c} \cdot (\underline{c}/\bar{c})^{(K-2)\sum_{k=1}^{K-1} \alpha_{n,k}} \le \exp[\bigoplus_{k=1}^{K-2} \{\psi_{i_k}^n - \psi_{i_k}^{n-1}\} \oplus \{\psi_{i_{K-1}}^n - \psi_{i_{K-1}}^{n-1}\}]$$
$$\le \bar{c} \cdot (\bar{c}/\underline{c})^{(K-2)\sum_{k=1}^{K-1} \alpha_{n,k}} ,$$
$$\underline{c} \cdot (\underline{c}/\bar{c})^{(K-2)\sum_{k=1}^{K-1} \alpha_{n,k}} \le \exp[\psi_{i_0}^n - \psi_{i_0}^{n+1}] \le \bar{c} \cdot (\bar{c}/\underline{c})^{(K-2)\sum_{k=1}^{K-1} \alpha_{n,k}} .$$

Now, we define $\alpha_{n+1,0} = \sum_{k=1}^{K-1} \alpha_{n,k}$ to obtain (a) for $k = 0$. Consider now $k \in \{1, \ldots, K-2\}$. Following the same steps as above, we recursively define

$$\alpha_{n+1,k} = \sum_{j=0}^{k-1} \alpha_{n+1,j} + \sum_{j'=k+1}^{K-1} \alpha_{n,j'} ,$$

which gives (a) at rank $n + 1$. Let us now prove (b) at rank $n + 1$. We have

$$\exp[\psi_{i_{K-1}}^n - \psi_{i_{K-1}}^{n+1}] = \frac{\int \exp[\bigoplus_{k=0}^{K-2} \psi_{i_k}^{n+1}] h \mathrm{d}\nu_{-i_{K-1}}}{\int \exp[\bigoplus_{k=0}^{K-2} \psi_{i_k}^n] h \mathrm{d}\nu_{-i_{K-1}}}$$

$$= \frac{\int \exp[\bigoplus_{k=0}^{K-2} \{\psi_{i_k}^{n+1} - \psi_{i_k}^n\} + \bigoplus_{k=0}^{K-2} \psi_{i_k}^n] h \mathrm{d}\nu_{-i_k}}{\int \exp[\bigoplus_{k=0}^{K-2} \psi_{i_k}^n] h \mathrm{d}\nu_{-i_{K-1}}} .$$

Using (a) at rank $n + 1$, we obtain

$$1/\bar{c}^{K-1} \cdot (\underline{c}/\bar{c})^{(K-2)\sum_{k=0}^{K-2} \alpha_{n+1,k}} \le \exp[\bigoplus_{k=0}^{K-2} \{\psi_{i_k}^{n+1} - \psi_{i_k}^n\}]$$
$$\le 1/\underline{c}^{K-1} \cdot (\bar{c}/\underline{c})^{(K-2)\sum_{k=0}^{K-2} \alpha_{n+1,k}} .$$

Therefore, by taking $\alpha_{n+1,K-1} = \sum_{k=0}^{K-2} \alpha_{n+1,k}$, we obtain (b), which concludes the proof. $\square$

Unfortunately, Lemma 22 only yields non-vacuous bounds in the case $K = 2$. Indeed, when $K > 2$, the sequence $\{\alpha_{n,k}\}_{n \in \mathbb{N}^*, k \in \{0, \ldots, K-1\}}$ leads to increase the bounds on the quantities $\exp[\psi_{i_k}^{n-1} - \psi_{i_k}^n]$, which motivates the use of **A**5.

### D.3 Proof of Section 5

For the rest of this section, we consider the multi-marginal Schrödinger bridge problem given by (TreeSB) and establish in Proposition 24 the correspondence with the regularized Wasserstein propagation problem presented in Solomon et al. (2014, 2015). We first state a technical result.

**Lemma 23.** *Let $\varepsilon > 0$. Assume that $\pi^0$ is given by (2), where $r \in \mathsf{V}$ is chosen arbitrarily. Then, for any $\pi \in \mathscr{P}_{\mathsf{T}_r}$, we have*

$$\varepsilon \mathrm{KL}(\pi \mid \pi^0) = \sum_{(v,v') \in \mathsf{E}_r} \{w_{v,v'} \mathbb{E}_{\pi_{v,v'}}[\|X_v - X_{v'}\|^2] - \varepsilon \mathrm{H}(\pi_{v,v'})\}$$
$$+ \varepsilon \sum_{v \in \mathsf{V}} \mathrm{card}(\mathsf{C}_v) \mathrm{H}(\pi_v) + \varepsilon \mathrm{KL}(\pi_r \mid \pi_r^0) ,$$

*where we recall that $\mathsf{C}_v = \{v' \in \mathsf{V} : (v, v') \in \mathsf{E}_r\}$.*

*Proof.* Since $\pi, \pi^0 \in \mathscr{P}_{\mathsf{T}_r}$, we obtain the following decomposition

$$
\begin{aligned}
\mathrm{KL}&(\pi \mid \pi^0) \\
&= \mathrm{KL}(\pi_r \textstyle\prod_{(v,v') \in \mathsf{E}_r} \pi_{v'|v} \mid \pi_r^0 \prod_{(v,v') \in \mathsf{E}_r} \pi_{v'|v}^0) \\
&= \mathrm{KL}(\pi_r \mid \pi_r^0) + \textstyle\sum_{(v,v') \in \mathsf{E}_r} \int_{\mathbb{R}^d} \mathrm{KL}(\pi_{v'|v}(\cdot|x_v) \mid \pi_{v'|v}^0(\cdot|x_v)) \mathrm{d}\pi_v(x_v) \\
&= \mathrm{KL}(\pi_r \mid \pi_r^0) - \textstyle\sum_{(v,v') \in \mathsf{E}_r} \int_{\mathbb{R}^d \times \mathbb{R}^d} \log \pi_{v'|v}^0 \mathrm{d}\pi_{v,v'} - \sum_{(v,v') \in \mathsf{E}_r} \int_{\mathbb{R}^d} \mathrm{H}(\pi_{v'|v}(\cdot|x_v)) \mathrm{d}\pi_v(x_v) \, .
\end{aligned}
$$

We finally obtain the result by using the definition of $\pi^0$ and noticing that $\int_{\mathbb{R}^d} \mathrm{H}(\pi_{v'|v}(\cdot|x_v)) \mathrm{d}\pi_v(x_v) = \mathrm{H}(\pi_{v,v'}) - \mathrm{H}(\pi_v)$ for any $(v,v') \in \mathsf{E}_r$. $\qquad\square$

**Proposition 24.** *Let $\varepsilon > 0$ and $\mu_0 \in \mathscr{P}$ such that $\mu_0 \ll$ Leb. Assume that $\pi^0$ is given by (2), where $r \in \mathsf{V}$ is chosen arbitrarily, and that $\varphi_r = \mathrm{d}\mu_0/\mathrm{dLeb}$. Also assume **A**2. Then, the set of marginals of the solution to (TreeSB) is exactly the solution to the entropic-regularized Wasserstein Propagation problem (Solomon et al., 2014, 2015) defined by*

$$
\arg\min\{\textstyle\sum_{(v,v') \in \mathsf{E}_r} w_{v,v'} W_{2,\varepsilon/w_{v,v'}}^2(\nu_v, \nu_{v'}) + \varepsilon \sum_{v \in \mathsf{V}} \mathrm{card}(\mathrm{C}_v)\mathrm{H}(\nu_v) + \varepsilon\mathrm{KL}(\nu_r \mid \mu_0) : \quad \text{(WP)}
$$
$$
\{\nu_v\}_{v \in \mathsf{V}} \in \mathscr{P}^{\ell+1}, \; \nu_i = \mu_i, \forall i \in \mathsf{S} \} \, ,
$$

*where we recall that $\mathrm{C}_v = \{v' \in \mathsf{V} : (v,v') \in \mathsf{E}_r\}$.*

*Proof.* Assume that $\pi^0$ is given by (2), where $r \in \mathsf{V}$ is chosen arbitrarily, and that $\varphi_r = \mathrm{d}\mu_0/\mathrm{dLeb}$. In particular, we have $\pi_r^0 = \mu_0$. Moreover, it is clear that $\pi^0$ verifies **A**1, and **A**3 by Proposition 16.

Let $\{\nu_v\}_{v \in \mathsf{V}} \in \mathscr{P}^{\ell+1}$ and $\{\nu^{(v,v')}\}_{(v,v') \in \mathsf{E}_r} \in (\mathscr{P}^{(2)})^{|\mathsf{E}_r|}$. We define

$$
F(\{\nu_v\}) = \textstyle\sum_{(v,v') \in \mathsf{E}_r} w_{v,v'} W_{2,\varepsilon/w_{v,v'}}^2(\nu_v, \nu_{v'}) + \varepsilon \sum_{v \in \mathsf{V}} \mathrm{card}(\mathrm{C}_v)\mathrm{H}(\nu_v) + \varepsilon\mathrm{KL}(\nu_r \mid \mu_0) \, ,
$$
$$
\begin{aligned}
G(\nu_r, \{\nu^{(v,v')}\}) = &\textstyle\sum_{(v,v') \in \mathsf{E}_r} \{w_{v,v'} \mathbb{E}_{\nu^{(v,v')}}[\|X_v - X_{v'}\|^2] - \varepsilon\mathrm{H}(\nu^{(v,v')})\} \\
&+ \varepsilon \textstyle\sum_{(v,v') \in \mathsf{E}_r} \mathrm{H}(\nu_v^{(v,v')}) + \varepsilon\mathrm{KL}(\nu_r \mid \mu_0) \, .
\end{aligned}
$$

By definition of the regularized Wasserstein distance given in (3), we have for any $\{\nu_v\}_{v \in \mathsf{V}} \in \mathscr{P}^{\ell+1}$

$$
F(\{\nu_v\}) = \min\{G(\nu_r, \{\nu^{(v,v')}\}) : \nu^{(v,v')} \in \mathscr{P}^{(2)}, \nu_v^{(v,v')} = \nu_v, \nu_{v'}^{(v,v')} = \nu_{v'}, \forall(v,v') \in \mathsf{E}_r\} \, . \tag{26}
$$

In particular, we have $F(\{\pi_v\}) \leq G(\pi_r, \{\pi_{v,v'}\})$ for any $\pi \in \mathscr{P}^{(\ell+1)}$. We now prove the result of Proposition 24 in two steps denoted by **Step 1** and **Step 2**.

**Step 1.** Let us not assume **A**2 for now. In this case, we prove in **Step 1.a** and **Step 1.b** that solving (WP) is equivalent to solving a modified version of (TreeSB) given by

$$
\pi^\star = \arg\min\{\mathrm{KL}(\pi|\pi^0) : \pi \in \mathscr{P}_{\mathsf{T}_r}, \pi_i = \mu_i, \forall i \in \mathsf{S}\} \, . \tag{$\mathsf{T}_r$-TreeSB}
$$

Remark that any solution to ($\mathsf{T}_r$-TreeSB) is a solution to (TreeSB), but the converse result may not be true.

**Step 1.a: (WP) $\implies$ ($\mathsf{T}_r$-TreeSB).** Consider a solution $\{\nu_v^\star\}_{v \in \mathsf{V}}$ to (WP). For any $(v,v') \in \mathsf{E}_r$, $W_{2,\varepsilon/w_{v,v'}}^2(\nu_v^\star, \nu_{v'}^\star)$ is well defined and thus, there exists $\nu^{(v,v')} \in \Pi(\nu_v^\star, \nu_{v'}^\star)$ such that

$$
\nu^{(v,v')} \in \arg\min\{\mathbb{E}_\pi[\|X_v - X_{v'}\|^2] - (\varepsilon/w_{v,v'})\mathrm{H}(\pi) : \pi \in \Pi(\nu_v^\star, \nu_{v'}^\star)\} \, . \tag{27}
$$

Using the gluing lemma, we build the probability measure $\pi^\star = \nu_r^\star \prod_{(v,v') \in \mathsf{E}_r} \nu_{v'|v}^{(v,v')}$ such that (i) $\pi^\star \in \mathscr{P}_{\mathsf{T}_r}$, and (ii) $\pi_{v,v'}^\star$ and $\nu^{(v,v')}$ have the same distribution for any $(v,v') \in \mathsf{E}_r$. In particular, we have $\pi_i^\star = \mu_i$ for any $i \in \mathsf{S}$.

Let us show now that $\pi^\star$ is a solution to ($\mathsf{T}_r$-TreeSB). Let $\pi \in \mathscr{P}_{\mathsf{T}_r}$ such that $\pi_i = \mu_i$ for any $i \in \mathsf{S}$. We have

$$
\begin{aligned}
\epsilon\mathrm{KL}(\pi \mid \pi^0) &= G(\pi_r, \{\pi_{v,v'}\}) && \text{(Lemma 23)}\\
&\geq F(\{\pi_v\}) &&\\
&\geq F(\{\nu_v^\star\}) && \text{(definition of } \nu^\star)\\
&= G(\nu_r^\star, \{\nu^{(v,v')}\}) && \text{(see (27))}\\
&= G(\pi_r^\star, \{\pi_{(v,v')}^\star\}) && \text{(definition of } \pi^\star)\\
&= \epsilon\mathrm{KL}(\pi^\star \mid \pi^0) \,. && \text{(Lemma 23)}
\end{aligned}
$$

Therefore, $\pi^\star$ is a solution to ($\mathsf{T}_r$-TreeSB).

**Step 1.b: ($\mathsf{T}_r$-TreeSB) $\Longrightarrow$ (WP).** Consider now a solution $\pi^\star$ to ($\mathsf{T}_r$-TreeSB). Since $\pi^\star \in \mathscr{P}_{\mathsf{T}_r}$, we have $\pi^\star = \pi_r^\star \prod_{(v,v')\in\mathsf{E}_r} \pi_{v'|v}^\star$ and $\pi_i^\star = \mu_i$ for any $i \in \mathsf{S}$.

Let us show that $\{\pi_v^\star\}_{v\in\mathsf{V}}$ is a solution to (WP). Let $\{\nu_v\}_{v\in\mathsf{V}} \in \mathscr{P}^{\ell+1}$ such that $\nu_i = \mu_i$ for any $i \in \mathsf{S}$.

Let $\{\nu^{(v,v')}\}_{(v,v')\in\mathsf{E}_r}$ be a family of probability measures such that $\nu^{(v,v')} \in \mathscr{P}^{(2)}, \nu_v^{(v,v')} = \nu_v, \nu_{v'}^{(v,v')} = \nu_{v'}$ for any $(v, v') \in \mathsf{E}_r$.

Using the gluying lemma, we build the probability measure $\pi = \nu_r \prod_{(v,v')\in\mathsf{E}_r} \nu_{v'|v}^{(v,v')}$, such that (i) $\pi \in \mathscr{P}_{\mathsf{T}_r}$ and (ii) $\pi_{v,v'}$ and $\nu^{(v,v')}$ have the same distribution for any $(v, v') \in \mathsf{E}_r$. We have

$$
\begin{aligned}
\varepsilon\mathrm{KL}(\pi \mid \pi^0) &= G(\pi_r, \{\pi_{(v,v')}\}) && \text{(Lemma 23)}\\
&= G(\nu_r, \{\nu^{(v,v')}\}) && \text{(definition of } \pi)\\
&\geq \varepsilon\mathrm{KL}(\pi^\star \mid \pi^0) && \text{(definition of } \pi^\star)\\
&= G(\pi_r^\star, \{\pi_{(v,v')}^\star\}) \,. && \text{(Lemma 23)}
\end{aligned}
$$

By taking the infimum in the previous inequality over the families $\{\nu^{(v,v')}\}_{(v,v')\in\mathsf{E}_r}$, we obtain by (26) that

$$
F(\{\nu_v\}) \geq G(\pi_r^\star, \{\pi_{(v,v')}^\star\}) \geq F(\{\pi_v^\star\}),
$$

and therefore, $\{\pi_v^\star\}_{v\in\mathsf{V}}$ is a solution to (WP).

**Step 2.** We now assume A2. By Proposition 3, there exists a unique solution $\pi^\star \in \mathscr{P}^{(\ell+1)}$ to (TreeSB) such that we $\pi^0$-a.s. have $(\mathrm{d}\pi^\star/\mathrm{d}\pi^0) = \exp[\bigoplus_{i\in\mathsf{S}} \psi_i^\star]$, where $\{\psi_i^\star\}_{i\in\mathsf{S}}$ are measurable potentials with $\psi^\star : \mathbb{R}^d \to \mathbb{R}$. Since $\pi^0 \in \mathscr{P}_{\mathsf{T}_r}$, we also have $\pi^\star \in \mathscr{P}_{\mathsf{T}_r}$, *i.e.*, the potentials $\{\psi_i^\star\}_{i\in\mathsf{S}}$ do not modify the Markovian nature of $\pi^0$. Therefore, $\pi^\star$ is also the unique solution to ($\mathsf{T}_r$-TreeSB). Using the equivalence between ($\mathsf{T}_r$-TreeSB) and (WP) established in **Step 1**, we finally obtain the result of Proposition 24. $\qquad\square$

In particular, Proposition 7 directly derives from Proposition 24 by taking $r = i_{K-1}$ and $\mu_0 = \mu_{i_{K-1}}$.

### D.4 Comparison with Haasler et al. (2021)

In their work, Haasler et al. (2021) study the *static* and *discrete-state* counterpart of our approach. Given a state space $\mathsf{X}$ such that $|\mathsf{X}| = n + 1$ with $n \in \mathbb{N}$, they establish a correspondence between multi-marginal EOT with a general tree-based cost and discrete-time multi-marginal static Schrödinger bridge, and provide an efficient method to solve these problems. In this section, we provide details on their framework and give a precise comparison between our theory and their results.

To be coherent with the setting of Haasler et al. (2021), we adapt here some of our notation. Let us define $\mathsf{Z}^{(q)} = \mathbb{R}_+^{(n+1)^q}$. For any $q \in \mathbb{N}^\star$, the set of probability measures on $\mathsf{X}^q$ is defined as $\mathscr{P}^{(q)} = \{M \in \mathsf{Z}^{(q)} : \langle M, \mathbf{1} \rangle = 1\}$. We denote $\mathscr{P} = \mathscr{P}^{(1)}$. For any tensors $M, P \in \mathsf{Z}^{(q)}$, the Kullback-Leibler divergence between $M$ and $P$ is defined as $\mathrm{KL}(M \mid P) = \langle M \log(M/P) - M + P, \mathbf{1} \rangle$ and the

entropy of $M$ is defined as $\mathrm{H}(M) = -\mathrm{KL}(M \mid \mathbf{1})$, where the operations are meant componentwise. In the rest of the section, we consider an undirected tree $\mathsf{T} = (\mathsf{V}, \mathsf{E})$ with $|\mathsf{V}| = \ell + 1$ such that $\mathsf{V}$ may be identified with $\{0, \ldots, \ell\}$.

**Details on the results of Haasler et al. (2021).** In their paper, the authors consider a cost tensor $C \in \mathsf{Z}^{(\ell+1)}$ that factorizes along $\mathsf{T}$, *i.e.*, for any $\{j_0, \ldots, j_\ell\}$ with for any $i \in \{0, \ldots, \ell\}$, $j_i \in \{0, \ldots, n\}$, we have

$$C_{j_0, \ldots, j_\ell} = \sum_{(v,v') \in \mathsf{E}} C_{j_v, j_{v'}}^{\{v,v'\}},$$

where $C^{\{v,v'\}} \in \mathsf{Z}^{(2)}$ is a cost matrix for transportation between the marginals at vertices $v$ and $v'$, see (Haasler et al., 2021, Eq. (3.1)). In particular, this cost can be seen as the discrete counterpart of the tree-based cost introduced in (1) in the quadratic setting.

Given a subset $\mathsf{S} \subset \mathsf{V}$ with $|\mathsf{S}| = K$ and a set of marginals $\{\mu_i\}_{i \in \mathsf{S}} \in \mathscr{P}^K$, Haasler et al. (2021) study the EOT problem associated to $\mathsf{T}$, see (Haasler et al., 2021, Eq. (2.4)), which is given by

$$\mathrm{argmin}\{\langle C, M \rangle - \varepsilon \mathrm{H}(M) : M \in \mathscr{P}^{(\ell+1)}, \mathrm{proj}_i(M) = \mu_i, \forall i \in \mathsf{S}\} . \qquad \text{(discrete-EmOT)}$$

This problem may be solved with Sinkhorn algorithm (Cuturi, 2013; Knight, 2008; Sinkhorn & Knopp, 1967), for which the authors provide an efficient implementation adapted to the tree-based setting, see (Haasler et al., 2021, Algorithm 3.1). Moreover, they state the convergence of their method in (Haasler et al., 2021, Theorem 3.5), as a direct consequence of the results presented in Luo & Tseng (1992).

In (Haasler et al., 2021, Section 4.2), it is assumed that $\mathsf{S}$ corresponds to the set of the leaves of $\mathsf{T}$, as we do, and it is shown an equivalence between (discrete-EmOT) and the discrete-state static SB problem stated in (Haasler et al., 2021, Eq 4.2), which is given by

$$\mathrm{argmin}\{\textstyle\sum_{(v,v') \in \mathsf{E}_r} \mathrm{KL}(M^{(v,v')} \mid \mathrm{diag}(\nu_v) A^{(v,v')}) : \qquad \text{(discrete-TreeSB)}$$

$$M^{(v,v')} \in \mathscr{P}^{(2)}, \{\nu_v\}_{v \in \mathsf{V}} \in \mathscr{P}^{\ell+1}, M^{(v,v')}\mathbf{1} = \nu_v, {M^{(v,v')}}^\top \mathbf{1} = \nu_{v'}, \ \nu_i = \mu_i, \forall i \in \mathsf{S}\} ,$$

where $\mathsf{T}_r = (\mathsf{V}, \mathsf{E}_r)$ is the directed version of $\mathsf{T}$ rooted in an arbitrary vertex $r \in \mathsf{S}$, and $A^{(v,v')} = \exp(-C^{(v,v')}/\varepsilon) \in \mathsf{Z}^{(2)}$ for any $(v, v') \in \mathsf{E}_r$. Remark that $A^{(v,v')}$ may not necessarily be a transition probability matrix.

Finally, Haasler et al. (2021) provide two main numerical experiments. In (Haasler et al., 2021, Section 5.2), they consider a tree with 15 vertices, 14 edges and 8 leaves, combined to the state-space $\mathsf{X} = \{0, 1\}^{50 \times 50}$, and solve the corresponding (discrete-EmOT) problem for the quadratic cost. In (Haasler et al., 2021, Section 6), they apply their methodology to estimate ensemble flows on a hidden Markov chain. Given $\tau \in \mathbb{N}^*$, they consider a tree $\mathsf{T}$ with $\tau$ internal vertices (modeling the distribution of $N$ agents at time $t \in \{1, \ldots, \tau\}$), that are linearly linked, and such that each of these vertices is independently linked to $S$ leaves of $\mathsf{T}$ (modeling observations at time $t \in \{1, \ldots, \tau\}$). In this setting, the state space is given by $\mathsf{X} = \{1, \ldots, 100\}^N$. They solve the formulation (discrete-TreeSB) where the reference measure is chosen as a random walk.

**Comparison with our results.** We now establish remarks on the main differences between our methodology and the work of Haasler et al. (2021).

First of all, the continuous state-space counterpart of (discrete-TreeSB) is given by

$$\mathrm{argmin}\{\mathrm{KL}(\pi \mid \pi^0) : \pi \in \mathscr{P}_{\mathsf{T}_r}, \pi_i = \mu_i, \forall i \in \mathsf{S}\} , \qquad (28)$$

where $\pi^0$ is a reference measure which factorizes along $\mathsf{T}_r$. In this case, $\pi_{v,v'}$, $\pi_v$ and $\pi^0_{v'|v}$ in (28) respectively correspond to the continuous version of $M^{(v,v')}$, $\nu_v$ and $A^{(v,v')}$ in (discrete-TreeSB). In contrast, our formulation of the multi-marginal Tree Schrödinger Bridge problem given in (TreeSB) is a minimization problem over all probability measures $\pi \in \mathscr{P}^{(\ell+1)}$, and is not restricted to the distributions that admit a Markovian factorization along $\mathsf{T}$ as in (28). Hence, our framework may be considered more general. Remark that under **A**1, **A**2 and **A**3, Proposition 3 states that (TreeSB) admits a unique solution $\pi^\star \ll \pi^0$ such that $(\mathrm{d}\pi^\star/\mathrm{d}\pi^0)$ can be written with potentials. Then, $\pi^\star \in \mathscr{P}_{\mathsf{T}_r}$ since $\pi^0 \in \mathscr{P}_{\mathsf{T}_r}$, and (TreeSB) is then equivalent to (28).

Furthermore, (EmOT) is more general than the continuous version of (discrete-EmOT), which we can recover by taking any measure $\nu$ of the form $(\mathrm{d}\nu/\mathrm{dLeb}) = \exp[\bigoplus_{i \in \mathsf{S}} \varphi_i]$ in (EmOT), where $\{\varphi_i\}_{i \in \mathsf{S}}$ is a family of potentials such that $\left|\int_{\mathbb{R}^d} \varphi_i \mathrm{d}\mu_i\right| < \infty$ for any $i \in \mathsf{S}$. As a consequence, our setting allows us to choose the root $r \in \mathsf{V}\backslash\mathsf{S}$ for the SB problem, whereas Haasler et al. (2021) only consider the case where $r \in \mathsf{S}$. In the latter case, we establish in Appendix E that $r$ can be chosen arbitrarily, as stated by (Haasler et al., 2021, Corollary 4.3).

Finally, TreeDSB deeply differs from the framework of Haasler et al. (2021) due its *dynamic* nature. Although we solve the same tree-based static SB problem (up to continuous/discrete state-space consideration), our approach consists in computing dynamic iterates (*i.e.*, path measures) using diffusion-based methods instead of static iterates (*i.e.*, distributions) using Sinkhorn algorithm. This paradigm is at the core of the DSB (De Bortoli et al., 2021) methodology, and offers an efficient approach to tackle high-dimensional settings, where Sinkhorn algorithm would fail.

Here, we present some advantages of the method proposed by Haasler et al. (2021) compared to ours. First, Haasler et al. (2021) may choose any kind of tree-based cost in practice, while our methodology only holds for the quadratic cost. This limitation is shared with all approaches based on the DSB (De Bortoli et al., 2021) methodology. Indeed, since the cost is determined by the reference path measure, we often choose quadratic costs associated with Brownian motions or Ornstein-Uhlenbeck processes. Moreover, Haasler et al. (2021) may consider various inhomogeneous (discrete) state spaces for the vertices of $\mathsf{T}$, as presented in their numerical experiments. In our case, this approach is not compatible with our diffusion-based method. Finally, unlike Haasler et al. (2021), our method is not scalable with the number of vertices or edges in $\mathsf{T}$ due to computational limits. This limitation is common to all multi-marginal approaches which rely on neural networks to parameterize the potential and/or the distributions of the multi-marginal OT method, see Li et al. (2020); Fan et al. (2020); Korotin et al. (2022, 2021) for instance.

# E  Further results on TreeSB

**Choice of the root $r$ in (TreeSB).**  We recall that the reference measure $\pi^0$ considered in (TreeSB), which is defined in (2), verifies $\pi^0 \in \mathscr{P}_{\mathsf{T}_r}$ for some fixed root $r \in \mathsf{V}$ and $\pi^0_r \ll \mathrm{Leb}$ with density $\varphi_r$. Moreover, we have $\pi^0_{v'|v}(\cdot \mid x_v) = \mathrm{N}(x_v, \varepsilon/(2w_{v,v'})\mathrm{I}_d)$ for any $(v,v') \in \mathsf{E}_r$, and thus, $\pi^0$ is entirely determined by the choice of the root $r$ and the density on the corresponding vertex $\varphi_r$.

As presented in Appendix D.1, we recall that (TreeSB) is equivalent to any multi-marginal Tree-SB problem with a reference measure $\bar{\pi}^0$ given by (15), *i.e.*, $\bar{\pi}^0$ writes as $(\mathrm{d}\bar{\pi}^0/\mathrm{d}\pi^0) = \exp[\bigoplus_{i \in \mathsf{S}} \psi^0_i]$, where $\{\psi^0_i\}_{i \in \mathsf{S}}$ is a family of measurable potentials with $\psi^0_i : \mathbb{R}^d \to \mathbb{R}$ such that $\left|\int_{\mathbb{R}^d} \psi^0_i \mathrm{d}\mu_i\right| < \infty$ for any $i \in \mathsf{S}$. In the case where $r$ is chosen as a leaf of $\mathsf{T}$, this result implies that (TreeSB) is unchanged if

(a)  $\varphi_r = \mathrm{d}\nu/\mathrm{dLeb}$ where $\nu \in \mathscr{P}$ is such that $\mathrm{KL}(\mu_r|\nu) < \infty$,

(b)  $r$ is replaced by $r' \in \mathsf{S}$, as long as $\mathrm{H}(\mu_r) < \infty$ and $\mathrm{H}(\mu_{r'}) < \infty$.

Therefore, under **A**0, the setting chosen in Section 3 is equivalent to any other setting where $r$ is arbitrarily chosen in $\mathsf{S}$ and $\varphi_r = \mathrm{d}\nu/\mathrm{dLeb}$ where $\mathrm{KL}(\mu_r|\nu) < \infty$.

Consider now the case where $r \in \mathsf{S}^{\mathrm{c}}$, *i.e.*, $r$ is not a leaf of $\mathsf{T}$. Then, the choice of $\varphi_r$ can not be made arbitrarily anymore, since it determines a further regularization on the $r$-th marginal of the solution to (TreeSB). In this setting, the sequence defined by (mIPF) is unchanged. Hence, TreeDSB proceeds in the same manner as presented in Section 3, except for the first iteration, which we detail now.

Let us define $\mathsf{P} = \mathrm{path}_{\mathsf{T}_{i_0}}(i_0, r)$, where $\mathsf{T}_{i_0} = (\mathsf{V}, \mathsf{E}_{i_0})$ is the directed version of $\mathsf{T}$ rooted in $i_0$. We recall that first iterate of (mIPF) is defined by

$$\pi^1 = \mathrm{argmin}\{\mathrm{KL}(\pi \mid \pi^0) : \pi \in \mathscr{P}^{(\ell+1)}, \pi_{i_0} = \mu_{i_0}\} \,.$$

Following the proof of Lemma 12, it is clear that

$$\pi^1 = \mu_{i_0} \bigotimes_{(v,v') \in \mathsf{P}} \pi^0_{v'|v} \bigotimes_{(v,v') \in \mathsf{E}_{i_0}\backslash\mathsf{P}} \pi^0_{v'|v} = \mu_{i_0} \bigotimes_{(v,v') \in \mathsf{E}_{i_0}} \pi^0_{v'|v} \,,$$

where we emphasize that $\mathsf{P} = \{(v,v') \in \mathsf{E}_{i_0} : (v',v) \in \mathsf{E}_r\}$. Therefore, Proposition 2 still applies between $r$ and $i_0$, by considering $r$ instead of $i_{K-1}$. In practice, this means that the first iteration of TreeDSB consists in computing the time reversal of the path measures $\mathbb{P}^0_{(v',v)}$ for any $(v,v') \in \mathsf{P}$.

**Extension of the regularized Wasserstein barycenter problem** (regWB).   Consider the regularized Wasserstein-2 barycenter problem defined as follows

$$\mu^\star_\varepsilon = \arg\min\{\textstyle\sum_{i=1}^\ell w_i W^2_{2,\varepsilon/w_i}(\mu,\mu_i) + \ell\varepsilon\mathrm{H}(\mu) + \varepsilon\mathrm{KL}(\mu \mid \mu_0) : \mu \in \mathscr{P}\} , \qquad (\mu_0\text{-regWB})$$

where $(w_i)_{i\in\{1,\dots,\ell\}} \in (0,+\infty)^\ell$ and $\mu_0 \in \mathscr{P}$ is a reference measure. This formulation admits a further regularization compared to (regWB), which tends to make $\mu^\star_\varepsilon$ closer to $\mu_0$. In particular, given a Wasserstein barycenter problem onto a star-shaped tree, the formulation ($\mu_0$-regWB) may be more adapted than (regWB) if we have an *a priori* on the form of the regularized barycenter. In the case where $\mu_0 = \mathrm{N}(0, \sigma_0^2 \mathrm{I}_d)$, letting $\sigma_0 \to \infty$, we recover the $(\ell\varepsilon, (\ell-1)\varepsilon)$ doubly-regularized Wasserstein barycenter problem (regWB). In the same spirit as Proposition 7, we can derive the following result from Proposition 24, which proves that ($\mu_0$-regWB) can be solved with TreeDSB.

**Proposition 25.** *Let $\varepsilon > 0$ and $\mu_0 \in \mathscr{P}$ such that $\mu_0 \ll$ Leb. Assume A0. Also assume that $\mathsf{T}$ is a star-shaped tree with central node indexed by $0$, and that the reference measure of (TreeSB) defined in (2) verifies $r = 0$ and $\varphi_r = \mathrm{d}\mu_0/\mathrm{dLeb} > 0$. Under A2, ($\mu_0$-regWB) has a unique solution $\pi^\star_0$, where $\pi^\star$ is the solution to (TreeSB).*

Below, we provide practical guidelines to parameterize $\mu_0$ when it is chosen as a Gaussian distribution.

**Gaussian design of $\mu_0$ in ($\mu_0$-regWB).**   Consider an undirected star-shaped tree $\mathsf{T}$ with $K + 1$ vertices and leaves $\{1, \dots, K\}$. In order to incorporate the marginal constraints in the penalization brought by $\mu_0$ when it is a Gaussian distribution, we set its mean to $\sum_{i=1}^K \mathbb{E}[\mu_i]/K$ and its diagonal covariance matrix as $\alpha \times (\sum_{i=1}^K \mathrm{diag}(\mathrm{Cov}[\mu_i])^{-1}/K)^{-1}$, where the inverse operation is component-wise and $\alpha$ is a positive hyperparameter. This choice of variance helps to correctly explore the state-space at the very first iteration of TreeDSB, which is key to ensure numerical stability. In this setting, (TreeSB) verifies **A**2 and **A**3, by Proposition 15 and Proposition 16. In particular, we use this approach for two of our experiments: synthetic Gaussian datasets and Bayesian fusion, see Appendix G.

# F   Algorithmic techniques

**Time discretization in TreeDSB.**   Denote $k_n = (n-1)\,\mathrm{mod}(K)$ for any $n \in \mathbb{N}$. Let $\mathsf{T} = (\mathsf{V}, \mathsf{E})$ be a weighted undirected tree and consider the multi-marginal Schrödinger bridge problem (TreeSB) associated to this tree. We recall that for any $\{v,v'\} \in \mathsf{E}$, we define $T_{v,v'} = \varepsilon/(2w_{v,v'})$.

Consider the path measures $\{\mathbb{P}^n_{(v,v')}\}_{n\in\mathbb{N},(v,v')\in\mathsf{E}_{k_n}}$ recursively defined by (a) and (b). By combining Proposition 1, Proposition 2 and results on time reversal theory (Haussmann & Pardoux, 1986), we obtain by recursion that for any $n \in \mathbb{N}$, any $(v,v') \in \mathsf{E}_{k_n}$, $\mathbb{P}^n_{(v,v')}$ is associated with a Stochastic Differential Equation on $[0, T_{v,v'}]$ given by

$$\mathrm{d}\mathbf{X}_t = f^n_{t,v,v'}(\mathbf{X}_t)\mathrm{d}t + \mathrm{d}\mathbf{B}_t, \quad \mathbf{X}_0 \sim \pi^n_v . \tag{29}$$

Let $N \in \mathbb{N}^*$. In order to sample from the dynamics (29) at iteration $n \in \mathbb{N}$, we consider its Euler-Maruyama discretization on $(N+1)$ time steps,

$$X_{m+1} = X_m + \gamma_{m+1} f^n_{t_m,v,v'}(X_m) + \sqrt{\gamma_{m+1}} Z_{m+1}, \quad X_0 \sim \pi^n_v , \tag{30}$$

where $Z_m \sim \mathrm{N}(0, \mathrm{I}_d)$ for any $m \in \{1, \dots, N\}$, $t_m = \sum_{i=1}^m \gamma_i$, and $\{\gamma_m\}_{m=1}^N \in (0,\infty)^N$ is a time schedule such that $\sum_{m=1}^N \gamma_m = T_{v,v'}$. This results in approximating the path measure $\mathbb{P}^n_{(v,v')}$ by the joint distribution $\pi^{n,N}_{(v,v')} \in \mathscr{P}^{(N+1)}$ defined by

$$\pi^{n,N}_{(v,v')} = \pi^n_v \bigotimes_{m=0}^{N-1} \pi^{n,N}_{(v,v'),m+1|m} ,$$

where $\pi^{n,N}_{(v,v'),m+1|m}(\cdot|x_m) = \mathrm{N}(x_m + \gamma_{m+1} f^n_{t_m,v,v'}(x_m), \gamma_{m+1}\mathrm{I}_d)$ for any $m \in \{0, \dots, N-1\}$. If $N$ is chosen large enough, then $\pi^{n,N}_{(v,v'),m}$ and $\mathbb{P}^n_{(v,v'),t_m}$ have approximately the same distribution

for any $m \in \{0, \dots, N\}$. Consequently, $(\mathbb{P}^n_{(v,v')})^R$ is naturally approximated by the joint distribution $\tilde{\pi}^{n,N}_{(v,v')} \in \mathscr{P}^{(N+1)}$ defined by

$$\tilde{\pi}^{n,N}_{(v,v')} = \pi^n_{v'} \otimes_{m=0}^{N-1} \pi^{n,N}_{(v,v'),N-m-1|N-m} \ .$$

If $N$ is chosen large enough, we obtain that

$$\pi^{n,N}_{(v,v'),N-m-1|N-m}(\cdot|x_{N-m})$$
$$= \mathrm{N}(x_{N-m} - \gamma_{N-m} f^n_{t_{N-m},v,v'}(x_{N-m}) + \gamma_{N-m} \nabla \log p_{v,v',t_{N-m}}(x_{N-m}), \gamma_{N-m} \mathrm{I}_d) \ ,$$

where $p_{v,v',t}$ is the density of $\mathbb{P}^n_{(v,v'),t}$ w.r.t. the Lebesgue measure.

Following the construction of our dynamic iterates, we now explain how the sequence $\{\pi^n_{(v,v')}\}_{n \in \mathbb{N}^*, (v,v') \in \mathsf{E}_{k_n}}$ is recursively defined. Let $n \in \mathbb{N}$, $k_n = (n-1) \mathrm{mod}(K)$. Define the path $\mathsf{P}_n = \mathrm{path}_{\mathsf{T}_{i_{k_n}}}(i_{k_n}, i_{k_n+1})$. Then, for any $(v,v') \in \mathsf{E}_{k_n+1}$,

(a) if $(v,v') \in \mathsf{E}_{k_n} \backslash \mathsf{P}_n$, then $\pi^{n+1,N}_{(v,v')} = \pi^{n+1}_v \otimes_{m=0}^{N-1} \pi^{n,N}_{(v,v'),m+1|m}$,

(b) if $(v',v) \in \mathsf{P}_n$, then $\pi^{n+1,N}_{(v,v')} = \pi^{n+1}_v \otimes_{m=0}^{N-1} \pi^{n,N}_{(v',v),N-m-1|N-m}$.

These computations may be obtained by considering the sequence given by (mIPF) to solve the multi-marginal Tree-SB problem associated to $\mathsf{T}^{(N)} = (\mathsf{V}^{(N)}, \mathsf{E}^{(N)})$, the $N$-discretized version of $\mathsf{T}$ (see Appendix B) with weights $w^{(N)}_{e_m} = 2\gamma_m/\varepsilon$, which is given by

$$\pi^\star = \mathrm{argmin}\{\mathrm{KL}(\pi|\pi^{0,N}) \ : \ \pi \in \mathscr{P}^{(\mathsf{V}^{(N)})}, \ \pi_i = \mu_i \ , \forall i \in \mathsf{S}\} \ ,$$

with $\pi^{0,N} = \pi^0_r \otimes_{(v,v') \in \mathsf{E}_r} \pi^{0,N}_{(v,v'),1:N|0}$.

To approximate the IPF recursion given by (a) and (b), we use **on each edge** of $\mathsf{T}$ the score-matching approach of De Bortoli et al. (2021), which avoids heavy computations of score approximations. The next proposition is direct adaptation of (De Bortoli et al., 2021, Proposition 3).

**Proposition 26.** *Assume that for any $n \in \mathbb{N}$, any $(v,v') \in \mathsf{E}_{k_n}$ with $k_n = (n-1) \mathrm{mod}(K)$, we have*

$$\pi^{n,N}_{(v,v'),m+1|m}(\cdot|x_m) = \mathrm{N}(F^n_{m,v,v'}(x_m), \gamma_m \mathrm{I}_d) \ .$$

*Let $n \in \mathbb{N}$. Consider the path $\mathsf{P}_n = \mathrm{path}_{\mathsf{T}_{i_{k_n}}}(i_{k_n}, i_{k_n+1})$. Let $(v,v') \in \mathsf{E}_{k_n+1}$. Define $p^n = \pi^{n,N}_{(v,v')}$ and $m_N = N - m - 1$. Then, if $(v',v) \in \mathsf{P}_n$, we have*

$$F^{n+1}_{m,v,v'} = \mathrm{argmin}_{F \in \mathrm{L}^2(\mathbb{R}^d, \mathbb{R}^d)} \tag{31}$$
$$\mathbb{E}_{p^n_{m_N, m_N+1}}[\|F(X_{m_N+1}) - (X_{m_N+1} + F^n_{m_N,v',v}(X_{m_N}) - F^n_{m_N,v',v}(X_{m_N+1}))\|^2],$$

*otherwise, we have $F^{n+1}_{m,v,v'} = F^n_{m,v,v'}$.*

In practice, we use two neural networks per edge $\{v,v'\} \in \mathsf{E}$, one for each possible direction of the edge, such that $F_{v,v'}(\theta^n_{v,v'}, m, x) \approx F^n_{m,v,v'}(x)$ and $F_{v',v}(\theta^n_{v',v}, m, x) \approx F^n_{m,v',v}(x)$. For any $\{v,v'\} \in \mathsf{E}$, the parameter $\theta^n_{v,v'}$ is updated at iteration $n$ via the score matching loss defined by (31) in Proposition 26 if $(v,v') \in \mathrm{path}_{\mathsf{T}_{i_{k_n}}}(i_{k_n}, i_{k_n+1})$, see Algorithm 1.

## G Additional experimental results and details

The numerical experiments presented in Section 7 are obtained by our own Pytorch implementation, which is inspired from the code[6] provided by De Bortoli et al. (2021). We first provide information on the general setting of our experiments in Appendix G.1, and then give details on each of them in Appendix G.2 along with additional results. We recall that a mIPF cycle is defined as a subset of $K$ consecutive iterations of (mIPF) and that the order of the leaves given by $\{i_0, \dots, i_{K-1}\}$ is randomly shuffled at each new mIPF cycle.

---

[6] https://github.com/JTT94/diffusion_schrodinger_bridge

### G.1 General experimental setup

**Implementation of Algorithm 1 in practice.** Let $n \in \mathbb{N}$, with $k_n = (n-1) \bmod(K)$, $k_n + 1 = n \bmod(K)$. Consider the path $\mathsf{P}_n = \mathrm{path}_{\mathsf{T}_{i_{k_n}}}(i_{k_n}, i_{k_n+1})$. Assume that we are provided with a dataset $\mathsf{D}_{i_{k_n}}$, which contains $M$ samples from $\pi_{i_{k_n}}^n$. Following Lines 7-9 in Algorithm 1, we apply processes (a) and (b) recursively on the edges $(v, v') \in \mathsf{P}_n$.

(a) Sampling step (Line 7). For any $x_0 \in \mathsf{D}_v$, we sample from the diffusion trajectory (30) given by the Euler Maruyama discretization of $\mathbb{P}_{v,v'}^n$ starting from $x_0$. This gives us $M \times N$ trajectory samples. We then store the last iterate of each trajectory in a new dataset $\mathsf{D}_{v'}$, which thus approximates $\pi_{v'}^n$.

(b) Training step (Lines 8-9). In order to avoid heavy computation, we approximate the *mean-matching* loss (31) by an unbiased estimator obtained by subsampling $b$ elements from the *full* trajectories computed in the sampling process, see (De Bortoli et al., 2021, Eq. (97)-(98)). Here, $b$ refers to the *batch-size* parameter of the neural networks. Then, we perform gradient descent to optimize the parameter $\theta_{v',v}$, which parameterizes the *backward* drift on the edge $(v, v')$.

To avoid any bias issue, the whole trajectories obtained at process (a) are refreshed at a certain frequency over the training iterations of the neural networks by once again simulating the diffusion (30). In our experiments, this refresh occurs each 500 iterations.

**Setting of the time discretization.** The number of time-steps $N$ in the time discretization of the diffusions is chosen to be even and identical for each of the edges of the tree. Let $\{v, v'\} \in \mathsf{E}$. We now give details on the design of the time schedule $\{\gamma_k\}_{k=1}^N$ related to the edge $\{v, v'\}$, see Appendix F. Following De Bortoli et al. (2021), we choose this sequence to be invariant by time reversal and consider $\gamma_k = \gamma_0 + (2k/N)(\bar{\gamma} - \gamma_0)$ for any $k \in \{0, \ldots, N/2\}$ (the rest of the sequence being obtained by symmetry) where $\gamma_0$ is a free parameter and $\bar{\gamma}$ is determined by $\sum_{k=1}^N \gamma_k = T_{v,v'}$. In our experiments, we set $N = 50$ and $\gamma_0 = 10^{-5}$.

**Sampling improvement.** In our code, we implemented the corrector scheme of Song et al. (2021) and the *probability flow*-based sampling approach detailed in (De Bortoli et al., 2021, Section H.3), but did not observe any significant improvement in our experiments using one of these techniques.

**Choice of the architectures of the neural networks.** In the case of the experiments related to synthetic datasets (two-dimensional toy datasets, Gaussian distributions) and to the subset posterior aggregation task, we implement the same architecture as presented in (De Bortoli et al., 2021, Figure 3). We refer to this model as "Basic Model" and detail it in Figure 6. In the "Basic Model", the PositionalEncoding block applies the sine transform described in Vaswani et al. (2017), with output dimension equal to 32, and each MLP Block represents a Multilayer Perceptron Network. In particular, MLP-Block (1a) has shape $(d, 128, \max(256, 2d))$, MLPBlock (1b) has shape $(32, 128, \max(256, 2d))$, and MLPBlock (2) has shape $(2 \times \max(256, 2d), \max(256, 2d), \max(128, d), d)$, where $d$ denotes the dimension of input data. We optimize the networks with ADAM (Kingma & Ba, 2014) with learning rate $10^{-4}$ and momentum 0.9. For each of the networks, we set the batch size to 4,096 and the number of iterations to 10,000 for the synthetic datasets and 15,000 for the subset posterior aggregation task. Our experiments ran on 1 Intel Xeon CPU Gold 6230 20 cores @ 2.1 Ghz CPU.

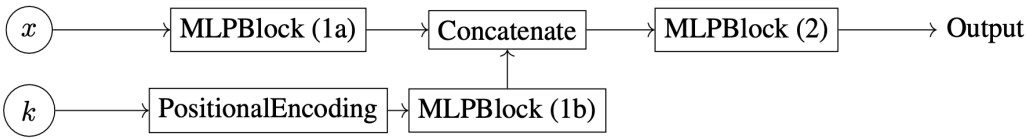

Figure 6: Architecture of the "Basic Model".

In the case of the experiments related to MNIST dataset, we use a reduced UNET architecture based on Nichol & Dhariwal (2021), where we set the number of channels to 64 rather than 128. We implement an exponential moving average of network parameters across training iterations, with rate 0.999. We optimize the networks with ADAM (Kingma & Ba, 2014) with learning rate $10^{-4}$ and momentum 0.9. Finally, we set the batch size to 256 and the number of training iterations to 30,000. Our experiments ran using 1 Nvidia A100.

**Details on regularized state-of the art methods.** We run the fsWB algorithm (Cuturi & Doucet, 2014) with the implementation provided by Flamary et al. (2021). For each experiment, we run 100 Sinkhorn iterations with 1500 samples for each dataset (*i.e.*, the maximum number of samples that it can generate) and set the regularization parameter $\varepsilon$ to its lowest value such that the algorithm is stable. Finally, for sake of fairness with our method, we initialise the barycenter measure with $\pi_r^0$ when solving the problem ($\mu_0$-regWB) for synthetic Gaussian datasets and Bayesian fusion. To run the crWB algorithm (Li et al., 2020), we use the code provided by the authors. We consider the quadratic regularization, which is shown to be empirically more stable than entropic regularization. Following Fan et al. (2020), we choose the potential networks to be fully connected neural networks with 3 hidden layers of shape $(\max(128, 2d), \max(128, 2d), \max(128, 2d))$. The activation functions are ReLu. We optimize the networks with ADAM (Kingma & Ba, 2014) with learning rate $10^{-4}$ for the subset posterior aggregation task and $10^{-3}$ for the Gaussian experiment. Finally, we set the batch size to 4,096 and the number of training iterations to 50,000. We highlight that fsWB and crWB solve a regularized Wasserstein barycenter problem, which does not contain an additional *penalization* term on the entropy of the barycenter, contrary to TreeDSB.

## G.2 Details on the experiments

**Synthetic Gaussian datasets.** For each dimension that we consider, we generate three different triplets of random non-diagonal covariance matrices whose condition number is less than 10. We then run the algorithms on each triplet and aggregate the obtained results. The Gaussian datasets contain 1,500 samples for fsWB, and 10,000 samples for crWB and TreeDSB. We run fsWB with the following settings $(d, \varepsilon) \in \{(2, 0.1), (16, 0.2), (64, 0.5), (128, 1.0), (256, 2.0)\}$. We run TreeDSB for 10 mIPF cycles with regularization parameter $\varepsilon = 0.1$, starting from the central node initialized to a Gaussian distribution $\mu_0$ chosen as detailed in Appendix F with $\alpha = 1$. Thus, we solve the regularized Wasserstein barycenter problem ($\mu_0$-regWB), which contains an additional regularization with respect to $\mu_0$. This choice is justified, since the non-regularized barycenter is known to be a Gaussian distribution, and $\mu_0$ can be seen as an *a priori* for the regularized barycenter. For each of the three settings, we keep the best result among the 30 mIPF iterations. In this setting, TreeDSB and crWB have roughly the same training time.

**Subset posterior aggregation.** When considering a dataset splitted into several subdatasets, a common paradigm in bayesian inference consists in running Monte Carlo Markov Chain methods separately on these subdatasets, and then merge the obtained posteriors to recover the full posterior. The barycenter of these subdataset posteriors is proved to be close to the full data posterior under mild assumptions (Srivastava et al., 2018). In our setting, we consider the posterior aggregation problem for the logistic regression model associated to the `wine` dataset[7] ($d = 42$) with 3 subdatasets. We consider here two splitting methods: (i) either, data is uniformly splitted between 3 subdatasets with respect to the label distribution, denoted by `wine-homogeneous`, or (ii) data is splitted with some heterogeneity according to a Dirichlet distribution whose parameter is randomly chosen, denoted by `wine-heterogeneous`. Following Korotin et al. (2021), we use the stochastic approximation trick so that the subset posterior samples do not vary consistently from the full posterior in covariance (Minsker et al., 2014). We implement the Unadjusted Langevin Algorithm (ULA) to sample from each subdataset posterior and from the full posterior. In each case, we run ULA for $5.5 \cdot 10^6$ iterations with a well chosen step-size, and obtain 9,900 samples after applying a *burn-in* of order $10\%$ and then a *thinning* of size 500. We provide in Figure 7 some metrics which assess the quality of this sampling process. We recall that the the full posterior samples serve as ground truth in this experiment.

The results presented in Table 2 were computed as follows. For fsWB, we first subsample 1,500 samples out of the 9,900 samples from each posterior, and then run the algorithm with $\varepsilon = 0.5$. We repeat three times this procedure and then aggregate the results. In the case of crWB and TreeDSB, we run the algorithms three times with various seeds. Similarly to the Gaussian setting, we run TreeDSB for 10 mIPF cycles with regularization parameter $\varepsilon = 0.1$. We start from the central node with a Gaussian distribution $\mu_0$ chosen as detailed in Appendix F with $\alpha = 1$, and thus solve the barycenter formulation ($\mu_0$-regWB). For each of the three settings, we keep the best result among the 30 IPF iterations. In this setting, TreeDSB and crWB have roughly the same training time.

---

[7]https://archive.ics.uci.edu/ml/datasets/wine

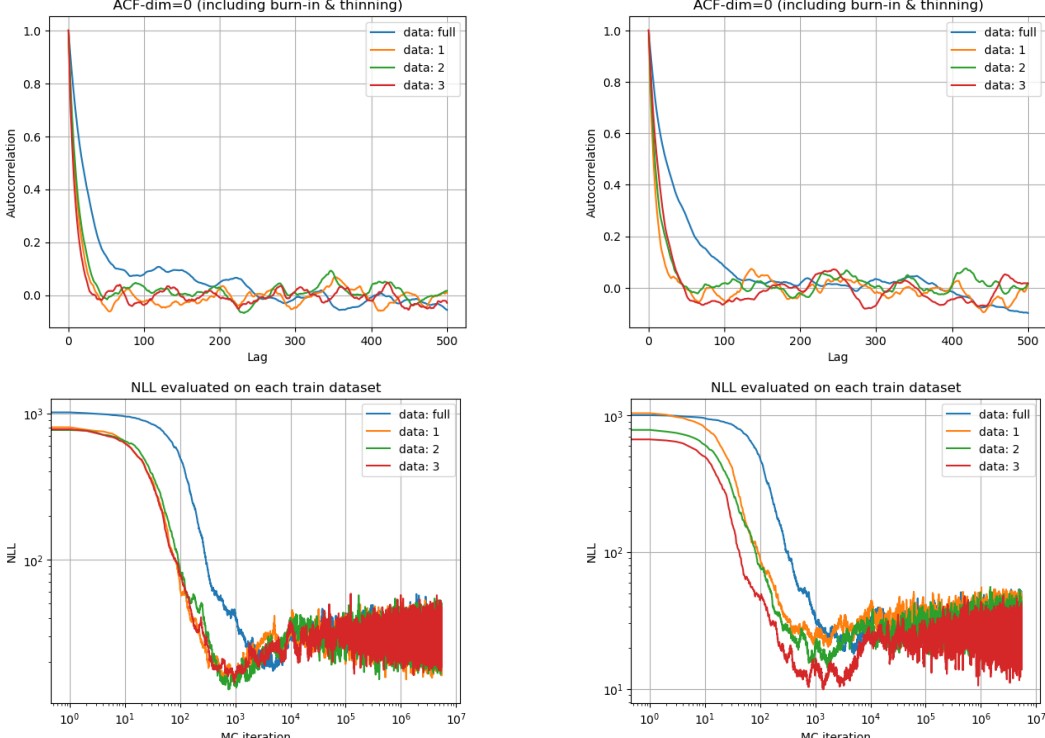

Figure 7: Evaluation of the sampling process for `wine-homogeneous` (left) and `wine-heterogeneous` (right). We display the Autocorrelation function on 500 lags (above) and the evolution over the iterations of ULA of the negative log-likelihood (NLL) evaluated on each training dataset (below). In particular, the samples are decorrelated and the NLL has a satisfying profile.

**Synthetic two-dimensional datasets.** In this setting, we consider three different datasets (*Swiss-roll*, *Circle* and *Moons*) that each contain 10,000 samples. Since we do not have an *a priori* on the shape of the barycenter between these datasets, we consider the regularized Wasserstein barycenter problem (regWB), *i.e.*, $r$ is chosen as a leaf and corresponds to one of the input datasets. We emphasize that this experiment is not intended to demonstrate the superiority of TreeDSB to compute 2D Wasserstein barycenters, but is rather meant to illustrate that (a) the marginals of the leaves are well recovered by the algorithm, see Figure 3, and that (b) the obtained barycenter is consistent when diffusing from the different leaves, see Figure 4. In all our experiments on 2D datasets, we observed that (a) was persistently verified without difficulty. In this section, we rather aim at illustrating (b) by providing additional results which assess the quality of the barycenter obtained by TreeDSB with respect to the choice of the starting leaf $r$ and to the choice of the regularization parameter $\varepsilon$.

To do so, we consider three different choices of regularization in TreeDSB: (i) $\varepsilon = 0.2$ (50 mIPF cycles), see Figure 8, (ii) $\varepsilon = 0.1$ (50 mIPF cycles), see Figure 9 and (iii) $\varepsilon = 0.05$ (60 mIPF cycles), see Figure 10. For each of these settings, we run TreeDSB with the starting leaf $r$ chosen as *Swiss-roll* (first row), *Circle* (second row) or *Moons* (third row), and display the final barycenter obtained by diffusing from *Swiss-roll* (first column), *Circle* (second column) and *Moons* (third column). Note that the vertex 0 always corresponds to the starting leaf, the vertex 1 to the barycenter node and that Figure 4 corresponds to the first row of Figure 9.

We can make the following observations. First, the estimated barycenter is always coherent within each row, which assesses the convergence of our method. Then, for each value of $\varepsilon$, the TreeDSB barycenter is rather consistent between the rows, *i.e.*, the choice of the starting leaf does not have a meaningful impact on our method. Finally, as expected, we observe that the support of the barycenter is less and less diffuse as long as $\varepsilon$ decreases.

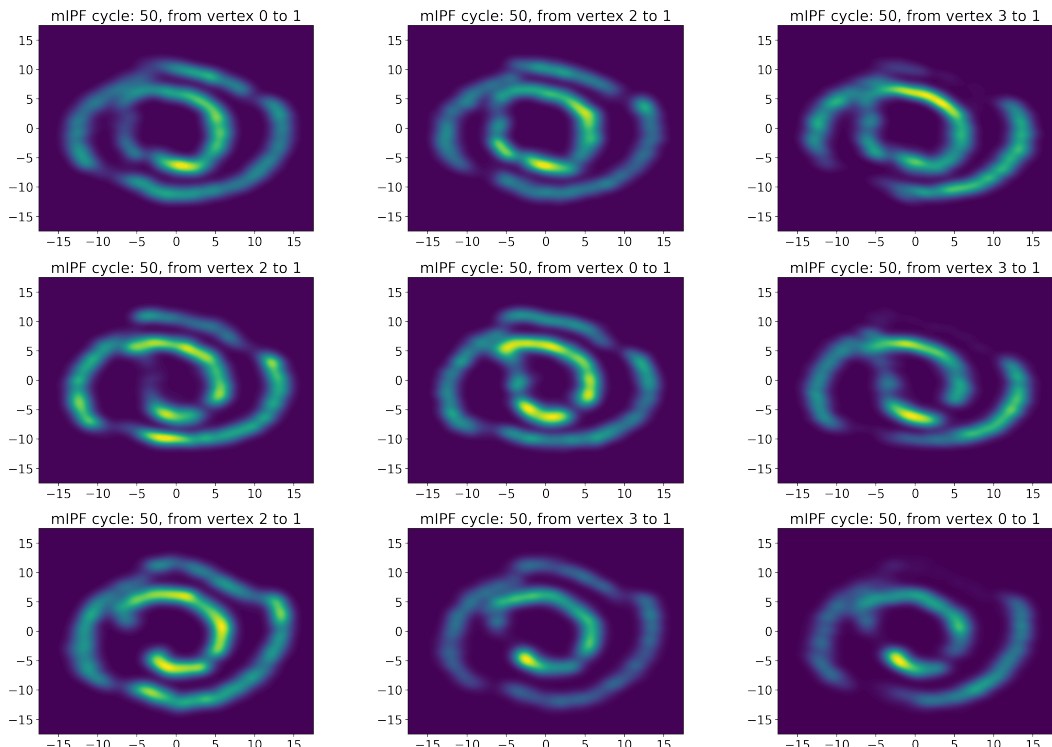

Figure 8: Estimated 2D barycenter obtained by TreeDSB with $\varepsilon = 0.2$ (50 mIPF cycles). First row: starting from *Swiss-roll*. Second row: starting from *Circle*. Third row: starting from *Moons*.

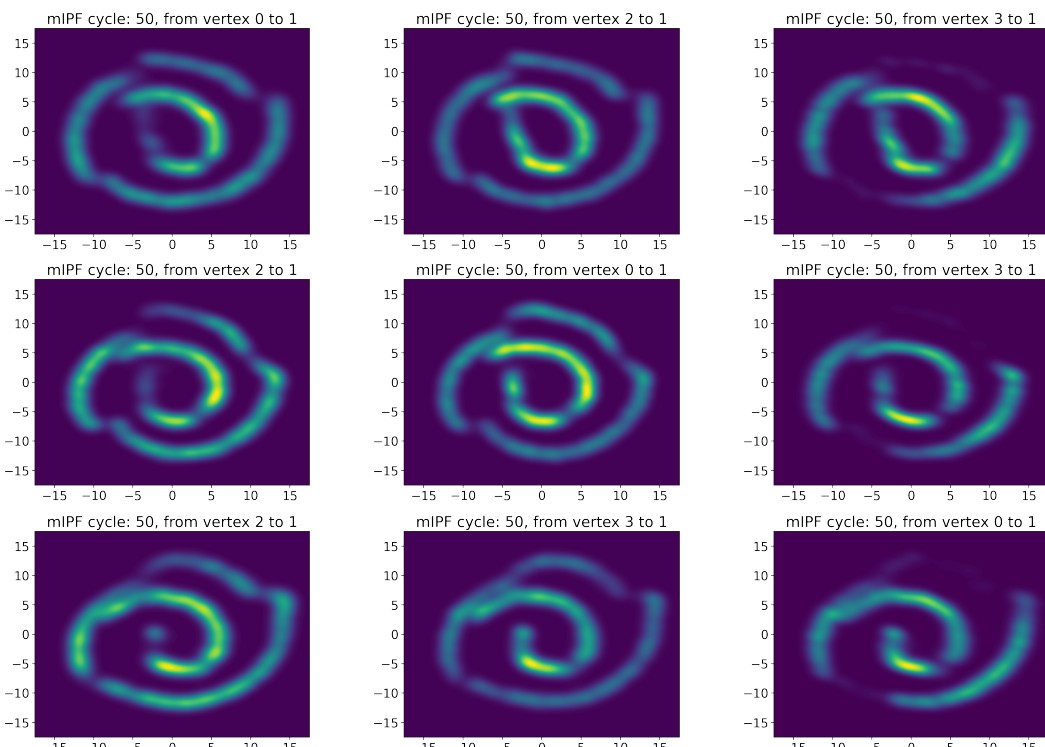

Figure 9: Estimated 2D barycenter obtained by TreeDSB with $\varepsilon = 0.1$ (50 mIPF cycles). First row: starting from *Swiss-roll*. Second row: starting from *Circle*. Third row: starting from *Moons*.

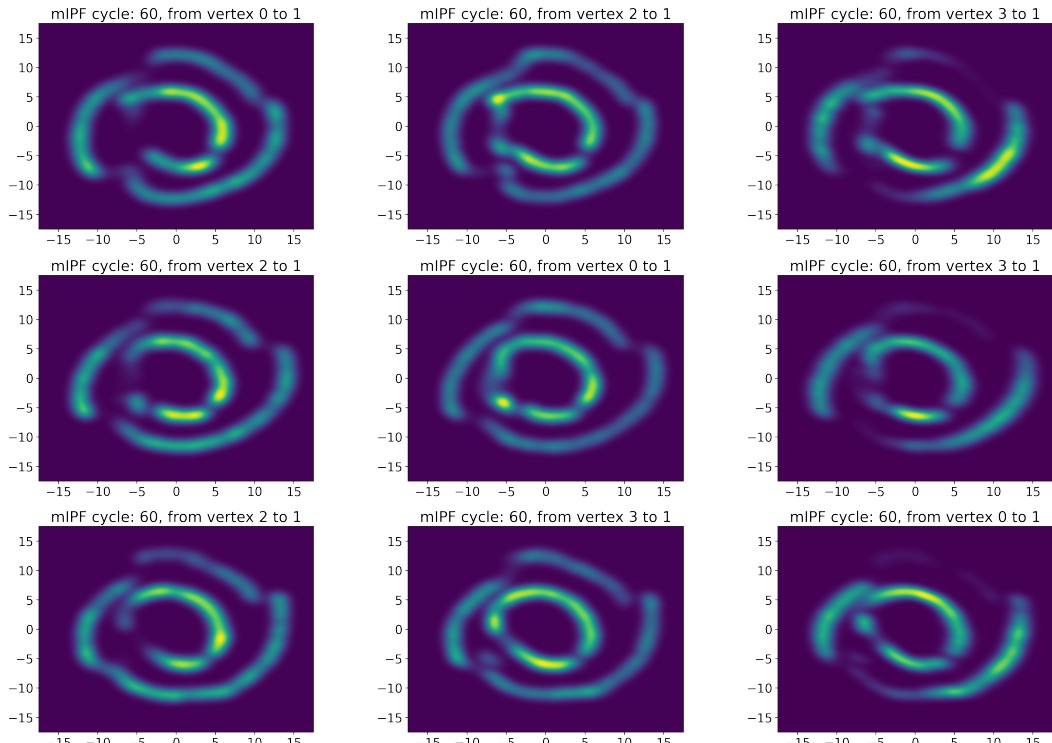

Figure 10: Estimated 2D barycenter obtained by TreeDSB with $\varepsilon = 0.05$ (60 mIPF cycles). First row: starting from *Swiss-roll*. Second row: starting from *Circle*. Third row: starting from *Moons*.

For purpose of illustration, we provide in Figure 11 the barycenter obtained by state-of-the-art two-dimensional *in-sample* methods that are available in POT library (Flamary et al., 2021): (i) non-regularized free-support Wasserstein barycenter (Cuturi & Doucet, 2014), (ii) entropic-regularized free-support Wasserstein barycenter (fsWB) with $\varepsilon = 0.5$ (Cuturi & Doucet, 2014) and (iii) entropic-regularized convolutional Wasserstein barycenter with $\varepsilon = 5.10^{-4}$ (Solomon et al., 2015), which is specifically designed for images. We notably observe that TreeDSB cannot capture the full complexity of the 2D barycenter compared to these methods. We infer that this gap comes from the *dynamic* nature of TreeDSB, since increasing the number of training iterations per IPF iteration or improving the complexity of the neural networks did not bring any significant change in our results. Finally, we recall that the methods (i), (ii) and (iii) do not scale well with dimension, and have to be completely run again when new data samples are available, contrary to TreeDSB.

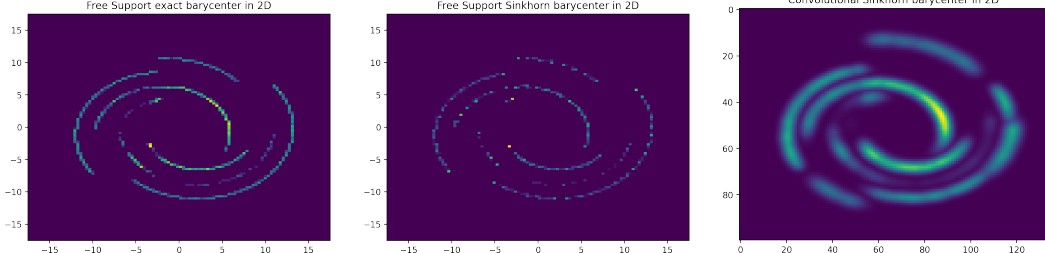

Figure 11: Estimated 2D barycenter obtained by *in-sample* algorithms. From left to right: Cuturi & Doucet (2014) (non-regularized), Cuturi & Doucet (2014) (regularized), Solomon et al. (2015).

**MNIST Wasserstein barycenter.** This setting can be qualified as *high-dimensional*, since the data dimension is $d = 784$. Here, each digit dataset contains 1,000 samples. As in the two-dimensional setting, we do not have an *a priori* on the shape of the barycenter between MNIST digits, and thus consider the formulation (regWB), where the root $r$ is chosen as a leaf. We propose below several experiments to assess the scalability of TreeDSB to this setting.

**Digits 0 and 1.** In Figure 12, we report the results obtained by running TreeDSB on MNIST digits 0 and 1, for 15 mIPF cycles with $\varepsilon = 0.5$, starting from the leaf MNIST-0. We display 25 samples from the reconstructed MNIST-0 marginal (first column), from the reconstructed MNIST-1 marginal (fourth column), from the estimated barycenter by diffusing from MNIST-0 (second column) and diffusing from MNIST-1 (third column). We notably observe that the digits are well recovered and that the barycenter samples are consistent. We draw the reader's attention to the fact that TreeDSB showed numerical instability with a regularization value $\varepsilon$ lower than 0.5. For purpose of illustration, we display in Figure 13 the Wasserstein barycenter obtained by *non-regularized* methods from Fan et al. (2020) and Korotin et al. (2021), and by the *regularized* approach from Li et al. (2020).



Figure 12: Tree DSB results for MNIST digits 0 and 1, after 15 mIPF cycles with $\varepsilon = 0.5$.

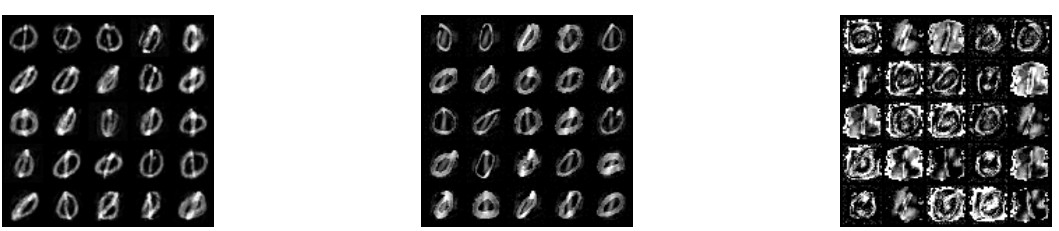

Figure 13: From left to right: Fan et al. (2020), Korotin et al. (2021) and Li et al. (2020).

**Digits 2,4 and 6.** In Figure 14, we report the results obtained by running TreeDSB on MNIST digits 2,4 and 6, for 10 mIPF cycles with $\varepsilon = 0.5$. Here, we consider three settings which differ by the starting leaf $r$ in the algorithm: MNIST-2 (first row), MNIST-4 (second row), or MNIST-6 (third row). For each of these settings, we display 30 samples from the estimated barycenter by diffusing from MNIST-2 (first column), diffusing from MNIST-4 (third column) and diffusing from MNIST-6 (third column). We notably observe a global consistency of the barycenter samples across the various settings. In Figure 15, we report the results obtained by running TreeDSB on MNIST digits 2,4 and 6, for 10 mIPF cycles with $\varepsilon = 0.2$, starting from MNIST-6. We display 30 samples from the reconstructed marginals (first row), from the estimated barycenter (second row) by diffusing from MNIST-2 (first column), diffusing from MNIST-4 (second column) and diffusing from MNIST-6 (third column). As expected, we observe less noisy barycenter samples compared to Figure 14, while still well recovering MNIST digits.

**Digits 0,1 and 4.** In Figure 16, we report the results obtained by running TreeDSB on MNIST digits 0,1 and 4, for 10 mIPF cycles with $\varepsilon = 0.5$. We consider two settings which differ by the starting leaf $r$ in the algorithm: MNIST-0 (second row) and MNIST-1 (first/third rows), for which we display samples from the reconstructed measures (first row). In Figure 17, we report the results obtained by running TreeDSB on MNIST digits 0,1 and 4, for 10 mIPF cycles with $\varepsilon = 0.2$. We consider two settings which differ by the starting leaf $r$ in the algorithm: MNIST-0 (first/second row), for which display samples from the reconstructed measures (first row), and MNIST-1 (third row). For all of these settings, we display 30 samples from the estimated barycenter by diffusing from MNIST-0 (first column), diffusing from MNIST-1 (third column) and diffusing from MNIST-4 (third column). Similarly to the digits 2-4-6, we observe consistency within the barycenter samples, unconditionally to the starting leaf, and less noise as $\varepsilon$ decreases. Note that the reconstructed MNIST digits are less truthful to the original datasets when $\varepsilon$ is low.

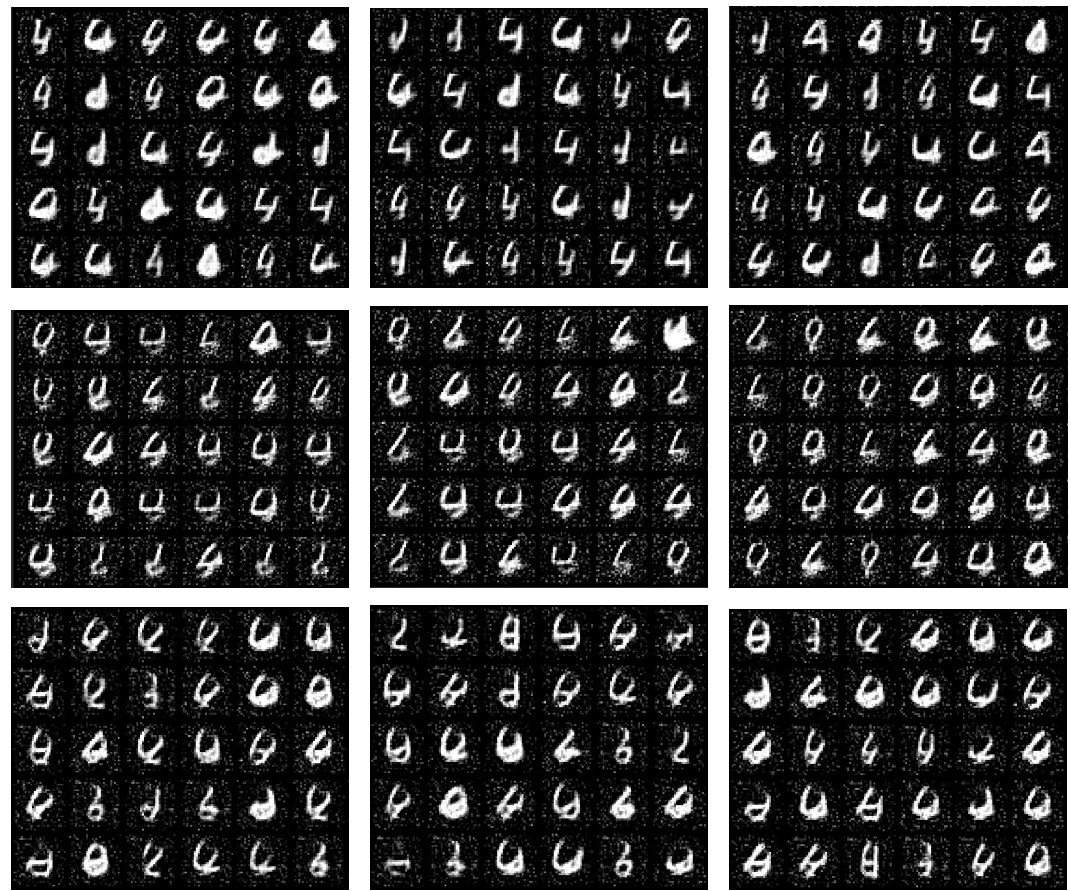

Figure 14: Tree DSB results for MNIST digits 2,4 and 6, after 10 mIPF cycles with $\varepsilon = 0.5$. First row: starting from MNIST-2. Second row: starting from MNIST-4. Third row: starting from MNIST-6.

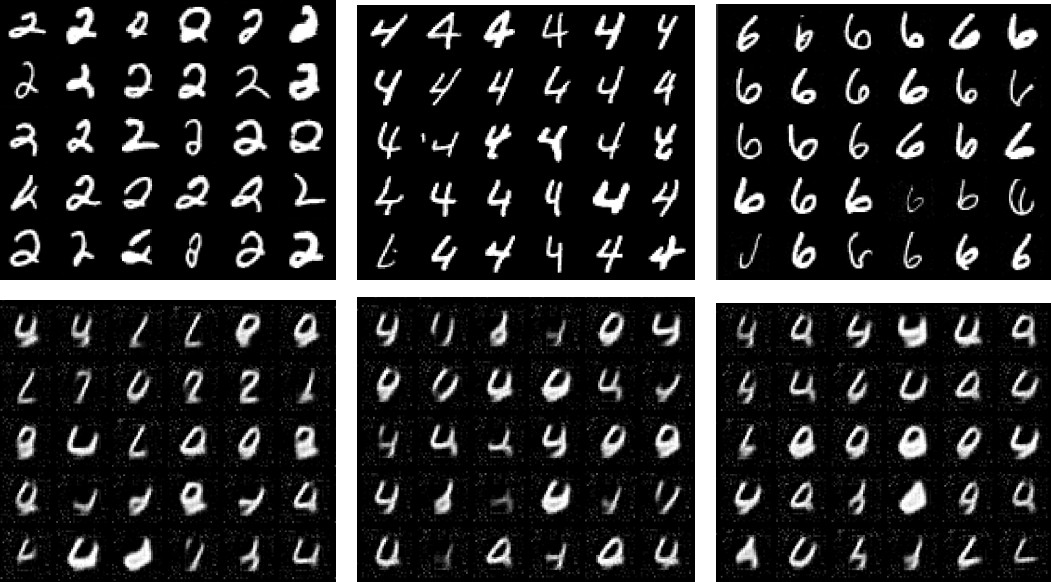

Figure 15: Tree DSB results for MNIST digits 2,4 and 6, after 10 mIPF cycles with $\varepsilon = 0.2$, starting from MNIST-6. First row: samples from the reconstructed marginals. Second row: samples from the estimated barycenter.

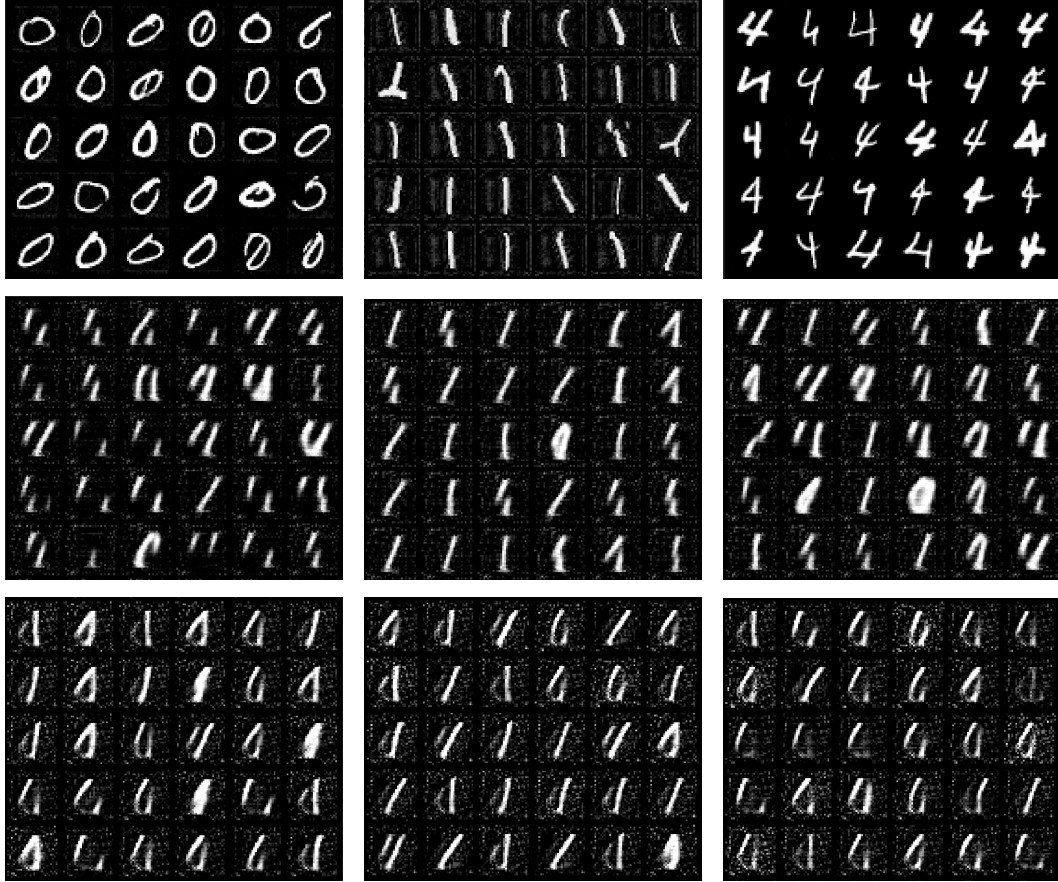

Figure 16: Tree DSB results for MNIST digits 0,1 and 4, after 10 mIPF cycles with $\varepsilon = 0.5$. First row: samples from the reconstructed marginals, starting from MNIST-1. Second row: samples from the estimated barycenter, starting from MNIST-0. Third row: samples from the estimated barycenter, starting from MNIST-1.

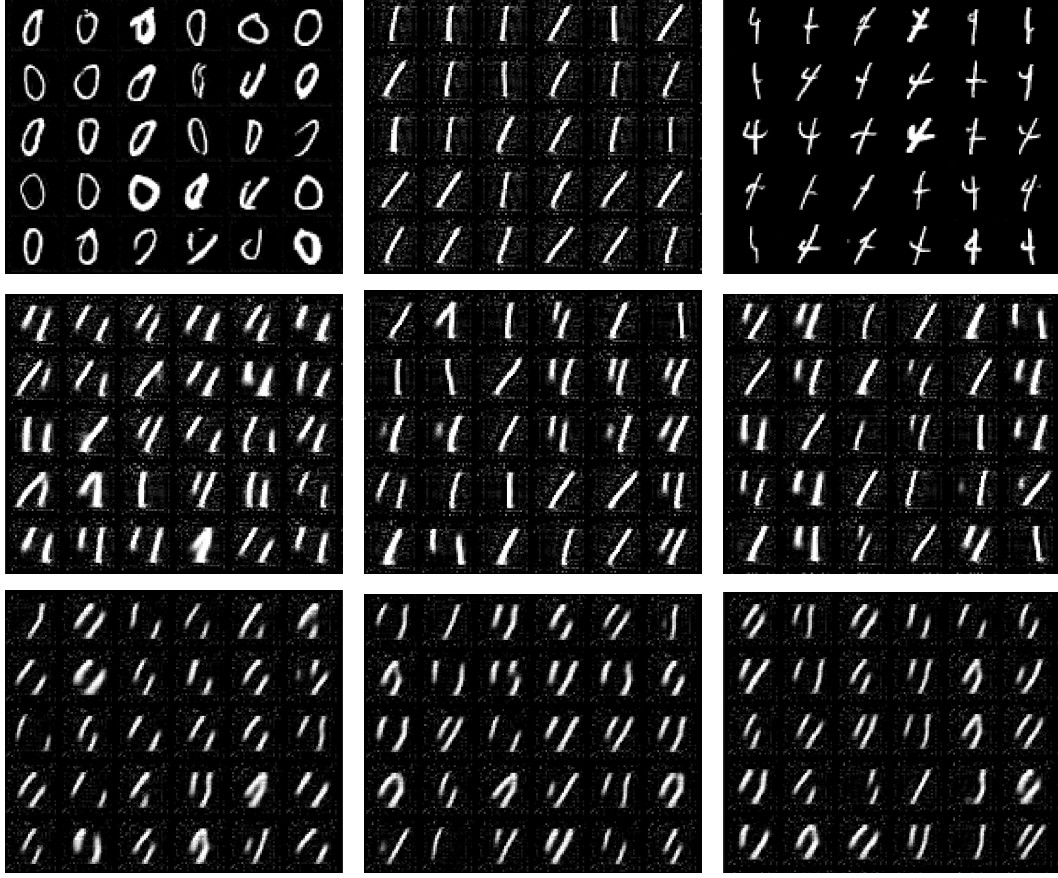

Figure 17: Tree DSB results for MNIST digits 0,1 and 4, after 10 mIPF cycles with $\varepsilon = 0.2$. First row: samples from the reconstructed marginals, starting from MNIST-0. Second row: samples from the estimated barycenter, starting from MNIST-0. Third row: samples from the estimated barycenter, starting from MNIST-1.

