{\mathsf{A}}_i^y) = 1$ for any $y \in \mathsf{A}_{-i}$, (b) $\tilde{\pi}_{-i}^0(\mathsf{A}_{-i}) = 1$. Similarly, this result holds for any $j \in \mathsf{S}^c$, *i.e.*, there exists a measurable set $\mathsf{A}_{-j} \subset \prod_{\substack{m=0 \\ m\neq j}}^{\ell} \mathsf{X}_i$ such that the following properties hold: (a) $\tilde{\mu}_j(\bar{\mathsf{A}}_j^y) = 1$ for any $y \in \mathsf{A}_{-j}$, (b) $\tilde{\pi}_{-j}^0(\mathsf{A}_{-j}) = 1$. We consider such sets $\{\mathsf{A}_{-m}\}_{m=0}^{\ell}$ for the rest of the proof and finally define the set

$$\tilde{\mathsf{A}} = \cap_{m=0}^{\ell} \tilde{\mathsf{A}}_m ,$$

where $\tilde{\mathsf{A}}_m = \mathsf{A}_{-m} \times \{u \in \bar{\mathsf{A}}_m^y : y \in \mathsf{A}_{-m}\}$. By definition, we have $\tilde{\mathsf{A}} \subset \mathsf{A} \cap \prod_{m=0}^{\ell} \mathsf{X}_m$, using the fact that $\tilde{\mathsf{A}}_m \subset \mathsf{A}$ for any $m \in \{0, \ldots, \ell\}$. In addition, for any $i \in \mathsf{S}$, we get by Fubini's theorem

$$\tilde{\pi}^0(\tilde{\mathsf{A}}_i) = \int_{\tilde{\mathsf{A}}_i} \mathrm{d}\mu_i(x_i) \otimes \mathrm{d}\tilde{\pi}_{-i}^0(x_{-i}) = \int_{\mathsf{A}_{-i}} \{\int_{\bar{\mathsf{A}}_