# OpenReview forum: "Tree-Based Diffusion Schrödinger Bridge with Applications to Wasserstein Barycenters"
_NeurIPS.cc/2023/Conference — NeurIPS 2023 spotlight_

### Official Review · Reviewer_fo2c · 2023-07-05

**Soundness:** 3 good
**Presentation:** 2 fair
**Contribution:** 3 good
**Rating:** 7
**Confidence:** 3

**Summary:**

The authors develop a novel method for approximating Wasserstein barycenters, which is based on diffusion models. This method combines ideas from the recently introduced Diffusion Schrodinger bridge and a framework for multi-marginal optimal transport with tree-structured costs.
Thus, although the paper focuses on the barycenter problem (corresponding to a star shaped tree), the proposed approach can be used for any multi-marginal optimal transport problem, where the cost is composed of pairwise terms that are structured as a tree.
A diffusion Schrodinger bridge type algorithm is presented that approximates the multimarginal iterative proportional fitting (mIPF) algorithm. The authors present novel convergence results of mIPF, which in contrast to previous works neither assume that the space is compact nor that the cost is bounded.
Finally, the application to Wasserstein barycenters is evaluated for several experiments.

**Strengths:**

I see the strengths of this paper in its originality and significance. The paper extends results for Diffusion Schrodinger bridges to a class of multi-marginal problems, including Wasserstein barycenters. Moreover, the paper extends results for tree structured multi-marginal optimal transport problems by developing a diffusion based method for these, which makes the problem scalable to high-dimensional settings.

**Weaknesses:**

The clarity of presentation should be improved in many parts of the paper. Please see comments below.

**Questions:**


My main question concerns the different update schemes mentioned in Section 3. The (mIPF) algorithm requires sampling from the whole tree. The authors argue that each (mIPF) update can be done on a path of the tree. In lines 160-173 it is described that one first samples along all edges of the path, and then learns the reverse dynamics for the path. In contrast, Algorithm 1 samples and learns the reverse dynamics for each edge on the path individually, and this is repeated iteratively along the path. That is, three different update schemes are mentioned:
a) sample along whole tree and learn reverse dynamics;
b) sample along whole path and learn reverse dynamics;
c) iterate over the edges on the path, and for each edge sample along each and learn reverse dynamics.
Please clarify why these are equivalent, especially the equivalence with scheme c) is not obvious to me.


The following list is ordered mostly chronological, and not according to relevance.

1. In Section 2 the authors should make sure that the background material and setting are introduced in a clear and concise manner. This includes my following comments, but I suggest to carefully work through the whole section.
- Are all the details on dynamic and static versions of Schrodinger bridges really needed? E.g., it is not clear what $\Phi$ and $a$ are, and I believe they are not used anywhere in the paper anyways, so this comment could be skipped.
- The link between (EmOT) and Schrodinger bridges is not well explained. A tree is defined in the second sentence, but it is only explained how it is related to (TreeSB) in the last sentence. Also you probably need a reference to equation (1) somewhere?
- The final paragraph "our framework" could be used to give a much more helpful summary of the main idea of this work. I suggest to state more clearly the motivation, contribution, and main result, and tone down the technical details (which are hard to understand at this point).

2. Proposition 3 should be presented in a self-sufficient way. It is not completely obvious here what $\mathbb{P}^n$ and $\pi^n$ denote.

3. What is the convergence criterion in Algorithm 1, line 5?

4. I feel that Section 4 does not really follow the flow of the paper, and I wonder why the authors don't stress the convergence result more as a main contribution of the paper. Also, I'm wondering if the result relies on any additional assumptions compared to previous results? Moreover, I'm a bit concerned about the fact that the one page of Section 4 relies on 10 pages of proofs, which are deferred to the appendix (and which I was not able to check). I find it problematic to have to hide 95% of the derivation for a strong theoretical result in the supplementary material. It might be preferable to not include this result in the paper due to space limitations, and publish it separately. This would also guarantee a thorough peer review of the result. In my opinion the paper makes a strong enough contribution without this Section.

5. For the synthetic two dimensional dataset it would be interesting to also see the true (possibly regularized) Wasserstein barycenter for reference.

6. For the MNIST application, I suggest to clarify in the main paper that you consider the problem in 784 dimensional space. This may not be clear, since images could be seen as distributions of 2-dimensional data, similar to the first numerical experiment.

7. What do you mean by one IPF cycle, and how do you chose the number of IPF cycles in the different experiments?

8. Since the paper presents methods for optimal transport problems with general tree structure, it would be very interesting to also include some experiments that are not a Wasserstein barycenter problem.

9. Typos:
43: hyperparameter and;
112: Section 5. We;
220 applied to trees

**Limitations:**

yes

---

> ### Author Rebuttal · Authors · 2023-08-09
>
> We thank the reviewer for their thorough review of our work.
>
> > My main question concerns the different update schemes mentioned in Section 3. The (mIPF) algorithm requires sampling from the whole tree [...]
>
> We clarify our training procedure. In Proposition 2 and Proposition 3, we draw a bridge between the mIPF static iterates (line 124) and the ones defined using a dynamic formulation. This connection is at the basis of TreeDSB. Contrary to the iterates of (mIPF), TreeDSB requires sampling trajectories at training and inference time. The description of the sampling procedure is presented in Figure 1. Our implementation follows the logic of this figure. We now provide some explanations regarding the example in Section 3 (lines 160-173).
>
> We acknowledge that the first sentence in line 163 might be confusing in light of Figure 1. The goal was to highlight that we can sample from $\pi^0$. However, to train the algorithm we only need to sample from a path (as highlighted in Figure 1).
>
> In our example, the initial measure is rooted at node $3$. Since in the next iteration we consider a tree rooted at node 1, we only need to consider samples along the path $3 \to 0 \to 1$, see the sentence "During [...]" (line 165). Similarly, as before we do not need to fully sample from $\pi^1$ to obtain $\pi^2$, and therefore acknowledge that the sentence "In order [...]" (line 167) might also be confusing, as we only need to follow the path $1 \to 0 \to 2$.
>
> We emphasize that during the training of the algorithm, we sample along paths and never sample from the full joint measure. So the option a) “sample along whole tree and learn reverse dynamics” is not used. Regarding the difference between b) and c), this is simply because each edge is parameterized by a different neural network and hence associated with its own loss function. The difference between b) and c) is only algorithmic.
>
> We thank the reviewer for this question which allowed us to clarify our main example. We hope to have resolved the reviewer’s concerns regarding the implementation of our algorithm.
>
> > In Section 2 [...] background material and setting are introduced in a clear and concise manner.
>
> > 1. Are all the details on dynamic and static versions of Schrodinger bridges really needed? [...]
>
> We would like to argue that the introduction of the dynamic and static Schrodinger Bridges are needed in order to explain the rationale behind Proposition 2 and Proposition 3. However, we acknowledge that the present introduction could be simplified (we will introduce the Brownian motion and not the Ornstein-Uhlenbeck).
>
> > The link between (EmOT) and Schrodinger bridges is not well explained [...]
>
> We acknowledge that the current presentation is misleading as the EmOT is not necessarily defined for tree. In the revised version of the paper, we will make clear that the link between EmOT and Schrodinger Bridges is valid in any multimarginal setting.
>
> > I suggest to state more clearly the motivation, contribution, and main result [...]
>
> We agree with the reviewer. In the revised version of the paper, we will highlight our contributions in a “Contributions” paragraph. We will relegate some technical details about mOT to the background section.
>
> > Proposition 3 should be presented in a self-sufficient way.
>
> In the updated version of the paper, we will be more precise regarding the definition of $\mathbb{P}^n$ (which is defined in Proposition 3). To clarify our results, we will merge Proposition 2 and Proposition 3.
>
> > I wonder why the authors don't stress the convergence result more as a main contribution of the paper.
>
> The motivation behind this theoretical section was to provide a mathematical background to our method. We were surprised to find that the literature has several gaps when it comes to entropic mOT. Section 4 is an attempt at filling some of them. We are constrained by space but would be happy to provide more details regarding the theoretical results including a sketch of proof.
>
> > For the synthetic two dimensional dataset it would be interesting to also see the true (possibly regularized) Wasserstein barycenter for reference.
>
> We have added this example to the one-page PDF associated with the general answer.
>
> > For the MNIST application, I suggest to clarify in the main paper that you consider the problem in 784 dimensional space.
>
> We agree with the reviewer that it is not uncommon in the OT community to treat images as densities. In contrast, we work in a high dimensional setting and will clarify this point.
>
> > What is the convergence criterion in Algorithm 1, line 5?
>
> The convergence criterion we choose is a fixed number of iterations. We stick with the common strategies in the training of neural network in deep learning which were used in [2].
>
> > What do you mean by one IPF cycle, and how do you chose the number of IPF cycles [...] ?
>
> By IPF cycle, we mean that we have iterated on all possible roots. For example, in the case of the star-shaped tree with leaves 1,2,3 and central node 0, the first IPF cycle allows us to obtain $\{\pi^1, \pi^2, \pi^3\}$. The number of IPF cycles (= number of IPF iterations/K) is chosen by hand like in other DSB approaches [2,3].
>
> > [...] to also include some experiments that are not a Wasserstein barycenter problem.
>
> Regarding the application of TreeDSB to other tree structured problems, we have described an application from single-cell genomics in the general answer. The goal is to predict the effect of a treatment but also to follow the time evolution on some cell features. We obtain a tree-structured problem which is not star-shaped.
>
> > Typos
>
> We have corrected all typos.
>
> [1] Multimarginal optimal transport with a tree-structured cost and the Schrödinger bridge problem. Haasler et al.
>
> [3] Diffusion Schrödinger bridge with applications to score-based generative modeling. De Bortoli et al.
>
> [4] Likelihood Training of Schrödinger Bridge using Forward-Backward SDEs Theory. Chen et al.

---

> > ### Comment · Reviewer_fo2c · 2023-08-14
> >
> > Thank you for the clarification.
> >
> > This paper makes some nice contributions. If the authors improve the presentation based on all the reviewers comments it will be more easy to follow and probably have a higher impact. Therefore, I'm increasing my rating to 7.

---

### Official Review · Reviewer_XAZX · 2023-07-06

**Soundness:** 3 good
**Presentation:** 3 good
**Contribution:** 3 good
**Rating:** 7
**Confidence:** 3

**Summary:**

Authors develop a new method for computing multi-marginal optimal transport in the case where the cost function is tree structured. Their algorithm cleverly exploits the dynamic formulation of the problem. The correctness of the algorithm is thoroughly demonstrated, and benefits are shown over a number of toy examples as well as competitive baselines involving the computation of the Wasserstein Barycenter.

**Strengths:**

It is a solid paper (theoretically and experimentally). Although the mathematical part is mostly technical and algorithm-related (but correct me if I am wrong), it is still quite sophisticated and praise deserving. Results section is compelling.

**Weaknesses:**

There are some weaknesses: first, the paper is slightly hard to read and while there are some helpful guides (as the first figure) at some points the paper gets helplessly hard to follow. Second, the paper should better explain what the contribution over the already existing methods (i.e Hasslear et al). It is not clear from the reading what this adds on top of that. Third, the paper is not compelling on its motivation. Why if the object of interest is a tree-based cost, all experiments focus on the barycenter problem? it gives the impression that the paper addresses a problem that doesn't exist in practice. Fourth, while the experimental results are super interesting, there is a lack of commentary as to why we should expect this.

**Questions:**

1) What is the difference between your method and the one of Haessler et al?
2)Could authors spell out applications where multimarginal OT inference in tree structured costs is important? Why is the focus only on the Barycenter?
3)Authors are improving upon the competitive baselines such as Cuturi and Doucet (table 1). Why the method was expected to bring this benefit? is it related to the tree-ness or the fact that it implements a Schrodinger Bridge?.
4)Likewise, why better results are obtained in data fussion? (Table 2). It seems to be implied that this is due to better performance in high dimensions. Why this should be the case?
5)Line 175 seems to be an error in the Ext() statement.
6)Type in line 305

**Limitations:**

n.a.

---

> ### Author Rebuttal · Authors · 2023-08-09
>
> We thank the reviewer for their feedback and their positive evaluation.
>
> > the paper is slightly hard to read [...]
>
> While we acknowledge that the current paper might be heavy with mathematical notation, we thought it was necessary to include them to precisely describe our TreeDSB methodology (Section 3) and our theoretical contributions (Section 4). However, we are happy to clarify the notation or to simplify our terminology if the reviewer has specific concerns regarding the presentation.
>
> > contribution over the already existing methods (i.e Haasler et al) [...]
>
> > What is the difference between your method and the one of Haasler et al?
>
> We have also provided a detailed answer to this comment in the main response. We agree that the static problem  is the main as the one from [1], as we acknowledge in our “Our framework” paragraph in Section 2 of the current paper. However, one of our key methodological contributions is to derive a **dynamic framework** to deal with this static problem as well as a practical algorithm to solve this dynamic formulation (see Section 3). The introduction of the dynamic formulation is absolutely central to our work as it allows us to leverage tools from the diffusion model literature to design an efficient algorithm. Other minor differences between our static framework and the one of [1] are reported in the general answer.
>
> > Third, the paper is not compelling on its motivation.
>
> We have provided a detailed answer to this point in the general response which we summarize here. First, TreeDSB introduces a new class of algorithms to solve EmOT. Contrary to variational-based or potential-based approaches (following the terminology of [2]), we do not rely on a static formulation of EmOT but on a dynamic one. This dynamic formulation allows us to leverage tools from the diffusion models literature. We believe that the idea of iterative refinement, i.e. solving a dynamic counterpart to the static problem, plays a key role in the efficiency and scalability of the method (see our general answer for an in-depth comment on the training-time, inference-time and memory costs properties of TreeDSB). We want to emphasize that if the static mOT is already available then there is no need to apply TreeDSB (or any other dynamic formulation). This is because the solutions of the dynamic mOT and the static mOT coincide up to marginalization. It is possible to view our work as a way to learn the static mOT solution by learning the dynamic one, the same way DSB [3] solves an entropic static OT problem using a dynamic formulation and leveraging tools from diffusion models.
>
> > Why if the object of interest is a tree-based cost, all experiments focus on the barycenter problem? it gives the impression that the paper addresses a problem that doesn't exist in practice.
>
> > Could authors spell out applications where multimarginal OT inference in tree structured costs is important? Why is the focus only on the Barycenter?
>
> We have written a detailed answer to this point in the main response. Regardless, we provide summarize it in what follows. Our methodology is defined for any tree. However, we agree with the reviewers that so far we have only applied our method to the case of Wasserstein barycenters. We chose the Wasserstein barycenter computation example as there already exists strong baselines. We agree that it would be valuable to test our algorithm on other tree structures. Regarding the application of TreeDSB to other tree structured problems, we have described an application from single-cell genomics in the general answer. We briefly summarize this new experiment. The goal is to predict the effect of a treatment but also to follow the time evolution on some cell features. We obtain a tree structured problem which is not star shaped.
>
> > Authors are improving upon the competitive baselines such as Cuturi and Doucet (table 1). Why the method was expected to bring this benefit? is it related to the tree-ness or the fact that it implements a Schrodinger Bridge?.
>
> > Likewise, why better results are obtained in data fusion? It seems to be implied that this is due to better performance in high dimensions. Why this should be the case?
>
> > Fourth, while the experimental results are super interesting, there is a lack of commentary as to why we should expect this
>
> We believe that the main reason behind the efficiency of the proposed approach in higher dimension is the dynamic formulation we introduce. Indeed by doing so we leverage efficient tools from the diffusion model literature which are known to scale with the dimension. We thank the reviewer for this question and have clarified this point in the revised version of the paper. We refer to the general answer for more details on the motivation and further discussions on the scalability of TreeDSB.
>
> > Line 175 seems to be an error in the Ext() statement
>
> > Typo in line 305
>
> We have corrected all typos.
>
> [1] Multimarginal optimal transport with a tree-structured cost and the Schrödinger bridge problem. Haasler et al.
>
> [2] Wasserstein iterative networks for barycenter estimation. Korotin et al.
>
> [3] Diffusion Schrödinger bridge with applications to score-based generative modeling. De Bortoli et al.

---

> > ### Comment · Reviewer_XAZX · 2023-08-15
> > **Thank you**
> >
> > Thanks a lot for your reply. I am keeping my score.

---

### Official Review · Reviewer_BvM6 · 2023-07-07

**Soundness:** 3 good
**Presentation:** 3 good
**Contribution:** 3 good
**Rating:** 7
**Confidence:** 3

**Summary:**

This paper generalizes the diffusion Schrödinger bridge framework to multidimensional problems with tree-structured quadratic cost function. The method builds on representing each egde in the graph as two SDEs (one for each direction) parameterized by neural networks. These are used to propagate the information (from the known marginals) in the graph. Under a set of assumptions the method is shown to converge in TV.

**Strengths:**

It is a nice extension of the Diffusion Schrödinger Bridge framework, which allows for addressing a relatively large class of multimarginal optimal transport problems where each marginal is a probability measure. Also it is nice that convergence of the method is provided.

**Weaknesses:**

I think it would be nice with a little bit more information on the neural nets, about their structure and how they are used and updated. See the question below.

**Questions:**

I do not see how the assumptions A1-A5 uses the neural nets. The neural nets typically have finite prespecified parameters. Can you explain how one can represent any measure (satisfying the assumptions), which are infinite dimensional objects.


**Limitations:**

The limitations are adequately addressed.

---

> ### Author Rebuttal · Authors · 2023-08-09
>
> Thank you for your comments, we appreciate your acknowledgment of the paper’s merits.
>
>
> > I think it would be nice with a little bit more information on the neural nets, about their structure and how they are used and updated.
>
> All information on the neural nets used in our experiments are given in Appendix G. The specific architecture of the neural network is given by a MultiLayer Perceptron (MLP) or a U-Net depending on the task (MLP for low dimensional data and U-Net for image data). The choice of these specific architectures follow from the ones made in DSB [1], which follow from the ones made in [2] in the case of the image data. We have added a sentence to refer to the supplementary in the main document.
> Finally, we briefly discuss what these neural networks model.  We recall that each edge {v,v’} of the tree is associated with 2 neural networks, and each one corresponds to a direction of this edge, either (v->v’) and (v’->v). Let {v,v’} be an edge. At iteration $(n+1)$ of TreeDSB, the neural net (v->v’) is updated if and only if the directed edge (v’,v) is in the path between the previous root chosen at iteration $n$ and the new root chosen at iteration $(n+1)$. This neural network models the drift of the SDE that describes the path measure $\mathbb{P}^{n+1}_{(v’,v)}$. Through an entire IPF cycle, each network is updated only once, since each edge is visited once in both ways. Regarding how they are updated, the losses associated with these networks are the ones associated with the time-reversal in diffusion models. In particular, we have used the mean-matching loss from [1] to learn the time-reversal along each edge. We will include some details about the mean-matching loss as well as a complete pseudo-code of the training algorithm in the updated supplementary.
>
> > I do not see how the assumptions A1-A5 uses the neural nets. The neural nets typically have finite prespecified parameters. Can you explain how one can represent any measure (satisfying the assumptions), which are infinite dimensional objects.
>
> We want to clarify that in the theoretical section of the paper (Section 4), we do not use the neural networks. This section is merely concerned with the static problem. In particular, we focus with the well-posedness of the mOT problem (in particular, we show that the solution of the mOT problem and the mIPF iterates can be obtained by updating dual potentials) and the convergence of mIPF procedure in that setting. The methodology introduced in our paper is an attempt at efficiently learning the mIPF iterates by leveraging a dynamic formulation and tools from the diffusion model literature.
> The main motivation for establishing such results is that **we could not find any literature providing a proper mathematical background** for the treatment of multimarginal Schrodinger bridges. We see our contribution as a first step towards more theoretical results in the field of multimarginal Schrodinger bridges. In particular, in future work we would like to investigate extension of the theory to the full dynamic formulation and to provide convergence bounds of the mIPF for the (approximate) scheme based on TreeDSB.
>
> [1] Diffusion Schrödinger bridge with applications to score-based generative modeling. De Bortoli et al.
>
> [2] Score-Based Generative Modeling through Stochastic Differential Equations. Song et al.

---

> > ### Comment · Reviewer_BvM6 · 2023-08-17
> >
> > Thank you. I am keeping my score.

---

### Official Review · Reviewer_GsNC · 2023-07-07

**Soundness:** 2 fair
**Presentation:** 2 fair
**Contribution:** 2 fair
**Rating:** 4
**Confidence:** 2

**Summary:**

The authors consider the multi-marginal optimal transport (mOT) with entropic regularization and a tree-structured quadratic cost. The authors propose to address it by developing the Tree-based Diffusion Schrodinger Bridge (TreeDSB), i.e., dynamic and continuous state-space of the multimarginal Sinkhorn algorithm. The authors apply the proposed approach to compute Wasserstein barycenter, i.e., equivalent to mOT with star-shaped tree.

**Strengths:**

+ The authors propose Tree-based Diffusion Schrodinger Bridge, a dynamic and continuous state-space counterpart of the multimarginal Sinkhorn algorithm.
+ The authors illustrate some advantages of the proposed approach for Wasserstein barycenter.

**Weaknesses:**

+ The motivation of developing the dynamic and continuous state-space counterpart of the multimarginal Sinkhorn algorithm and experiments seem quite weak. For Wasserstein barycenters and problems illustrated in the experiments, it suffices to address the multimarginal OT (i.e., the static problem).
+ Although there is some discussion in the appendix, it is essential that the submission is very closely related to (Hassler et al. 2021). It is an important part and should elaborate with details in the main manuscript to emphasize the contribution.
+ The notations seem heavy and hard to follow. It is better to simplify the notations or give more explanation for the notations when they are introduced. At the present, it is hard to follow the meaning of the notation system.

**Questions:**

+ It is not clear why one needs to address the dynamic and continuous state-space of the multimarginal Sinkhorn. It suffices to consider the multimarginal OT directly for Wasserstein barycenters and problems in the experiments. Could the authors comment on this?
+ Although the authors give some discussion about the relation with Hassler et al. (2021) in the appendix, it is essential to clarify their difference in the main manuscript (e.g., in related work section) to emphasize the contribution. Could the authors summary their main differences? (since they are very closely similar).
+ In line 110, $S$ is the set of probability measures on the “leaves” of $T$. However, in line 118, why $r \in S$ (it seems that $r$ is the root of tree T as in line 113)? So, what does the tree $T$ look like?
+ Please explain the Figure 2, its description is too short and hard to follow.
+ As in line 157-158, there are $2|E|$ neural networks, how are these networks related?
+ Kitagawa and Pass (2015) showed the equivalent between multimarginal OT and the Wasserstein barycenter. Could the authors comment the result in Proposition 8 and the claim in line 271-272 with those in Kitagawa and Pass (2015)?

Ref: Kitagawa, J. and Pass, B., 2015, September. The multi-marginal optimal partial transport problem. In Forum of Mathematics, Sigma (Vol. 3, p. e17). Cambridge University Press.
+ In line 284, how does the choice of $\epsilon$ affect the result, and how to choose it in applications?
+ For results in Tables 1, 2, Figures 6, 7, it is better to also report the corresponding time consumption?
+ It seems unclear how to choose $\epsilon$ (hyperparameters) for the barycenter problem? Does it affect the result? Some hyperparamters seem missing, e.g., how many free supports are used for fsWB?
+ For results in Figures 6 and 7, how to interpret these results? (especially Figure 7)
+ in line 330, “a scalable scheme”? it is unclear whether the proposed approach is scalable from the proposed algorithm and experiments? Please elaborate the scalability of the proposed method.

Some minor points:
+ In line 2, what is $C([O, T], R^d)$? There is no explanation for the new notation $C( , )$
+ in line 78, it is better to explain the Stochastic Differential Equation form, e.g., there is no explanation about several new notations, $X_t, a, B_t$
+ Please explain the Figures. The description for each Figure is very short, it is unclear the meaning of each Figure, e.g., how to understand the Figures 2, 3?
+ In line 125, what is the role of $K$?
+ In Proposition 1, what is the relation between $T_{K-1}$ and $T$ (although there is some explanation in line 71-72, it is hard to follow it). It is better to consider to simplify the notations or explain the notations with more details.

---

Thank you for the rebuttal.

**Limitations:**

The authors have discussed the limitation of the work. However, there is no discussion about the potential negative societal impact of their work.

---

> ### Author Rebuttal · Authors · 2023-08-09
>
> Thank you for taking the time to review our submission and for your constructive feedback.
>
> > The motivation [...] seem quite weak
>
> We have provided a detailed answer to this point in the general response. First, TreeDSB introduces a new class of algorithms to solve EmOT using a dynamic framework. This formulation allows us to leverage tools from the diffusion models literature. We believe that the idea of iterative refinement, i.e. solving a dynamic counterpart to the static problem, plays a key role in the efficiency and scalability of the method.
>
> > closely related to (Hassler et al. 2021) [...]
>
> We have also provided a detailed answer to this comment in the main response. We emphasize that the static problem is the same as the one from [6] (see “Our framework” paragraph in Section 2). However, one of our key contribution is to derive a **dynamic framework** to deal with this static problem as well as a practical algorithm to solve this new formulation (see Section 3).
>
> > What is $S$ ? [...]
>
> S is the set of leaves. As every leaf of the tree is associated with a marginal in a unique way, we make the slight abuse of notation to say that $S$ is the set of marginals of the leaves. We will correct this.
> The tree $T$ corresponds to the graphical model of the prior joint distribution. The nodes are associated with the marginals while the edges are associated with the couplings.
>
> > The description for each Figure is very short, [...]
>
> **Figure 2** provides an illustration of how to bridge the edge iterates from Proposition 3 (i.e., the path measures $P^n_{(v,v')}$), which solve the dynamic Schrödinger Bridge (SB) problem, to the edge iterates of mIPF (i.e., the couplings $\pi^n_{(v,v')}$), which solve the static SB problem. This is done by the operator $\text{Ext}$ which takes as input a path measure $P$ (blue line in Fig. 2), and outputs the coupling $P\_{0,T}$. In our case, we have $\pi^n_{(v,v')}=\text{Ext}(P^n_{(v,v')})$.
>
> **Figure 3** illustrates Proposition 3 with an example. Assume that $T$ is rooted in vertex 3 at iteration $k$ of our algorithm. The directed edges of $T$ are the red edges. Assume that we want to solve the dynamic formulation of mIPF at iteration $k+1$, where the marginal constraint is on vertex 4. Then, by Proposition 3, we simply have to operate **the time reversal of the path measures corresponding to the red edges that lie in the path between 3 and 4** (yellow dotted path). Once this is done, we can sample from the $(k+1)$-th iterate by following the red edges, starting from 4.
>
> > How are the NNs related to each other ?
>
> We have two NNs per edge. Each NN has the same architecture.  This is similar to other neural-network based approaches [1,2,3,4].
>
> > Comparison with Kitagawa and Pass ?
>
> Thank you for pointing us towards [7]. In [7] the authors consider a *partial* multi marginal OT problem while we consider an entropic regularized approach. As a result, our static formulation and the one from [7] are drastically different.
>
> > How does the reg affect results in 2D ?
>
> The computational trade-off of TreeDSB recovers the classic behaviour of OT methods: whenever $\varepsilon$ is chosen too small, the method requires more iterations to converge. This trade-off is illustrated in the results of the 2D experiments in the PDF, where we consider 50 IPF cycles for $\varepsilon=0.1, 0.2$ and 60 IPF cycles for $\varepsilon=0.05$. As $\varepsilon$ decreases, the obtained barycenter is less noisy, but requires more IPF iterations to converge. We will add these considerations in the section 'Experiments' in the main manuscript.
>
> > Figs 6 and 7, how to interpret these results?
>
> In Fig.6 we show non-regularized [2,3] and regularized [4] baselines for the barycenter computation. We also present the barycenter obtained with TreeDSB which is visually closer to the ground-truth computed with [2] than the competing method [4].
> In Fig.7 we illustrate that TreeDSB not only reconstruct the barycenter distribution but also the distributions on the leaves.
>
> > in line 330, “a scalable scheme”?
>
> > report [...] time consumption?
>
> We provide a detailed answer to this point in the general response. Compared to other out-of-sample competing  [1,2,3,4] methods, TreeDSB enjoys a similar training time and memory cost. In particular, the number of NNs also scales linearly with the number of edges.
>
> > What is $C([0,T], \mathbb{R}^d)$ ?
>
> $C([0,T], \mathbb{R}^d)$  denotes the continuous functions from [0,T] to $\mathbb{R}^d$. We will add it to the notation section.
>
> > explain the SDE form [...]
>
> Path measures are defined at the beginning of Section 2. In line 78, the path measure $\mathbb{Q}$ is associated with the process $(X_t)$, i.e. $(X_t) \sim \mathbb{Q}$. $(B_t)_{t \geq 0}$ is a  $d$-dimensional Brownian motion. We will precise this in the new version of the paper.
>
> >  role of $K$?
>
> This quantity is defined in line 115 “In what follows, we define K as the number of leaves of $T$”. To perform a full cycle of IPF, we have to solve iteratively on the whole set of the leaves which explains $\mathrm{mod} (K)$.
>
> > relation between $T_{K-1}$ and $T$ ?
>
> T is the undirected tree used in the general formulation of SB. $T_v$, for some vertex v, is the directed version of T rooted in v, which is uniquely defined from T. The link between general trees and rooted trees is explained in our notation section.
>
> [1] Wasserstein iterative networks for barycenter estimation. Korotin et al.
>
> [2] Scalable computations of Wasserstein barycenter via input convex NNs. Fan et al.
>
> [3] Continuous Wasserstein-2 barycenter estimation without minimax optimization. Korotin et al.
>
> [4] Continuous regularized Wasserstein barycenters. Li et al.
>
> [5] Tackling the Generative Learning Trilemma with Denoising Diffusion GANs. Xiao et al.
>
> [6] Multimarginal optimal transport with a tree-structured cost and the Schrödinger bridge problem. Haasler et al.
>
> [7] The multi-marginal optimal partial transport problem. Kitagawa and Pass.

---

> > ### Comment · Reviewer_GsNC · 2023-08-18
> >
> > Thank you for the rebuttal. I have no other raised points.

---

### Official Review · Reviewer_2tnG · 2023-07-07

**Soundness:** 2 fair
**Presentation:** 2 fair
**Contribution:** 3 good
**Rating:** 5
**Confidence:** 3

**Summary:**

Schröninger bridges (SB) can be understood as dynamical counterpart of the Entropic Optimal Transport problem.

This paper introduces the Tree Diffusion Schröninger Bridge (TDSB) model, which can be seen as a multi-marginal generalization of SB when the cost $c$ has a tree structure: given $\ell$ marginals $\pi_1,\dots,\pi_\ell$, and a tree $T = (V,E)$ with $|V| = \ell$ nodes, one has $c(x_1,\dots,x_\ell) = \sum_{(v,v') \in E} \| x_v - x_{v'}\|^2$. Let $S$ denote the leaves of $T$.

In that setting, the goal of the paper is to solve the TreeSB problem:
$$ \min_\pi \mathrm{KL}(\pi | \pi^0),\quad \pi_i = \mu_i,\ \forall i \in S. $$
where the $\mu_i$ are prescribed marginales, and $\pi^0 = \pi_r^0 \bigotimes_{(v,v')} \pi^0_{v,v'}$ is a reference measure that factorize along $T$, and $r \in V$ is identified as a root of $T$, and the factors $\pi^0_{v,v'}$ are chosen as normal distributions.

If $T$ has a star-structure, this problem is equivalent to the (doubly-regularized) Wasserstein barycenter problem (in sense that, at optimality, the marginal of $\pi$ in the center of the star is the Wasserstein barycenter of the marginals on the leaves).

The contribution of the work is to solve this problem using an iterative algorithm along _paths_ of $T$ which can be summarized as a succession of $\mathrm{KL}$ projections where only one of the marginals are constrained; if the tree is only made of two nodes (and one edge) we retrieve the Sinkhorn algorithm. In practice, the transition along a given edge (during one step of the algorithm) are performed by SDE of the form $\dot X_t = f_\theta(X_t) \mathrm{d}t + \mathrm{d} B_t$, where $B_t$ is the Brownian motion, and $f_\theta$ is a drift that is learned (and parametrized by a neural network). This leads to an iterative scheme where one repetitively select path between two leaves and learns the drift to solve the KL projection along the path.

From a theoretical standpoint, authors prove that (assuming the intermediate step converge, e.g. the drifts $f_\theta$ end up being optimal) the proposed algorithm solves (asymptotically) the TDSB problem.

From a numerical perspective, they showcase the practicability of their approach on synthetic and simple benchmark datasets.

**Strengths:**

- I find the developed approach quite elegant. It mixes different notions in optimal transport literature (multi-maginal OT, entropic OT/Schrödinger bridge, doubly-regularized Wasserstein barycenters, parametrization of "transport maps" (the drifts) by neural networks, etc.). It is nice to see all these approaches implemented in a _practical_ setting (which can be implemented).
- Comparison with previous work is well presented.
- The convergence results of the proposed approach are strong.


**Weaknesses:**

## Numerical experiments

My main remark would be about the numerical experiments. They are quite limited in their comparison with state-of-the-art methods and rather acts like proof-of-concept. In details:
- They do not showcase how the proposed approach solves new problems. Though reasonable, none of the results showcase feel "groundbreaking". This may just be a matter of exposition, but right now, loosely speaking, I am not "impressed" by the experimental results. For instance, the choice of synthetic datasets (Figure 4) is questionable in that the actual barycenter is not very intuitive and thus it is hard to visually tell whether the displayed output (Figure 5) is good or not; in addition, the "consistency" of the proposed approach is not striking (I admit that there is a decent similarity between the three outputs, but we cannot faithfully consider that the model did converge).
- The comparison with SotA methods is incomplete; e.g. there is no running time involved. Given the (maybe only superficial) complexity of the proposed approach---such as the need for a possibly huge amount of NN (which are not very small according to Section G, as they are MLPs), one may be feared by a possibly huge running time (compared to simpler approaches like, say, the one of Cuturi and Doucet). If there is a computational tradeoff (better result for longer training time), it must be mentioned and discussed in details.
- I would expect a comparison with the _Iterative Bregman Projection_ of (Benamou et al., 2015), given the "similarity" (in that they are iterative KL projections) with the proposed approach (and I would consider IPB as being "more SotA" than Cuturi&Doucet, for instance; for instance the IPB paper claims in Section 1.3 that "IPB converges (...) orders of magnitude faster than Cuturi&Doucet").
- A bit minor, but the description of the _subset posterior aggregation_ experiment is very minimalist. It is hard for the reader that is not already familiar with such experiment to have an idea of what is done there.

**Questions:**

- My main question, which can be developed (see below) would be _why should one use this approach rather than existing ones?_ (e.g. IPB, fsWb, etc.). Of course, the paper gives some insight, but:

1. As said above, there is no insight on the computational burden of the proposed approach.
2. While I find very interesting that the approach works on any tree-structured coast, the work only focuses on Wasserstein barycenter, somewhat limiting its impact (that is, one strength of this work is that it solves a more general problem than previous Wasserstein-barycenter-dedicated methods). Giving an example + experiment where a (non-stared) tree cost is involved would be insightful (it would give a situation where the proposed approach as no challenger, as far as I know!).

**Limitations:**

authors have addressed the limitations of their approach.

I do not identify ethical concern specific to this work.

---

> ### Author Rebuttal · Authors · 2023-08-08
>
> Thank you for your comments and thoughtful questions. Below we provide a detailed answer to the reviewer’s concerns.
>
> > Why should one use this approach rather than existing ones?
>
> We have provided a detailed answer to this point in the general response which we summarize here. First, contrary to previous out-of-sample methods, TreeDSB does not rely on a static formulation of EmOT but on a dynamic one, which allows us to leverage tools from diffusion models. We believe that the idea of solving a dynamic counterpart to the static problem plays a key role in the efficiency and scalability of the method. Additionally, the entropic regularization induced by our dynamic formulation is different from the one usually considered in static formulations and allows us to compute Wasserstein barycenters with low entropic regularization while avoiding the numerical issues of having to choose $\varepsilon$ too small. This is well highlighted in our experiments on MNIST dataset, see Figure 6, where our estimated barycenter samples are close to the ones given by unregularized methods while the samples from [1] are of poor visual quality.
>
> > They do not showcase how the proposed approach solves new problems.
>
> > While I find very interesting [...] as far as I know!)
>
> We have provided a detailed answer to this point in the general response which we summarize here. We chose the Wasserstein barycenter problem because there already exists strong baselines for this problem, and therefore, we can compare TreeDSB to these existing methods. We agree with the reviewer that it would be valuable to apply TreeDSB in a non star-shaped setting. For instance, in the case of single-cell genomics, it would be interesting to infer the distribution of features of cells at different time steps (temporal tracing) as well as the probability paths between treated and untreated states (effect of the treatment), see the general answer for details. We leave the application of TreeDSB to this framework for future work as it requires some domain knowledge of single-cell genomics and additional data collection.
>
> > I would expect a comparison [...] faster than Cuturi&Doucet").
>
> We would like to emphasize that the experiments in [2], where a comparison is made with [3], are only run on 2D data (see Figures 3 and 4 in [2]). For our general setting, we chose [3] as a baseline, following the comparison made in [4]. Nonetheless, we provide in the PDF a numerical result of this method in the 2D setting.
>
> > Though reasonable, [...] that the model did converge).
>
> We emphasize that the only simple case where we have access to the true Wassertein barycenter is the Gaussian setting (for which we already show that TreeDSB performs well). We chose these 2D synthetic datasets as an illustration of our method, and aimed at showing that the marginals on the leaves are well recovered. In the PDF, we give improved results in this setting (which have better consistency on the barycenter node) with different values of $\varepsilon$. We acknowledge that TreeDSB performs worse than in-sample methods in this setting, which can be explained by the fact that the diffusion-based setting suffers from compactness of the supports of the 2D distributions (see vertex 3). In this very specific case, it would be preferable to use [2] and [3] instead of TreeDSB. However, we highlight that [3] performs worse than TreeDSB in higher dimensional settings.
>
> > The comparison with SotA methods [...] the one of Cuturi and Doucet).
>
> > As said above, there is no insight on the computational burden of the proposed approach.
>
> We have provided a detailed answer to this point in the general answer. We briefly summarize it. We highlight that the training time of TreeDSB should not be compared with the complexity of the algorithms from [2] and [3] which are in-sample methods. Instead, TreeDSB should be compared with the out-of-sample neural network-based methodologies such as [1]. In our experiments, we observed that the training time of TreeDSB and the algorithm from [1] were comparable. We acknowledge that the inference time of TreeDSB is however longer since it requires to sample from a SDE, a limitation common to all diffusion-model based techniques which could be addressed by using distillation methods.
>
> > If there is a computational tradeoff (better result for longer training time), it must be mentioned and discussed in details.
>
> The computational trade-off of TreeDSB recovers the classic behaviour of OT methods: whenever $\varepsilon$ is chosen too small (i.e., we expect to be closer to the true barycenter), the method takes more time to converge. This trade-off is well highlighted in the results of the 2D experiments in the PDF, where we consider 50 IPF cycles for $\varepsilon=0.2, 0.1$ and 60 IPF cycles for $\varepsilon=0.05$. As $\varepsilon$ decreases, the obtained barycenter is less noisy, but requires more IPF iterations to converge. We will add these considerations in the section 'Experiments' in the main manuscript.
>
> > A bit minor [...] an idea of what is done there.
>
> We briefly explain here how posterior aggregation works. In this setting, we consider some dataset, which is splitted into $\ell$ subsets, and a Bayesian model. We assume that one posterior distribution is computed for each data subset. Then, posterior aggregation consists in computing the Wasserstein barycenter of these $\ell$ posterior distributions, expecting that it recovers the posterior distribution from the whole dataset, see [5] for a theoretical justification.
>
> [1] Continuous regularized Wasserstein barycenters. Li et al.
>
> [2] Iterative Bregman projections for regularized transportation problems. Benamou et al.
>
> [3] Fast computation of Wasserstein barycenters. Cuturi and Doucet.
>
> [4] Scalable computations of Wasserstein barycenter via input convex neural networks. Fan et al.
>
> [5] Scalable Bayes via barycenter in Wasserstein space. Srivastava et al.

---

> > ### Comment · Reviewer_2tnG · 2023-08-12
> > **Thanks**
> >
> > Thank you for taking time to answer my comments, along with the global answer with additional experiments. I am increasing my grade.

---

### Author Rebuttal · Authors · 2023-08-08

We thank the reviewers for their insightful comments and are encouraged by their positive feedback regarding the proposition and soudness of our work. We provide detailed responses to each reviewer but summarize here their main feedback.

### 1. Motivation for the application to the Wasserstein Barycenter (WB) problem

First, we emphasize that our method is **out-of-sample** (OuS), similarly to recent non-regularized methods [1,2] or regularized ones [3]. This is in contrast to algorithms such as [4] or [5], which are **in-sample** (IS). Contrary to IS methods, OuS ones do not require re-running the full procedure when given a new datapoint.

Second, our WB formulation (line 225) is different from the one considered in the existing literature $\sum_{i=1}^\ell W^2_\{2,reg}(\mu, \mu_i)$, since it has an additional entropic term $\ell \varepsilon H(\mu)$. This term **penalizes the entropy of the barycenter** instead of promoting it, as done classically. Consequently, even for large values of $\varepsilon$, the entropic regularization on the barycenter may remain low [6], which allows us to avoid the numerical issues of having to choose $\varepsilon$ too small.

Finally, TreeDSB is, to the best of our knowledge, the **first methodology to extend ideas from diffusion-based models to the computation of WBs**. In particular, we believe that the idea of iterative refinement, i.e., solving a dynamic counterpart to the static problem plays a key role in the efficiency and scalability of the method.

### 2. Extension of TreeDSB to other trees

Although TreeDSB is defined for any tree, we agree with the reviewers that so far we have only applied it to star-shaped trees. We chose this specific example as there already exists strong baselines. We agree that it would be valuable to test our algorithm on other tree structures. In what follows, we briefly describe **the example of a tree-based setting in single-cell genomics** that we are currently investigating.

Assume that we only have access to distributions of features of cells from a subset of the following nodes: $T$ (treated) or $U$ (untreated), at times $t_i$. This gives us a **non-star graph**, where we aim at modeling the edges $(T,t_i \to T, t_\{i+1})$, $(U,t_i \to U, t_\{i+1})$ (temporal tracing) and $(U,t_i \to T, t_i)$ (effect of the treatment), see [8]. As this application requires some prior domain knowledge in biology, we do not include it in the current work.

### 3. Comparison with [9]

Our main contribution is to find a **dynamic counterpart** to the static problem of [9]. This allows us to introduce ideas from Diffusion Schrödinger Bridge (DSB) [7], which are known to scale well with data dimension. We also highlight a few limitations of our framework compared to the one of [9], that are inherent to all existing dynamic versions of entropic OT [7,10]. First, our work is limited to the case where the cost is quadratic. Moreover, contrary to [9], the distributions on the nodes in our setting are required to be defined on the same state-space. This prevents us from evaluating our algorithm against [9] to estimate ensemble flows on a hidden Markov chain, see [9, section 6].

### 4. Running time

First, regarding the training time, we should not compare TreeDSB with IS procedures like [4] and [5], which do not rely on the use of neural networks. In contrast to IS methods, **OuS do not need to be rerun** when presented with a new sample, but require a  training procedure. For our experiments on Gaussian datasets and posterior aggregation, the training of TreeDSB required approximately 12 hours while the method of [3] required 8 hours. The memory cost of TreeDSB is similar to the one of competing methods.

Second, we acknowledge that at inference time, TreeDSB is slower than the competing one-shot methods [1,2,3], since it relies on a dynamic formulation. This is a well-identified bottleneck of diffusion models [11], that can be reduced through distillation procedures, see [12] for instance. In our experiments, we considered 50 time-steps per edge (3.5 s to infer 100 MNIST barycenter samples starting from a leaf on a A100 GPU).

### 5. Experiments in the PDF

In the PDF, we present **improved visual results** of the experiments conducted in the main paper (2D and MNIST datasets), which were obtained by slightly increasing the variance of $\mu_0$ (the initial barycenter measure). We also present new experimental results. In the 2D case, we illustrate the classic OT trade-off between the accuracy of the method and its convergence rate through the choice of the regularisation parameter ($\varepsilon=0.2, 0.1,0.05$) (Figure 1), and also present the results from SotA IS methods [4,5] (Figure 2). Finally, we present Wasserstein barycenters obtained for the MNIST experiment (Figure 3) with this new initialization.

[1] Scalable computations of Wasserstein barycenter via input convex neural networks. Fan et al.

[2] Continuous Wasserstein-2 barycenter estimation without minimax optimization. Korotin et al.

[3] Continuous regularized Wasserstein barycenters. Li et al.

[4]  Fast computation of Wasserstein barycenters. Cuturi and Doucet.

[5]  Iterative Bregman projections for regularized transportation problems. Benamou et al.

[6] Doubly regularized entropic Wasserstein barycenters. Chizat.

[7] Diffusion Schrödinger bridge with applications to score-based generative modeling. De Bortoli et al.

[8] Neural Unbalanced Optimal Transport via Cycle-Consistent Semi-couplings. Lübeck et al.

[9] Multimarginal optimal transport with a tree-structured cost and the Schrödinger bridge problem. Haasler et al.

[10] Likelihood training of Schrödinger bridge using forward backward sdes theory. Chen et al.

[11] Tackling the Generative Learning Trilemma with Denoising Diffusion GANs. Xiao et al.

[12] Knowledge Distillation in Iterative Generative Models for Improved Sampling Speed, Luhman et al.

---

### Decision · Program_Chairs · 2023-09-21

**Decision:**

Accept (spotlight)

**Comment:**

All the reviews are positive. The paper is nicely written and proposes interesting theoretical and numerical contributions. The rebuttal included additional numerical results, which are convincing. It helped clarify several raised issues, in particular regarding computational complexity and connexions to closely related works. I recommend acceptance.